# Macroprudential Policy in a Heterogeneous Environment—An Application of Agent-Based Approach in Systemic Risk Modelling

**DOI:** 10.3390/e22020129

**Published:** 2020-01-21

**Authors:** Jagoda Kaszowska-Mojsa, Mateusz Pipień

**Affiliations:** 1Institute of Economics, Polish Academy of Sciences, Nowy Swiat St. 72, 00-330 Warsaw, Poland; jagoda.kaszowska@inepan.waw.pl; 2Department of Empirical Analyses of Economic Stability, Cracow University of Economics, Rakowicka St. 27, 31-510 Cracow, Poland

**Keywords:** systemic risk, macroprudential policy, agent-based modelling, inequality, central-banking

## Abstract

Assessment of welfare effects of macroprudential policy seems the most important application of the Dynamic Stochastic General Equilibrium (DSGE) framework of macro-modelling. In particular, the DSGE-3D model, with three layers of default (3D), was developed and used by the European Systemic Risk Board and European Central Bank as a reference tool to formally model the financial cycle as well as to analyze effects of macroprudential policies. Despite the extreme importance of incorporating financial constraints in Real Business Cycle (RBC) models, the resulting DSGE-3D construct still embraces the *representative agent* idea, making serious analyses of diversity of economic entities impossible. In this paper, we present an alternative to DSGE modelling that seriously departs from the assumption of the representativeness of agents. Within an Agent Based Modelling (ABM) framework, we build an environment suitable for performing counterfactual simulations of the impact of macroprudential policy on the economy, financial system and society. We contribute to the existing literature by presenting an ABM model with broad insight into heterogeneity of agents. We show the stabilizing effects of macroprudential policies in the case of economic or financial distress.

## 1. Introduction

The new setting of financial supervision tailored after the global financial crisis of 2008–2009 has highlighted the need for research on the nature and measurement of risk in the financial system, also called systemic risk [1,2,3,4]. In response to the problems that occurred during the global financial crisis, the Basel III regulatory framework for financial institutions was adopted in 2010–2011. In the updated version of the Basel document the capital and liquidity requirements were established. In addition, the methods of conducting stress tests in the financial system have been subject to revision. Basel III was designed to strengthen the effects of banks’ capital requirements by increasing the liquidity of the banking sector and reducing leverage undertaken by banks. In the European Union (EU), the implementing act of the Basel Agreements has been issued in the form of a new legislative package covering CRD IV/CRR (i.e., CRD IV Directive No. 2013/36/EU on access to the activity of credit institutions and the prudential supervision of credit institutions and investment firms and CRR Regulation No. 575/2013 on prudential requirements for credit institutions and investment firms).

The literature on selected macroprudential policy tools [5,6] presented in Basel II and III has been mainly focused on theoretical and empirical research on linkages between real sector and the financial system. According to International Monetary Fund, Financial Stability Board and Bank for International Settlements (IMF-FSB-BIS) [7], the assessment of the *effectiveness* of macroprudential policies includes: the assessment of the extent to which the macroprudential instrument increases the resilience of the macro-financial system; and the assessment of the extent to which the macroprudential instrument has impact on credit dynamics and asset prices (the impact on the cycle). The adoption of the new institutional framework for macroprudential supervision in the EU Member States took place in most countries during last three years. Therefore, the results of empirical studies on the effectiveness of macroprudential instruments are biased by substantial uncertainty. Alternatively one may carry out counterfactual analyses on the impact of a *combination* of macroprudential instruments on a stylised economy using the following simulation methods: dynamic stochastic general equilibrium models (DSGE) or non-equilibrium models (e.g., ABMs) [8]. Although from the presentation of ABM models in opposition to DSGE models, the conclusion can be drawn that the DSGE models are always less useful, it is important to remember that DSGE models are an extremely important tool used mainly in central banking. A defence of the legitimacy of using DSGE models even after the financial crisis can be found in Christiano et al. [9].

Analyses of the impact of macroprudential policies on the financial system and the real part of the economy have been primarily focused on capital requirements [10,11,12,13], countercyclical capital buffer [14,15,16,17] and leverage [18]. In the literature, we can also observe successive attempts to incorporate stylised macroeconomic and macroprudential policies in the form of financial frictions into Dynamic Stochastic General Equilibrium (DSGE) models [19,20,21,22,23,24,25]. The main goal of these attempts was to examine the effects of monetary policy or the general equilibrium welfare effects of capital requirements and leverage. Despite the role these studies played in formulating theoretical background to design of macroprudential policy, the assumptions of DSGE models are subject to critique [26,27]. DSGE models share the assumption of a perfectly rational representative agent that dynamically optimizes the use of resources. Failure to take into account the heterogeneity of agents in most DSGE models is particularly acute from the perspective of social welfare analysis performed within an equilibrium environment [28].

Both empirical and theoretical studies on the effects of the Basel III have led to the formulation of criticisms of the adopted regulatory framework. In particular, some researchers highlighted insufficient risk and uncertainty sensitivity of macroprudential tools, over-reliance on external rating regulations, improper tool calibration and lack of synchronization of adopted rules at institutional and national level. EU expert groups are still working to incorporate changes within these areas into the Basel IV framework. New research tools are required to examine the impact of regulatory changes on the economy, financial sector and society. The new setting should allow greater flexibility in modelling of risk-taking, risk aversion and decision-making under uncertainty. Consequently assumptions of macromodels should be more realistic to allow for a study of changes in welfare in heterogeneous economy beyond the social planner framework.

The aim of this paper is to analyse the impact of selected macroprudential policy tools on the economic and financial system using agent-based modelling (ABM). Modelling of interactions between agents within the ABM approach was confronted with the DSGE model with three layers of default (‘3D’) [21], which is currently used by experts within the European Systemic Risk Board (ESRB) and European Central Bank (ECB).

This paper contributes to the existing literature of agent-based modelling through detailed and relatively broad insight into heterogeneity of agents. In the approach taken, decision-making rules, preferences and behaviours may vary across units. In our model, all agents, ie banks, individuals, households, consumers, firms, establishments, industries, suppliers, properties, are heterogenous.

We conducted extended simulation experiments that were based on an ABM model that had been calibrated to reflect the features of a small open economy. Our choice was Poland as an exemplar case. The reason for calibrating the model relying on Polish data is that among the EU countries, the Polish economy is relatively small, open and strongly connected to the rest of the European Union countries. Moreover, the Polish banking system still remains strongly influenced by investors from the European Union, who treat Poland as a host country. Generally, the smaller Central and Eastern European (CEE) countries that host foreign financial institutions are exposed to various dimensions of systemic risks more strongly. At the same time, the degree of the development of financial intermediation is relatively low, which results in a rather weak credit channel, especially in the case of investments. Although the financial system in Poland generates limited systemic risk, it is more vulnerable to regulatory arbitrage and the propagation of the shocks that are caused by the activity of international financial groups.

Consequently, the CEE economies and other emerging economies may need to conduct a more active macroprudential policy because of the higher risks that stem from volatile capital flows or credit booms and so forth. These issues also relate to the Polish economy and its financial system. Hence, both the ABM model and the simulations presented in our paper are valuable for gaining a detailed insight into the effects of macroprudential policy, especially in the case of small emerging open economies.

In order to study the macroeconomic effects of macroprudential instruments and their interaction with monetary policy in the case of a hypothetical small open economy, Aoki et al. [29] applied a DSGE framework. The analysed model captured some critical features of the emerging market economies with macroprudential instruments that were defined as the capital requirements that were imposed on banks and a tax on foreign currency (FX) lending. However, there are some relevant aspects that were not taken into consideration in the Aoki et al. [29] model. For example, the possibility of the government or central bank intervening in the foreign exchange markets through the use of official foreign reserves is not discussed. Moreover, what is missing in the model is a more flexible specification of international capital flows (no equity flows or foreign direct investment) and the role of cross border gross flows, which could play a destabilizing role for financial stability. The ABM construct that is presented in our paper and the simulation study seem to be a step forward in addressing some of these issues but in particular in relaxing the assumption of the homogeneity of the economic units that interact in a system.

## 2. Comparison of the ABM and DSGE-3D Model

The use of DSGE models historically has been primarily focused on the analysis of technological changes and their impact on the real economy [30,31] as well as the impact of monetary policy on the business cycle [32,33]. The first DSGE models with financial frictions were not used for impact studies of macroprudential policies on the macro-financial system and social welfare. The research has been mainly focused on the formal explanation of the financial accelerator hypothesis [34,35] and the role of the net worth channel in credit supply [23,36]. In a few models, the impact of LTV changes and capital requirements on the economy was analysed explicitly [25].

After the global financial crisis, interest in systemic risk assessment and macroprudential policies increased [37,38,39]. Currently, one of the most important examples showing the use of DSGE models in research on macroprudential policies is the DSGE model with 3 layers of default (‘3D’ model) [21] developed in ECB. The main goal of the ‘3D’ model was to create a framework for analysing positive and normative macroprudential policies. This model enables to set the optimal levels of capital requirements as well as to analyse welfare within the social planner framework. However, heterogeneity of agents within the model which seems more realistic would change the optimal values for capital requirements and would make the welfare analyses more meaningful.

The paper includes a comment on the ‘3D’ model, due to its similarity to the prepared heterogeneous agent-based simulation. In both cases, the behaviour and decisions of major market players are taken into account. The insolvency of individuals (households), companies and banks describe formally sources of systemic risk. In the ‘3D’ model and in the ABM model, stability of the financial sector is related to the default of agents. Nonetheless, the method of modelling agent’s decisions differs between both approaches. More importantly definition of the insolvency also differs. In the ‘3D’ model, individuals can deposit funds in banks and take loans for the purchase of real estate; entrepreneurs borrow from banks to accumulate capital. Business insolvency is associated with the occurrence of idiosyncratic and aggregate shocks. In the ABM model, the insolvency of agents is tied to internal market dynamics, driven by business and financial factors mainly the supply and investment finance policies of the banks and the demographic factors. The external shocks can be taken into account in the analyses but their significance is smaller than in the DSGE approach.

The ‘3D’ model has been a novelty in the DSGE literature. Traditionally, in the DSGE models, due to appropriately formulated contracts, insolvency at steady state was impossible. Risk of insolvency itself was fully hedged in the model [35]. In a ‘3D’ model, a borrower’s insolvency implies changes in the lender’s balance, which in turn affects his or her optimal behaviour in the market. Moreover, the bank’s insolvency also entails costs to individuals and businesses, in spite of deposit insurance, which in turn strengthens the impact of bank insolvency.

In both approaches, the entire chain of interconnections between agents is formally modelled. Households save and deposit their funds in banks, while other households and companies borrow funds from the same banks. The ABM model departs from the stylised division into two dynasties: savers and borrowers. In an agent-based simulation, each household is made up of individuals with their own heterogeneous profiles in terms of savings, income, spending or additional financing. The way households make decisions depends on the interaction of an individual with family members and other agents in the model. In this way it is possible to trace the way of transferring the risk of insolvency between sectors. However, the transmission of default risk between sectors in DSGE models is accomplished by further optimisation of resources by a representative agent assuming appropriate restrictions and rational expectations. In the ABM model, insolvency transmission is accomplished not only through actual economic and balance-sheet relations and constraints but also the perception of risk and uncertainty of heterogeneous agents in the system [40,41,42]. DelliGatti [41] binds DSGE models with the financial accelerator hypothesis, while ABM models utilise the instability hypothesis of H. Minsky, which take into account not only the financial accelerator hypothesis but also Knightian uncertainty [43]. Additionally, the ABM model presented includes not only the transmission of risk of insolvency between the agents but also between industries.

In the ‘3D’ model the attention is drawn to two types of distortions, which drive banks to excessive use of leverage and significant exposure to risk. They also explain the need for macroprudential policy. The first is related to the existence of deposit insurance agency. Banks run increased risk at the expense of an external agency, leading to higher credit supply and increased demand for deposits. The second distortion is related to the fact that the insolvency is expensive, not only for the lender but also for the borrowers. Occurrence of costly state verification [19] leads to lower demand for credit. The net effect of the two market distortions may be different for each sector. Consequently, the supply of credit for each sector may be lower or greater than the level that maximises the welfare of society.

According to the logic of the ‘3D’ model, in a steady state, when the probability of bank insolvency is high, the risk premium is raised. We assume in the ‘3D’ model that the risk premium is for the whole system and not for a given bank; therefore, banks are willing to take a higher risk because their funding cost is the result of decisions made by all participants within the market. These results seem obvious and a natural conclusion of homogeneity of individuals particularly banks. In the ‘3D’ model, we assume a certain probability of bank default, which is characteristic of the state of equilibrium that we analyse. In financial markets, the decisions of a bank depend on its perception of counterparty risk and estimation of the way a bank is assessed by other units (the >>perception of perception<< of uncertainty). One of the dimensions of the heterogeneity of agents is related to differences in perception of reality, inter alia the perception of counterparty risk, overall uncertainty in the market and the state of the economy. Those elements are clearly omitted in the aforementioned DSGE ‘3D’ model. General frameworks built upon an idea of equilibrium make it impossible to generalise existing DSGE models regarding the aforementioned issues. Also, assumption that a particular individual’s decision may or may not lead to achieving equilibrium seems more realistic. The ABM approach helps overcome these drawbacks. Analyses conducted only within the state of general equilibrium are departed from and the field of interest of non-equilibrium theories and the instability hypothesis is entered [44,45]. The ABM approach is therefore a step towards overcoming the DSGE modelling imperfections indicated by many authors [46,47]. According to empirical literature, the response of systems to the occurrence of shocks may be nonlinear and the financial system itself may be unstable. In addition, shock effects exhibit asymmetric nature. By design, mainly due to the use of log-linearisation, DSGE models are not able to accommodate non-linearity.

DSGE models with financial frictions [19,48], including the ‘3D’ model, refer to the hypothesis of the financial accelerator system. At the same time, the system is characterised by market failures, including the asymmetric information and externalities. A number of researchers highlight the presence of pecuniary externalities [49,50,51]. Pecuniary externalities complement technological externalities and aggregate demand externalities [52]. After the recent financial crisis, one may observe an increased interest in explaining the impact of pecuniary externalities on the system and social welfare. Pecuniary externalities are incorporated mainly into the models by means of credit restrictions. An example of pecuniary externality may be the lack of internalisation of effects of investment decisions in housing and capital prices, which in turn affects the required collateral. In the ‘3D’ model the level of leverage of households and firms is affected endogenously by prices, including real estate prices. At the same time, the direction and size of the impact of pecuniary externalities on allocation of resources are difficult to estimate using the ‘3D’ model.

In agent-based models, it is possible to go one step further to include the premises of instability hypothesis in the analysis. The instability hypothesis is closely tied to financial accelerator hypothesis and it can assume existence of pecuniary externalities. Nonetheless, it goes far beyond it. DelliGatti [41] distinguishes two ways of presenting Minsky’s hypothesis. The first does not refer explicitly to the heterogeneity of agents, and the second assumes the existence of three types of agents; hedging, speculative and Ponzi agents. According to the first explanation, the level of investment in the economy depends on the volume of internal finance and the difference between the market price of assets and the price of the final good. The market price of assets depends on long-term profit expectations. Final good price depends on the expected demand for that good. In the absence of heterogeneity of agents, the representative agent’s investment level decisions are a function of internal financing, which in principle is consistent with the financial accelerator hypothesis. In practice, investment decisions are also made according to how the agents perceive risk. Hence, according to Delli Gatti, in order to fully understand the Minsky’s hypothesis, we need to distinguish between three types of agents that have different attitudes towards external financing.

During the economic boom period, both the borrower and lender expect future cash flows to increase at a pace that will allow the borrower to repay their obligations. As the expectations develop, asset and product prices increase, stimulating investment growth. As a result, production, profits and employment in the economy increase. Banks increase the supply of credit, often requiring lower collateral. Companies are less cautious when they borrow money. Consequently, the proportion of speculative and Ponzi agents increase and the financial system resilience decreases. If the level of debt in relation to its service is perceived as too high on aggregate, the number of insolvency announcements in the system increases, leading to an eventual financial crisis.

Both the ‘3D’ model and the ABM simulation are examples of stochastic dynamic systems that describe the evolution of basic components of the economy. However, while in the ‘3D’ model the economic agents are homogeneous, fully rational and dynamically optimising, in the ABM simulation model, the agents are fully heterogeneous, bounded rational and they perform heuristic optimisation [53,54,55,56].

In order to include the conclusions of the Minsky’s hypothesis in analyses, heterogeneity of agents was included in the agent-based simulation. The heterogeneity of the economy is understood here, however, more broadly than the differentiation of attitudes towards external financing. It is understood as a differentiation of states, behaviour rules and expectations; this implies heterogeneous distributions of variables *ex ante* and *ex post*.

Both groups of researchers, working on the DSGE models and ABM models respectively have recognised the need to consider heterogeneity of the economy in order to analyse the optimality of policies as well as welfare implications. The heterogeneity of agents in the ABM approach is, however, understood differently than in DSGE models with heterogeneous agents [57,58,59]. In the DSGE models with heterogeneous agents, the discontinuation of the assumption of a representative agent is made primarily by allowing for idiosyncratic shocks and by removing the assumption of completeness of asset markets. In particular, such a definition of heterogeneity requires the redefinition of basic model and analyses elements, including the definition of steady state and equilibrium [58]. States of the economy are generally considered to be the realisation of a complex stochastic process with approximate properties to the Markov processes. Therefore, in such models, stationary equilibrium is considered, within which the stationary (ergodic) distribution exists. Within these model types, decision functions and price process realisations are approximated numerically. Some of these techniques were also adopted in the ABM approach [60,61,62].

The heterogeneity of agents in ABM is understood differently. This could be due to differentiation of attributes and states, differentiation of decision rules [63,64], attitudes towards risks or expectations [65,66,67]. In most ABM models with heterogeneous expectations, agents typically have adaptive expectations, as opposed to the rational expectations of representative agent within the DSGE approach. All these dimensions of heterogeneity appear in the simulation presented in the next section. Among others, the empirical distributions of basic economic categories such as income, expenses or credits were used to calibrate the states and the decision functions and procedures were selected after conducting the empirical research on the patterns of consumption and production on the market. The adaptive expectations were imposed as well on simulation design. A key difference between the presented simulation and other agent-based models is inclusion of varied attitudes towards risk and uncertainty in Knight’s sense and the risk sensitivities.

The final distinguishing element of the ABM approach is the possibility to introduce more realistic assumptions in the model than in the DSGE approach. Good examples are the assumption of visibility and satiation. This visibility means that agent decisions take into account not only purely economic conditions and factors, such as the price of the product but also the proximity of the supplier in a spatial sense. In the case of analysis of financial system, the idea of visibility has an additional dimension. It does not come down to visibility in a geographic or spatial sense but rather to the perception of banks as relatively safe institutions. The perception of banks does not have to be reflected in economic foundations or stances [42]. Adopting the saturation principle leads to a departure from the global optimisation of underlying criteria with restrictions on the choice of local maxima.

The presented ABM model is based on the traditions of the EURACE [68,69], FP7 MOSIPS and the population dynamics model in the EU regions models [70] as well as FP7 CRISIS models. Our ABM model is also consistent with the stock-flow approach [71,72]. Impact studies of macroprudential policies on the economy within the ABM approach is relatively new. However, the topic refers to the tradition of agent-based models within financial markets [73,74,75,76,77] as well as literature on credit and financial markets from the agent-based perspective [72,78,79,80,81,82,83,84,85,86,87,88,89,90,91,92,93,94,95,96,97]. In the broader sense, the study also refers to the coevolution models successfully applied in [98,99] to explain the stylized fact of persistency in a time series. For more general reviews on complex network theory refer to References [100,101], while spatial interactions in agent-based modeling were discussed in Reference [102,103].

The purpose of the model is to bridge the gap in the literature on the role of macroprudential policies in systemic risk mitigation. In the following section, details of the ABM model, simulation results and an explanation of the logic behind robustness checks is provided. Comments on welfare analysis within the DSGE and ABM approach are also provided; hence a critical perspective on ‘3D’ modelling results is presented.

## 3. Model Description

Presented within this section is an ABM model suitable for performing simulations that provide detailed insight into the nature of the relationship between the financial system and the real part of the economy. Due to the specifics of agent-based modelling, initially presented is the software environment in which simulations were developed. Next, attributes and activities of agents are presented. Also discussed is the method of sequential updating of the states within simulation modules. The form used to present the model and simulation is consistent with ‘A Common Protocol for Agent-Based Social Simulations’ [104].

### 3.1. The Software

The simulation was developed using object-oriented programming in Java-NetBeans and Eclipse environments. Statistical data to determine the attributes of system agents were grouped in a relational database (PostGreSQL-pgAdmin III). The simulation was linked to the database using Hibernate and SQL queries. According to the logic of object-oriented programming, initially agents and their attributes are described, and then the simulation workings from the perspective of individual agents and their activities are discussed. In the next subsection, a sequential update of agents’ attributes in simulation modules is presented.

### 3.2. Agents and Attributes

In the macro-finance model, 9 agents are distinguished: *Banks*, *Individuals*, *Households*, *Consumers*, *Firms*, *Establishments*, *Suppliers*, *Properties* and *Industries*. All *Parameters* in a separate object are also defined. The relations between agents in the model are presented in the Figure 1. The attributes of the agents can be found in the Table A1, Table A2, Table A3, Table A4, Table A5, Table A6, Table A7, Table A8, Table A9 and Table A10 in the Appendix A.

#### 3.2.1. Individuals, Households, Consumers & Properties

Individuals do not determine the behaviour of the system in the initial *Initialisation*, *Production*, *Supply chain* and *Public contracts* modules. Their significance is enhanced only in the *Households consumption, Households mobility* and *Individuals’ records updating* modules. Nonetheless, individuals play a special role in the model because they determine the functioning of the program and the way data that describe agents’ attributes is mapped in the relational database.

In the *Households consumption* module, the sum of the income of individual entities forming the given household is initially calculated. Total household income is not only equal to the sum of family member income but also includes additional income from rental property and alimony payments. After calculating total household income, resources are divided between consumption and savings. The program counts the number of household consumers and their total disposable income. Depending on the level of disposable income, household savings are determined to update household members’ deposits. If a household consists of a couple (with or without children), then savings are distributed between them. Otherwise, if the household consists of a single adult or single parent, the savings are given to that adult. If the disposable income is exceptionally low, the scheme works similarly, with the common deposits used primarily to cover the basic needs. Decisions on the distribution of income between consumption and savings and the distribution of savings between household members are dynamic. For example, if the household changes as a result of divorce or death of the spouse, the states are updated.

Households represent different types of consumers in the model that determine the purchase of goods and services from a given industry. Households purchase products from suppliers according to the price offered by the supplier relative to the average industry price, quality of goods or services relative to average industry quality and depending on supplier’s spatial location. Households of a certain consumer type seek suppliers sequentially further from their location. Ultimately, they seek suppliers globally, taking into account only the price and quality of the product or service.

Purchases of goods and services do not need to be financed solely by funds deposited in a bank account. Households with creditworthiness are eligible for consumer credit. In the model, loan maturity depends on the amount of the loan.

In the *Households mobility* module, households decide the place of residence and purchase or lease the property. If the household already owns the property, the model verifies whether the cost of repaying the mortgaged loan relative to the household income is too high. If repayment cost is too high, the property is designated for sale. If the property in the previous iteration has already been marked for sale, the price is reduced.

If a household leases a property, rent is equal to the sum of the property owners bank loan repayment obligations; rent is calculated as the ratio of property price to the number of families renting the property. If the rental cost is too high, the household seeks a new rental property. Preference is given to properties near the current place of residence. If household members are not working, further conditions are laid down for disposable income and burdens on household loans. If income after deductions is relatively low and the person is over 25 and not working, then the individual defaults. The banks which have granted the loans to this individual update the non-performing loans value. In this module, income from renting real estate property to other households is also updated. If the property is not leased, it is marked for sale. If it is not purchased, subsequent iterations reduce the desired property sale value.

One of the most important elements of this module is the ability to purchase real estate. Households with high savings buy property in cash with a given probability. Nonetheless, some households, despite their resources, decide to apply for a mortgage loan. Households that do not have sufficient savings, but meet the requirements, also take a loan. If the household is already the owner of one of the properties, it may additionally take a non-residential loan on the pledge of the first property. In practice, either the household takes a residential or non-residential loan, taking into account whether it is already a property owner. Probabilities of taking a residential and non-residential loan were estimated based on empirical data.

Attributes of individual entities change their values in the *Individuals’ records updating* module. Depending on age and sex, the program determines the probability of death of an individual. If the probability is high, the person dies and the assets capital and deposits go to the heir. If the deceased person was the owner of a business, the heir can continue the business or forgo, depending on their previous income as an employee in one of the firms. Simultaneously, the division of capital in the economy changes. If the probability of individual’s death is low, the program directs the individual to the *Education* and *Pairing* modules. In the selected modules the consumer type of households is updated and in the final modules the remaining individuals’ records are updated.

#### 3.2.2. Firms & Establishments

The role of *Firms* is highlighted primarily in the modules: *Firm demography, Mergers* & *Acquisitions* and *Firm growth*. An entrepreneur can open a new business according to a certain probability that depends on the experience of running the company, the age and level of education completed by the individual. When deciding to set up a new business, the entrepreneur takes into account the average profitability of individual industries in the economy, the ease of obtaining licenses for running a business in a given industry and the ability to raise additional funds for opening and running a business. In the case of small and medium-sized enterprises, these funds can be obtained from banks, while in the case of large companies, capital is obtained from many individuals, who are henceforth shareholders of this company. Obtaining funding from a bank requires a number of formal requirements to be met, including leverage, investment and industry risk, investment risk mitigation and a good credit history of the applicant. Implicitly, banks also take into account the cost of labour, equity and size of the enterprise relative to the average enterprise size in the industry.

As a result of the acquisition of another company or as a result of the company’s strategic development, firms can increase the number of establishments they own. In the model, it is possible to obtain additional funds from banks for expansion. Firms can also cease their business activity.

In the simulation, companies announce insolvency when the level of indebtedness and the risk of business exceed the acceptable level. A low percentage of business that has been run in the low-profit sector at high operating costs defaults as well. In the event of a company’s inactivity on the market for six quarters, the program automatically removes the company from the database and program. The adoption of such assumptions ensures adequate market dynamics in the simulation. Firms and establishments are auxiliary objects for the remaining agents in the model. In the final modules, other attributes of firms and establishments are updated.

#### 3.2.3. Establishments & Suppliers

Establishments in the simulation allow for the spatial location of businesses. In the *Initialisation* module, the maximum potential production of goods and services of the establishment is computed. The price of the goods produced by the establishments changes depending on the demand. In each period, the optimal number of products to be stored for future sales is calculated. In the next period, the facility will only produce the number of goods equal to the number of goods demanded minus the number of goods stored. In addition, the production process takes into account the manufacturing risk and the overall level of corporate debt, which should not exceed the level specified by prudential regulation and policies. For the production of final goods, the establishments purchase inputs from others acting as suppliers. The choice of supplier is designed in such a way as to take into account economic categories such as price or product quality but also supplier availability in a geographical or spatial sense.

The demand from the private sector is supplemented by the need for goods from the public sector. It was assumed that the ability to sign a contract in a public sector depends on the size of the establishments producing the given good and the price and quality of the product offered compared to the average values in the industry. The establishments then decide on the destination of the final goods. Establishments may allocate all products for sale in a given sector or export some of their products to another sector and other spatial units.

In addition, the establishments play an auxiliary role in other modules. Individuals seek buildings near the workplace, that is, in the territorial unit within which the establishments are located. At the time of setting up a new company, new establishments are also created. Similarly, when a new company is created as a result of an acquisition or merger, the affiliation of the company and the owner of the establishments change. The owners of establishments decide to increase or decrease the workforce and firms with a strong market position increase the number of establishments. In addition, depending on the demand for work, the number of employees to be hired and fired in each establishment is calculated. In the final modules, firms pay salaries to establishments’ employees and the remaining attributes of establishments are updated.

#### 3.2.4. Industries

The existence of major branches of the economy is assumed. The role of industries is crucial when calculating large exposures (LE) of banks to particular industries. *Industries* is an auxiliary object as well. Firms and establishments operate within the industries. The establishments may import and export goods between industries as part of the purchase of inputs for production. When firms apply for a loan, the total exposure to the industry is checked, which should not exceed the regulatory thresholds. The average values of variables for the industries are treated as a benchmark for business operations of firms and establishments. The main values calculated by the program are the number of units operating in the sector, average product price in the sector, average good quality, average firm size in the sector, average import and export, average industry earnings and average industry workforce. When entrepreneurs decide to run a business, first they attempt it in the relatively most profitable sector, and then in the next sectors.

#### 3.2.5. Banks & Macroprudential Policies

Banks provide loans to individuals, households and companies. In the case of individuals and households, we distinguish between consumer loans and mortgages for residential and non-residential purposes. Companies can apply for a loan to purchase inputs and increase sales and business development (investment loans). Banks analyse loans granted to each of the industries and examine the risks associated with high exposures to the industry. According to regulatory requirements, banks are not allowed to lend to specific industries above certain thresholds.

Individuals and establishments accumulate funds in a bank account. To each individual and establishment at least one bank is matched based on survey data. The model assumes the existence of network and reputation effects. According to the results of empirical research, individuals and households are not guided solely by price in the decision to allocate funds. With relatively high probability, the entity will decide to remain with the bank assigned to them. On the other hand, if they decide to change bank, they will take into account interest rates and additional transaction costs. In the case of consumer credit and the purchase of inputs, agents are more driven by the reputation of the bank than the interest rate. On the other hand, in the case of residential and non-residential loans as well as investment loans, households and firms are primarily guided by the long-term interest rates of banks.

In the module *Banks Supply side & regulatory requirements*, banks set the supply of different types of loans. Banks compete with each other in terms of price, offering different interest rates on deposits and loans, as well as in terms of creditworthiness criteria. Risk-taking banks provide loans to individuals, households or companies with a lower credit history or lower creditworthiness (income or equity respectively). In the absence of macroprudential policy solutions, banks could be willing to lend to more risk-prone entities, which would jeopardise the stability of the financial sector. Thus, the model takes into account the examples of macroprudential policy tools. Firstly, the existence of capital requirements (CAR) and recommendations of the financial supervision authority regarding the capital maintained by banks were taken into account. The model then included the large exposures (LE) and exposures of risk to the industries. Banks cannot lend to a given industry over a specified amount and will not choose to lend to a company that runs risk-prone business without sufficient collateral. By providing housing and non-residential loans, banks pay attention to the following indicators: debt to assets (DTA), debt service to income (DSTI) and loan to value ratios (LTV). Moreover, the liquidity ratio (LCR) and leverage ratio (LR) have been taken into account as well.

From the supervisor’s point of view, it is extremely important to analyse the value of these indicators on aggregate for the economy. The survey data allows only for a static description of the level of these ratios for a given period. The simulation allows for the investigation of changes that occur in the indicator values as a result of dynamic interactions between fully heterogeneous agents. Similarly, for companies and premises leverage requirements (LR) are analysed.

Individuals, households, firms and establishments make decisions about depositing funds and obtaining a loan based on interest rates. In the system it is possible to introduce interest rates offered by banks. In this case, after taking into account the network and reputation effects, banks compete on interest rates. It is possible to take into account counterparty risk indicators and indicators of >>perception of perception<< in the analysis. It is then possible to integrate a macro-financial model into a financial model that simulates the role of risk perception and uncertainty in generating systemic risk in the interbank market [42].

## 4. Sequential Updating of States in the Model

The graphical representation of sequential updating of states in the model is presented in Appendix B.

### Module 0: Initialisation Sector Profitability (M.1)

In the *Initialisation* module, we calculate the average profit (that is ${\bar{\pi }}_{t}^{s}$) of the $N_{t}^{{firm}_{s}}$ firms doing business in the *S* industries ($s,\;\mathrm{where}\;s\in \left\{1,2\ldots S\right\},\;S\in N$) at time *t*. The procedure classifies sectors according to their average profitability. Information on average profitability is used by individuals when they decide to establish a new firm in a given industry. Each firm in the sector *s* generates profits ($\Pi_{t}^{firm_{s}}$).
(1)$$\begin{array}{c} {\bar{\pi }}_{t}^{s}=\overset{N}{\underset{{firm}_{s}=1}{\sum }}\frac{\Pi_{t}^{firm_{s}}}{N_{t}^{{firm}_{s}}}. \end{array}$$
In this module, the model also stores the initial supply of different credit types (for each bank *b*): consumer loans $S_{t}^{{l.C}^{ind}}$, residential $S_{t}^{{l.H}^{ind}}$ and non-residential loans ($S_{t}^{{NH}^{ind}}$), firm investment loans $\left(S_{t}^{l.I^{est}}\right)$ and short-term loans for firms ${(S}_{t}^{l.{SH}^{est}})$ as temporal variables. This information is used later in module (M.55) to determine how much supply was used during this iteration by different agents.

### Module 1: Production Price Updating (M.2)

In the *Price updating* submodule, the price ($P_{t}^{est})$ is updated according to the demand $\left(Q_{t}^{d_{est}}\right)$ for a given good or service in relation to the expected demand ($E\left(Q_{t}^{d_{est}}\right)$) and the level of production $\left(Y_{t}^{est}\right)$ relative to the maximum potential production of each establishment (${Y_{t}}_{max}^{s})$. In this sub-module, the values of variables determining the number of employees to be hired $\left(L_{t}^{{HI}^{est}}\right)$ and fired $\left(L_{t}^{{FI}^{est}}\right)$ in the current period in each establishment are set to zero.

The maximum production of premises in a given sector is then calculated according to the Cobb and Douglas production function:(2)$$\begin{array}{c} {Y_{t}}_{max}^{s}=\;\alpha_{1.s}{\left(L_{t}^{est}+1\right)}^{\alpha_{2.s}}{\left(K_{t}^{est}\right)}^{\alpha_{3.s}}{\left(\frac{{\left(Ql\right)}_{t}^{est}}{{\left(Ql\right)}_{t}^{s}}\right)}^{\alpha_{4.s}}{\left(A_{t}^{est}\right)}^{\alpha_{5.s}}, \end{array}$$
where $L_{t}^{est}$ is the labour force, $K_{t}^{est}$ is the capital, $A_{t}^{est}$ is technology and ${\left(Ql\right)}_{t}^{est}/{\left(Ql\right)}_{t}^{s}$ represents the relative quality of the establishment’s product (or service) with respect to the average quality of product and services in the sector (industry). The value of parameters $\alpha_{1.s}$$\alpha_{2.s}$$\alpha_{3.s}$$\alpha_{4.s}$$\alpha_{5.s}$ are specific to each industry. When initialising the system, the price is defined based on the initial conditions in the database. In subsequent iterations, the price of the previous period is assumed to be the initial value of the good ($P_{t}^{est}=P_{t-1}^{est}$). This price may change depending on the demand relative to expected demand and production in comparison to the maximum capacity, that is to say the maximum potential production. If the demand for good produced by a given facility is greater than the expected demand for that good and production is greater than the specified part $\alpha_{6}\;$ of maximum production, then the price increases in proportion to the given parameter $\;\alpha_{7.s}$. The $\;\alpha_{7.s}$. parameter is industry-specific. If the demand for a good is less than the part ${\;\alpha \;}_{8}\;$ of expected demand for this good and output is less than the specified $\alpha_{9}$ part of maximum output, the price drops by the percentage given by the parameter $\alpha_{10.s}$. The $\alpha_{10.s}\;$ parameter is industry-specific. This procedure is consistent with the adoption of adaptive expectations in the model.

### Module 1: Production Expected Demand Updating (M.3)

In this sub-module, we update the expected demand for the next iteration ($E(Q_{t}^{d_{est}})$). The formula of the expected demand for a good depends on the production experience of the establishment. If the premises have been operating on the market for at least a quarter, the expected demand for its good is calculated according to the following formula:(3)$$\begin{array}{cc} E\left(Q_{t}^{d_{est}}\right) & =\left[\alpha_{11.s}\left(\frac{P_{t}^{est}}{P_{t}^{s}}\right)+\alpha_{12.s}\left(\frac{P_{t-1}^{est}}{P_{t-1}^{s}}\right)+\alpha_{13.s}\left(\frac{{\left(Ql\right)}_{t}^{est}}{{\left(Ql\right)}_{t}^{s}}\right)+\alpha_{14.s}\left(\frac{{\left(Ql\right)}_{t-1}^{est}}{({Ql)}_{t-1}^{s}}\right)\right]\times Q_{t-1}^{d_{est}}. \end{array}$$
The expected demand depends on the price of the product relative to the average price in the industry within the periods *t* and t−1, product quality relative to the average quality in the industry within the periods *t* and t−1, given parameter values, and demand for the goods up to date. The parameters $\alpha_{11.s}$, $\alpha_{12.s}$$\alpha_{13.s}\;$ and $\alpha_{14.s\;}$ are industry-specific.

However, if the establishment is new, then the expected demand is calculated to take into account the workforce in the newly-created establishment and the average sales per worker in the industry ${\left(Sl\right)}_{t}^{s}$ within which it operates.
(4)$$\begin{array}{cc} E\left(Q_{t}^{d_{est}}\right) & =\left[\alpha {\;}_{11.s}\left(\frac{P_{t}^{est}}{P_{t}^{s}}\right)+\alpha_{12.s}\left(\frac{P_{t-1}^{est}}{P_{t-1}^{s}}\right)+\alpha_{13.s}\left(\frac{{\left(Ql\right)}_{t}^{est}}{({Ql)}_{t}^{s}}\right)+\alpha_{14.s}\left(\frac{{\left(Ql\right)}_{t-1}^{est}}{{\left(Ql\right)}_{t-1}^{s}}\right)\right]\times \left(L_{t}^{est}+1\right)\times {\left(Sl\right)}_{t}^{s}. \end{array}$$

### Module 1: Production Expected Stock Updating (M.4)

If the establishment is new (${\left(New\right)}_{t}^{est}=1$), we calculate the optimal level of stock (${Inv}_{{opt}_{t}}^{est})$ as part of the expected demand, specified by the parameter $\alpha_{15}$. If the establishment is already established, the optimum stock level is calculated to take into account the expected demand for the good, the ratio of expected revenue from the sale of goods ($P_{t}^{est}\times E\left(Q_{t}^{d_{est}}\right)$) to the costs of producing that good in the current period (${TC}_{t}^{est}$), and the ratio of sales revenue in the current period ${(P}_{t}^{est}\times {\left(Sl\right)}_{t}^{est})$ to the total costs incurred in the previous period (${TC}_{t-1}^{est}$).
(5)$$\begin{array}{c} {Inv}_{{opt}_{t}}^{est}=\alpha_{15}\times \;E\left(Q_{t}^{d_{est}}\right)+\;\alpha_{16}\times \left(\frac{P_{t}^{est}\times E\left(Q_{t}^{d_{est}}\right)}{{TC}_{t}^{est}}\right)+\alpha_{17}\times \left(\frac{P_{t}^{est}\times {\left(Sl\right)}_{t}^{est}}{{TC}_{t-1}^{est}}\right). \end{array}$$

### Module 1: Production Production Decision Making (M.5)

If the optimum stock of products ${(Inv}_{{opt}_{t}}^{est})$ is less than or equal to the actual stock ${(Inv}_{t}^{est})$, the establishment will not produce goods in the current period. However, if the optimal stock is greater than the stored number of goods, the establishment should produce the difference between the optimum level and the current stock.
(6)$$\begin{array}{c} Y_{t}^{est}=\min\left({Y_{t}}_{max}^{est};\;{Inv}_{{opt}_{t}}^{est}-{Inv}_{t}^{est}\right). \end{array}$$
Nonetheless, these establishments will produce goods only when the level of leverage and financial risk associated with the debt of the establishments does not exceed the levels specified by the parameters $\alpha_{18}$ and $\alpha_{19}$. If the establishment meets the conditions to produce goods, the production is equal to the lower value of either maximum production of the establishment or the difference between the optimum stock and the actual stock (inventory) of goods.

### Module 2: Supply Chain Quantity of Inputs, Import & Export (M.6)

In this module, establishments buy inputs and decide on the import and export of goods between industries. In order to minimise costs, they choose a supplier from the nearest spatially located area, thus limiting the cost of transport. In addition, the adopted mechanism allows the modelling of continuation of transaction relationships between suppliers and recipients of goods. Each company is located spatially in the form of establishments. Each establishment is a supplier for another establishment. For each establishment in all sectors, the initial value of inputs ${\left((Inp\right)}_{t}^{est})$, profits from sales $({\left(Sl\right)}_{t}^{est}$) and demand for goods ${(Q}_{t}^{d_{est}})$ are set to zero. Next, the amount of inputs ${(q}_{t}^{est.buy.(sup)(.s)})$ (provided by suppliers from sectors s) necessary to ensure continuity of production, taking into account import and export of inputs between industries is calculated. If the facility imports or exports semi-finished products, the amount of inputs that the establishment is going to purchase is obtained using the following formula:(7)$$\begin{array}{c} q_{t}^{est.buy.(sup)(.s)}=\alpha_{20.s-buy.s}\times Y_{t}^{est}\times \alpha_{21.s-buy.s}. \end{array}$$
In the model there are 2×s values for parameters $\alpha_{20.s-buy.s}$ and $\alpha_{21.s-buy.s}$ (*Cf.* Calibration for the explanation how the values of parameters were obtained). For each establishment in the industry, the value of the parameter is the same but the values vary between sectors (industries). If the establishment does not import goods then the quantity of purchased goods is equal to the part of production specified by parameter $\alpha_{20.s-buy.s}$:(8)$$\begin{array}{c} q_{t}^{est.buy.(sup)(.s)}=\alpha_{20.s-buy.s}\times Y_{t}^{est}. \end{array}$$
The total quantity of inputs is the sum of inputs purchased from all suppliers in all sectors ($q_{t}^{est.buy}$).

### Module 2: Supply Chain Supplier Selection (M.7)

When searching for a supplier, the establishment takes into account the amount of goods stored by the supplier $({Inv}_{t}^{est(.\sup)}>$
$q_{t}^{est.buy(.s)}$), and compares the ratio of quality to price of a supplier (establishment in the sector *s*) with the average ratio within the industry (sector):(9)$$\begin{array}{c} \frac{\alpha_{22.s-est}\times {\left(Ql\right)}_{t}^{est(.s)}}{\alpha_{23.s-est}\times P_{t}^{est(.s)}}>\frac{\alpha_{24.s-est}\times {\left(Ql\right)}_{t}^{s}}{\alpha_{25.s-est}\times P_{t}^{s}}. \end{array}$$
In addition, it also takes into account supplier location (i.e., compares the spatial codes at NUTS 1-4 levels: ${\vartheta \;}_{t}^{{est}^{1}},\;{\vartheta \;}_{t}^{{est}^{2}},\;{\vartheta \;}_{t}^{{est}^{3}},\;{\vartheta \;}_{t}^{{est}^{4}}$). If the current supplier has a sufficient number of inputs for sale, and the quality and price of the good are acceptable in relation to the average price and quality in the sector, then the establishment can buy inputs from the supplier. The model consolidates the network effects developed during the cooperation of businesses. If the supplier does not meet the requirements, the establishments seek a new supplier locally in increasingly distant locations and then globally. The parameter values from $\alpha_{22.s-est}$ at time t to $\alpha_{25.s-est}\;$ are specific to the supplier’s sector.

### Module 2: Supply Chain Inputs Purchase (M.8)

After selecting a supplier, the establishment purchases inputs. To purchase inputs, the establishment must have sufficient liquid assets to cover the wages and the cost of buying the inputs: ${\left(LA\right)}_{t}^{est.buy}-W_{t}^{est.buy}\geq \left(1+\;\alpha_{26.s-sup}\times \varpi \right)\times P_{t}^{est(.sup)(.s)}\times q_{t}^{est.buy(.s)}$, where ϖ is the binary variable expressing whether the cost of transportation should be added. If it has sufficient liquid assets, it can finance the purchase of inputs from accumulated funds. Therefore, in the model, with the probability $pr_{1}$ the establishment will not apply for a loan. In that case, for the establishment-buyer inputs $\left({\left(Inp\right)}_{t}^{est.buy}\right)$ and liquid assets $\left({\left(LA\right)}_{t}^{est.buy}\right)$ are updated to. The signs “+=” shall be interpreted as the incrementation of the value of the variable by the amount quantified by the formula given on the right-hand side. Respectively, “-=”, shall be interpreted as a decrease in the value.
(10)$$\begin{array}{c} {\left(Inp\right)}_{t}^{est.buy}+=\left(1+\;\alpha_{26.s-sup}\times \varpi \right)\times \;P_{t}^{est(.sup)(.s)}\times q_{t}^{est.buy(.s)} \end{array}$$
(11)$$\begin{array}{c} {\left(LA\right)}_{t}^{est.buy}-=\left(1+\;\alpha_{26.s-sup}\times \varpi \right)\times \;P_{t}^{est(.sup)(.s)}\times q_{t}^{est.buy(.s)}. \end{array}$$
While for all establishments-suppliers from each sector, the sales expressed in monetary terms $\left({\left(Sl\right)}_{t}^{est.(sup)(.s)}\right)$, demand for goods $\left(Q_{t}^{d_{est(.sup).(s)}}\right)$, liquid assets $\left({\left(LA\right)}_{t}^{est.(sup)(.s)}\right)$ and stock $\left({\left(Inv\right)}_{t}^{est.(sup)(.s)}\right)$ are updated.
(12)$$\begin{array}{c} {\left(Sl\right)}_{t}^{est.(sup)(.s)}+=P_{t}^{est(.sup).(s)}\times q_{t}^{est.buy(.s)} \end{array}$$
(13)$$\begin{array}{c} {\left(LA\right)}_{t}^{est.(sup)(.s)}+=P_{t}^{est(.sup).(s)}\times q_{t}^{est.buy(.s)} \end{array}$$
(14)$$\begin{array}{c} Q_{t}^{d_{est.\left(sup\right).\left(s\right)}}+=q_{t}^{est.buy(.s)} \end{array}$$
(15)$$\begin{array}{c} {\left(Inv\right)}_{t}^{est.(sup)(.s)}-=q_{t}^{est.buy(.s)}. \end{array}$$
In particular cases, with the probability of ${1-\;pr}_{1}$, despite sufficient liquid assets, the establishment may apply for a loan to purchase additional inputs that will allow the facility to increase its production capacity and sales. If the establishment applies for a loan, the applicant’s creditworthiness is checked even if its accumulated funds are sufficient to cover the purchase. In the submodule *Bank credit admissibility 1* (M.9), conditions in addition to liquidity funds are checked. In accordance to the market dynamics of short-term loans, some of the applicants will not obtain a loan from the bank due to lack of creditworthiness. The possibility of establishments applying for a short-term loan in the case of temporary liquidity problems has also been included in the model (i.e., ${\left(LA\right)}_{t}^{est.buy}<{(1+\;\alpha_{26.s-sup}\times \varpi )\times P}_{t}^{est(.\sup)(.s)}\times q_{t}^{est.buy(.s)}$). If, in spite of short-term liquidity problems, the establishment has not completely lost its creditworthiness, the bank may grant him credit for the purchase of inputs in the submodule *Bank credit admissibility 2* (M.10). If the establishment has no creditworthiness, it has to *adjust* the quantity to buy $\left({\tilde{q}}_{t}^{est.buy}\right)$:(16)$$\begin{array}{c} {\tilde{q}}_{t}^{est.buy(.s)}\;=round\left(\frac{{\left((LA\right)}_{t}^{est.buy}-\;W_{t}^{est.buy})\times \alpha_{20.s-buy.s}}{P_{t}^{est.(sup)(.s)}\times \left(1+\alpha_{26.s-sup}\times \varpi \right)}\right). \end{array}$$
After the purchase, the value of inputs (${\left(Inp\right)}_{t}^{est})$ and liquid assets of establishment-buyer (${\left(LA\right)}_{t}^{est.buy})$ are updated.
(17)$$\begin{array}{c} {\left(Inp\right)}_{t}^{est.buy}+={\tilde{q}}_{t}^{est.buy(.s)}\times P_{t}^{est.sup(.s)}\times \left(1+\alpha_{26.s-sup}\times \varpi \right) \end{array}$$
(18)$$\begin{array}{c} {\left(LA\right)}_{t}^{est.buy}-={\tilde{q}}_{t}^{est.buy(.s)}\times P_{t}^{est.sup(.s)}\times \left(1+\alpha_{26.s-sup}\times \varpi \right) \end{array}$$
At the same time, we update the values of sales (${\left(Sl\right)}_{t}^{est.(sup)}$), demand for a good $\left(Q_{t}^{d_{est.(sup)}}\right)$, liquid assets (${\left(LA\right)}_{t}^{est.buy}$) and stock of suppliers from all sectors that provided inputs to establishments (${Inv}_{t}^{est.sup}$) are updated:(19)$$\begin{array}{c} {\left(Sl\right)}_{t}^{est1.(sup)}+=P_{t}^{est.sup(.s)}\times {Inv}_{t}^{est1.sup} \end{array}$$
(20)$$\begin{array}{c} {\left(Sl\right)}_{t}^{est2.(sup)}+=P_{t}^{est.sup(.s)}\;\times \left({\tilde{q}}_{t}^{est.buy(.s)}\;-\;{Inv}_{t}^{est1.sup}\right) \end{array}$$
(21)$$\begin{array}{c} Q_{t}^{d_{est1.(sup)}}+\;={Inv}_{t}^{est1.sup} \end{array}$$
(22)$$\begin{array}{c} Q_{t}^{d_{est2.(sup)}}+\;={\tilde{q}}_{t}^{est.buy(.s)}-{Inv}_{t}^{est1.sup} \end{array}$$
(23)$$\begin{array}{c} {\left(LA\right)}_{t}^{est1.buy}+=P_{t}^{est.sup.(s)}\times {Inv}_{t}^{est1.sup} \end{array}$$
(24)$$\begin{array}{c} {\left(LA\right)}_{t}^{est2.(sup)}+=P_{t}^{est.sup.(s)}\times \left({\tilde{q}}_{t}^{est.buy(.s)}-\;{Inv}_{t}^{est1.sup}\right) \end{array}$$
(25)$$\begin{array}{c} {Inv}_{t}^{est2.sup}-=\left({\tilde{q}}_{t}^{est.buy(.s)}-\;{Inv}_{t}^{est1.sup}\right) \end{array}$$
(26)$$\begin{array}{c} {Inv}_{t}^{est1.sup}=0 \end{array}$$
where ${\tilde{q}}_{t}^{est.buy(.s)}$ is the quantity of inputs that has been selected according to the adaptive algorithm.

### Module 2: Supply Chain Short Term Credit Admissibility 1 (M.9)

In this submodule we analyse the case of an establishment without liquidity problems. The requested amount is given by the formula:(27)$$\begin{array}{c} l_{t}^{{SH}^{est}}+=\alpha_{27}\times {\left((LA\right)}_{t}^{est.buy}-W_{t}^{est.buy}-P_{t}^{est\left(.sup\right)\left(.s\right)}\times q_{t}^{est.buy\left(.s\right)}) \end{array}$$
In the future, the model could also recognize different business types, similarly to the consumer types in the model, however at this stage, access to such disaggregated data was unavailable. Firstly, it is checked whether the matched bank in the database is able to loan this quantity ($S_{t}^{l.{SH}^{est}}\geq l_{t}^{{SH}^{est}})$, as is the creditworthiness of the applicant. The values of ROA, ROE and leverage ratios as well as the value of average financial risk associated with the establishment operating in a given sector and its default history are checked. If the loan is granted, then the values of loans $\left(l_{t}^{{SH}^{est}}\right)$, quarterly payments ($l_{t}^{SH_{q}^{est}}$), interest to be paid (in total) ($R_{t}^{{SH}^{ind}})$ and quarterly $\left(R_{t}^{{SH}_{q}^{ind}}\right)$, inputs (${\left(Inp\right)}_{t}^{est})$ and liquidity assets $\left({\left(LA\right)}_{t}^{est}\right)$ are updated (for the establishment-buyer).
(28)$$\begin{array}{ll} R_{t}^{{SH}^{ind}} & +=\frac{1}{\kappa \;}\times \left[\alpha_{27}\times {\left((LA\right)}_{t}^{est.buy}-W_{t}^{est.buy}-P_{t}^{est\left(.sup\right)\left(.s\right)}\times q_{t}^{est.buy\left(.s\right)}\right)\times {\left(1+0.25\times i_{l^{SH}}\right)}^{M^{{SH}^{est}}}+\\ & -\alpha_{27}\times {\left((LA\right)}_{t}^{est.buy}-W_{t}^{est.buy}-P_{t}^{est\left(.sup\right)\left(.s\right)}\times q_{t}^{est.buy\left(.s\right)}))] \end{array}$$
(29)$$\begin{array}{c} R_{t}^{{SH}_{q}^{est}}=\frac{R_{t}^{{SH}^{est}}}{M^{{SH}^{est}}} \end{array}$$
(30)$$\begin{array}{c} l_{t}^{SH_{q}^{est}}=\frac{l_{t}^{{SH}^{est}}}{M^{{SH}^{est}}} \end{array}$$
(31)$$\begin{array}{c} {\left(Inp\right)}_{t}^{est.buy}+=P_{t}^{est\left(.sup\right)\left(.s\right)}\times q_{t}^{est.buy\left(.s\right)} \end{array}$$
(32)$$\begin{array}{c} {\left(LA\right)}_{t}^{est.buy}-=P_{t}^{est\left(.sup\right)\left(.s\right)}\times q_{t}^{est.buy\left(.s\right)}-\alpha_{27}\times {\left((LA\right)}_{t}^{est.buy}-W_{t}^{est.buy}-P_{t}^{est\left(.sup\right)\left(.s\right)}\times q_{t}^{est.buy\left(.s\right)}) \end{array}$$
At the same time, the revenue of banks (${\left({Rev}_{l^{SH}}\right)}_{t}^{b})$ and supply of short-term credit for firms ($S_{t}^{l.{SH}^{est}})$ are updates as well as sales $\left({\left(Sl\right)}_{t}^{est}\right)$, demand $\left(Q_{t}^{d_{est}}\right)$, liquidity assets $\left({\left(LA\right)}_{t}^{est}\right)$ and stock of all establishments from sectors $({\left(Inv\right)}_{t}^{est})\;$ that provided inputs to establishments (buyers).
(33)$$\begin{array}{c} {\left({Rev}_{l^{SH}}\right)}_{t}^{b}+=R_{t}^{{SH}_{q}^{est}} \end{array}$$
(34)$$\begin{array}{c} S_{t}^{l.{SH}^{est}}-=\alpha_{27}\times {\left((LA\right)}_{t}^{est.buy}-W_{t}^{est.buy}-P_{t}^{est\left(.sup\right)\left(.s\right)}\times q_{t}^{est.buy\left(.s\right)}) \end{array}$$
(35)$$\begin{array}{c} {\left(Sl\right)}_{t}^{est(.sup)}+=P_{t}^{est\left(.sup\right)\left(.s\right)}\times q_{t}^{est.buy\left(.s\right)} \end{array}$$
(36)$$\begin{array}{c} Q_{t}^{d_{est}}+=q_{t}^{est.buy\left(.s\right)} \end{array}$$
(37)$$\begin{array}{c} {\left(LA\right)}_{t}^{est(.sup)}+=P_{t}^{est\left(.sup\right)\left(.s\right)}\times q_{t}^{est.buy\left(.s\right)} \end{array}$$
(38)$$\begin{array}{c} {\left(Inv\right)}_{t}^{est(.sup)}-=q_{t}^{est.buy\left(.s\right)}. \end{array}$$
Short-term loans make it possible to guarantee the solvency of establishments in everyday business transactions. Restrictions to funding provision could result in an establishment’s loss of liquidity and production capacity. If the matched bank does not agree to grant credit, the same conditions are checked with other banks in the market. Firstly, the conditions are checked in the bank that offers the lowest interest rate. If none will grant the loan, the establishment needs to adjust the quantity of inputs to be purchased $\left({\tilde{q}}_{t}^{est.buy}\right)$.

### Module 2: Supply Chain Short Term Credit Admissibility 2 (M.10)

The requested amount is given by the formula:(39)$$\begin{array}{c} l_{t}^{{SH}^{est}}=\;q_{t}^{est.buy\left(.s\right)}\times P_{t}^{est(.sup)(.s)}\times \left(1+\alpha_{26.s-sup}\times \varpi \right)-\left({\left(LA\right)}_{t}^{est.buy}-W_{t}^{est.buy}\right). \end{array}$$
Similar to the submodule 9, the supply conditions and creditworthiness are checked. In this case, the conditions for granting credit are also tightened, hence the differences in the parameters in the sub-modules *Bank credit admissibility 1* (M.9) and *Bank credit admissibility 2* (M.10). If a loan is not granted by a given bank, the establishment tries to obtain a loan from another bank. If there is no bank that is willing to supply a loan, the establishment is only able to purchase a portion of the planned amount of inputs. The logic of adaptive algorithms is used here. The values of variables are updated in the similar way as in the previous module (M.9). Purchases by establishments from the suppliers are supplemented by the purchases of consumers and the governmental sector.

### Module 3: Household consumption - Individuals & households income (M.11) & (M.12)

Firstly, in the module, the individuals’ and households’ incomes (respectively $y_{t}^{ind}\;\mathrm{and}\;y_{t}^{HH})$ are computed. Individual income includes income from various sources: wages, business activity, dividends, public pensions, pension benefits, pre-retirement benefits and training allowances. The model distinguishes three main categories: wage ($w_{t}^{ind}$), subsidy (${\left(sub\right)}_{t}^{ind}$) and interest from bank savings (deposits) ($i_{d}\times \;d_{t}^{ind}$). An individual’s income is expressed by the following formula:(40)$$\begin{array}{c} y_{t}^{ind}=w_{t}^{ind}+{\left(sub\right)}_{t}^{ind}+\;i_{d}\times \;d_{t}^{ind}. \end{array}$$
Household income includes the sum of individual incomes, supplemental security income, alimony, donations, property rental income, interest and dividends from savings accounts, bonds, investment funds and income earned from participation in companies in which family members were investors or inactive partners. All items have been grouped into categories at the database level. The total income of the household (composed of $N^{ind.HH}$ individuals) is given by:(41)$$\begin{array}{c} y_{t}^{HH}=\overset{N^{ind.HH}}{\underset{i=1}{\sum }}y_{it}^{ind}+{\left(Don\right)}_{t}^{HH}+{\left(Rent\right)}_{{RE}_{t}}^{HH}, \end{array}$$
where ${\left(Don\right)}_{t}^{HH}$ are donations and ${\left(Rent\right)}_{{RE}_{t}}^{HH}$ is an additional income from renting apartments.

At the beginning of the cycle, at least two banks are assigned to each individual; the bank to which the individual entrusted their savings (${Id}_{d_{t}}^{bank.(ind)}$) and the bank that may grant consumer loans to the individual (${Id}_{l_{t}^{C}}^{bank.\left(ind\right)}$). In the model, it is assumed that individuals are less prone to change the bank to which they entrusted their saving and current account funds than to change the ‘bank-lender.’ If they decide to change bank, when looking for a new bank, they take into account the offered interest rates on deposits and transaction costs, that is, the costs of changing the bank and opening a new bank account. The likelihood of a bank changing in the case of a consumer loan is higher than in the case of deposits. In the case of deposits, psychological factors such as habit formation or the perception of a bank as a reputable institution, which reinforce network effects, play a greater role. In the case of mortgages, households primarily rely on the interest rate on the loan.

### Module 3: Households Consumption Net Savings (M.13)

After calculating household income, savings ${(s}_{t}^{HH})$ are calculated. Disposable income is the household income after deducting the cost of living ($h_{t}^{HH}$), which includes the cost of renting or repaying the mortgage. If the disposable income is higher than the specified income per person in the household, where the per person income is given by the parameter $\alpha_{41}$, then the savings are computed according to the following formula:(42)$$\begin{array}{c} s_{t}^{HH}=max\left\{0;\;\frac{\sqrt{{\left(Age\right)}_{t}^{eldest.ind}}}{\alpha_{42}}\times \left(y_{t}^{HH}-h_{t}^{HH}-\alpha_{41}\times N_{t}^{ind.HH}\right)\right\}, \end{array}$$
where $\alpha_{41}\;$ is the minimum cost of food per person in the household according to Central Statistical Office statistics, $N_{t}^{ind.HH}\;$ counts the number of individuals in the household.

However, if disposable income is lower than the specified parameter $\alpha_{38}$ (income per capita) multiplied by the number of consumers in the household ($N_{t}^{ind.HH}$), two possibilities are considered. First, when the disposable income is positive, the savings are calculated according to the following formula:(43)$$\begin{array}{c} s_{t}^{HH}=max\left\{0;\;\frac{\sqrt{{\left(Age\right)}_{t}^{eldest.ind}}}{3\times \alpha_{42}}\times \left(y_{t}^{HH}-h_{t}^{HH}-\alpha_{41}\times N_{t}^{ind.HH}\right)\right\}. \end{array}$$
Household savings are redistributed between the accounts of adult household members. If the household is a single person, then all savings are transferred to his or her account. If the household is a couple or an extended one (more than two adults in the family), then the corresponding parts of the savings are transferred to the adults’ bank accounts.
(44)$$\begin{array}{c} d_{t}^{HH}+=\frac{1}{N_{t}^{ind.HH}}\times s_{t}^{HH}. \end{array}$$
Second, if the disposable income is negative, household members spend their savings and current deposits on consumption and try to sell the properties on the real estate market. The algorithm looks for households which have sufficient funds after deducting the debt burden to pay for the property:(45)$$\begin{array}{c} \overset{N^{ind}}{\underset{i=1}{\sum }}d_{it}^{ind}-\overset{N^{ind}}{\underset{i=1}{\sum }}\left(l_{it}^{C_{q}^{ind}}+R_{it}^{C_{q}^{ind}}+l_{it}^{H_{q}^{ind}}+R_{it}^{H_{q}^{ind}}+l_{it}^{NH_{q}^{ind}}+R_{it}^{{NH}_{q}^{ind}}\right)\;\geq P_{t}^{prop}. \end{array}$$
If this kind of household is found, then the values of deposits of buyers and sellers are updated as well as the status of the ownership of buyers $\left({\left(Own\right)}_{t}^{HH}=1\right)$.
(46)$$\begin{array}{c} d_{it}^{ind.seller}+=\frac{1}{N^{ind}}P_{t}^{prop} \end{array}$$
(47)$$\begin{array}{c} d_{it}^{ind.buy}-=\frac{1}{N^{ind}}P_{t}^{prop}. \end{array}$$
If the household does not receive additional resources, this can eventually lead to default ${(PD}_{t}^{ind}\;+\;+\;)$ and the household is removed from the database. The program updates the value of non-performing loans (${\left({NPL}_{l^{I}}\right)}_{t}^{b},\;{\left({NPL}_{l^{C}}\right)}_{t}^{b},\;{\left({NPL}_{l^{H}}\right)}_{t}^{b}\;,\;{\left({NPL}_{l^{NH}}\right)}_{t}^{b}$). The probability of default of the bank increases.
(48)$$\begin{array}{c} {\left({NPL}_{l^{I}}\right)}_{t}^{b}+={\left(pl\right)}_{t}^{I^{ind}} \end{array}$$
(49)$$\begin{array}{c} {\left({NPL}_{l^{C}}\right)}_{t}^{b}+={\left(pl\right)}_{t}^{C^{ind}} \end{array}$$
(50)$$\begin{array}{c} {\left({NPL}_{l^{H}}\right)}_{t}^{b}+={\left(pl\right)}_{t}^{H^{ind}} \end{array}$$
(51)$$\begin{array}{c} {\left({NPL}_{l^{NH}}\right)}_{t}^{b}+={\left(pl\right)}_{t}^{{NH}^{ind}}, \end{array}$$
where ${\left(pl\right)}_{t}^{I^{ind}}$ is the sum of liabilities to the bank for outstanding investment loans that have to be repaid in the given iteration, ${\left(pl\right)}_{t}^{C^{ind}}$ is the sum of liabilities to the bank for outstanding consumer loans, ${\left(pl\right)}_{t}^{H^{ind}}$ is the sum of liabilities to the bank for outstanding housing loans and ${\left(pl\right)}_{t}^{NH^{ind}}$ is the sum of liabilities to the bank for outstanding non-housing loans.

The bank may become a new temporary owner of the property if the household was removed from the database after the default. The property is marked for sale. After selling, we update the value of bank’s revenues:(52)$$\begin{array}{c} {\left({Rev}_{l^{H}}\right)}_{t}^{b}+=P_{t}^{prop}. \end{array}$$

### Module 3: Households Consumption Consumer Loans Update (M.14)

In this submodule, the desired consumption of goods in the current period in a particular industry by the household is determined. Households can finance their consumption entirely by their own means or apply for consumer credit. The basic amount of good purchased from the industry $\left(Q_{basic}^{HH.(cons)}\right)$ is given by:(53)$$\begin{array}{c} Q_{basic}^{HH.(cons)}=\;\alpha_{43.tc.s}\times \frac{\sum_{i=1}^{N^{ind.HH}}d_{it}^{ind}-\sum_{i=1}^{N^{ind.HH}}\left(l_{it}^{C_{q}^{ind}}+R_{it}^{C_{q}^{ind}}+l_{it}^{H_{q}^{ind}}+R_{it}^{H_{q}^{ind}}+l_{it}^{NH_{q}^{ind}}+R_{it}^{{NH}_{q}^{ind}}\right)}{P_{t}^{s}\times \left(1+t_{VAT}^{s}\right)}, \end{array}$$
where parameter value $\alpha_{43.tc.s}$ is specific to industry and customer type. The parameter expresses the percentage of total consumption that is, household purchases from all industries. When buying from several industries, we assume $\sum_{s=1}^{S}\alpha_{43.tc.s}=1,\;$ where *S* is the number of industries. If the household consume only the quantity $Q_{t}^{HH.(cons)}=Q^{HH.{\left(cons\right)}_{basic}}$ we proceed to the *Supplier searching module*.

In the case of taking a loan, consumption funds are increased by the amount of the loan less debt burden and service. The loan can only be granted if the following basic condition is met:(54)$$\begin{array}{c} \left\{y_{t}^{HH}-h_{t}^{HH}-\overset{N^{ind.HH}}{\underset{i=1}{\sum }}\left(l_{it}^{C_{q}^{ind}}+R_{it}^{C_{q}^{ind}}+l_{it}^{H_{q}^{ind}}+R_{it}^{H_{q}^{ind}}+l_{it}^{NH_{q}^{ind}}+R_{it}^{{NH}_{q}^{ind}}\right)\right\}\geq \alpha_{44}\times N_{t}^{ind.HH}. \end{array}$$
In this case the quantity of loans is given by the formula:(55)$$\begin{array}{c} l_{temp}^{C^{ind}}=\alpha_{45.tc.s}\times \left\{y_{t}^{HH}-h_{t}^{HH}-\overset{N^{ind.HH}}{\underset{i=1}{\sum }}\left(l_{it}^{C_{q}^{ind}}+R_{it}^{C_{q}^{ind}}+l_{it}^{H_{q}^{ind}}+R_{it}^{H_{q}^{ind}}+l_{it}^{NH_{q}^{ind}}+R_{it}^{{NH}_{q}^{ind}}\right)\right\}+\alpha_{46.s}. \end{array}$$
The parameter value $\alpha_{45.tc.s}$ is specific to industry and customer type, while $\alpha_{46.s}$ is industry-specific. With a certain probability (${pr}_{2}$), an individual tries to obtain the quantity in the matched bank in the database (${Id}_{l_{t}^{C}}^{bank.\left(ind\right)}$). In this case, the supply from the bank and creditworthiness are checked in the *Supply side checking 1* (M.15) and *Consumer credit admissibility 1* (M.17). The individual can also try to obtain the loan from other banks. In such cases, interest rates are compared, and *Supply side checking 2* (M.16) and *Consumer credit admissibility 2* (M.18) are proceeded to.

### Module 3: Households Consumption Supply Side Checking 1 & 2 (M.15) & (M.16)

In the *Supply side checking 1* (M.15) submodule it is checked whether the bank assigned to the household has sufficient funds to grant the loan ($S_{t}^{{l.C}^{ind}}\geq l_{t}^{C^{ind}}$), whether the bank’s policy will allow another loan to be granted, and whether the regulatory requirement for sectoral exposures is met ($l_{t}^{C^{ind}}<\alpha_{47})$. In the *Supply side checking 2* (M.16) submodule, we check whether any bank selected from the list of banks offering consumer loans at a specified interest rate has sufficient credit supply and that the regulatory requirements for sectoral exposures are met. If none is able to give this amount, the amount of loan is adjusted using adaptive algorithm (${\tilde{l}}_{t}^{C^{ind}}$).

### Module 3: Households Consumption Consumer Credit Admissibility 1 & 2 (M.17) & (M.18)

In submodules M.17 & M.18, household creditworthiness is checked with the bank assigned to the household or other bank (from the list of banks) selected in the *Supply side checking 2* (M.18) submodule. The first condition relates to the level of income per person after deduction of repayments of other loans. This level is specific to each bank:(56)$$\begin{array}{c} \left\{y_{t}^{HH}-{h_{t}^{HH}}_{t}-\left(\overset{N^{ind.HH}}{\underset{i=1}{\sum }}\left(l_{it}^{C_{q}^{ind}}+R_{it}^{C_{q}^{ind}}+l_{it}^{H_{q}^{ind}}+R_{it}^{H_{q}^{ind}}+l_{it}^{NH_{q}^{ind}}+R_{it}^{{NH}_{q}^{ind}}\right)\right)\right\}\geq \alpha_{50.b}\times N_{t}^{ind.HH}. \end{array}$$
The next conditions relate to credit history of the household (the probability of default of members of the family) ($({PD)}_{t}^{ind.HH}\leq \alpha_{49.b}$), and the maximum number of loans that can be granted to one household. If all conditions are met, the loan can be granted. For all loans granted to individuals and households, the applicant age (${\left(Age\right)}_{t}^{ind}\geq 18)$ and status of the labour market ($\Xi_{t}^{ind}=\left\{3||5\right\}$) are also checked. It is also possible to include a variable which counts the elapsed time since the last change in status on the labour market. If the individual works less than 4 quarters and is under 30, it is assumed that they have a fixed-term contract and no creditworthiness.

### Module 3: Households Consumption Consumer Credit and Purchase After Passing Supply Side Conditions and Credit Admissibility (M.19)

In this submodule, maturity is assigned to the loan ($M^{C^{ind}}$) depending on the amount of loan granted. Next, the value of debt service is updated, taking into account the civil status of the household members.
(57)$$\begin{array}{c} l_{t}^{C^{ind}}=\frac{1}{N_{t}^{adults.HH}}\times l_{temp}^{C^{ind}} \end{array}$$
(58)$$\begin{array}{c} l_{t}^{C_{q}^{ind}}=\frac{1}{N_{t}^{adults.HH}}\times \frac{l_{temp}^{C^{ind}}}{M^{C^{ind}}} \end{array}$$
(59)$$\begin{array}{c} R_{t}^{C^{ind}}+=\frac{1}{N_{t}^{adults.HH}}\times l_{temp}^{C^{ind}}\;{\times \left(1+i_{l^{C}}\right)}^{M^{C^{ind}}}-l_{temp}^{C^{ind}} \end{array}$$
(60)$$\begin{array}{c} R_{t}^{C_{q}^{ind}}=\frac{R_{t}^{C^{ind}}}{M^{C^{ind}}}. \end{array}$$
The amount of credit granted by the bank, the supply of the credit and the revenues of the bank in the given period are also increased.
(61)$$\begin{array}{c} S_{t}^{{l.C}^{ind}}-=l_{temp}^{C^{ind}} \end{array}$$
(62)$$\begin{array}{c} {\left({Rev}_{l^{C}}\right)}_{t}^{b}+=R_{t}^{C_{q}^{ind}}. \end{array}$$
Finally, the amount of consumer goods to be purchased from different industries is computed.
(63)$$\begin{array}{c} Q_{t}^{HH.(cons)}=\;Q_{basic}^{HH.(cons)}+\frac{\alpha_{43.tc.s}\times l_{t}^{C^{ind}}}{M^{C^{ind}}\times \left\{P_{t}^{s}\times \left(1+t_{VAT}\right)\right\}}. \end{array}$$

### Module 3: Households Consumption Supplier Searching & Purchase.hh (M.20)

The household next chooses the supplier. If the current supplier has sufficient stock of good (${Inv}_{t}^{est.(sup)}$) and the ratio of quality to price is higher than average ratio in the sector, then the household purchases the goods ($Q_{t}^{HH.(cons)}$) from this supplier.
(64)$$\begin{array}{c} {Inv}_{t}^{est.(sup)}\geq Q_{t}^{HH.(cons)}\;\&\&\frac{\alpha_{22.s-sup}\times {\left(Ql\right)}_{t}^{est.(sup)}}{\alpha_{23.s-sup}\times P_{t}^{est(.sup)}}>\frac{\alpha_{24.s-sup}\times {\left(Ql\right)}_{t}^{s}}{\alpha_{25.s-sup}\times P_{t}^{s}}. \end{array}$$
Otherwise, the household seeks a new supplier from incrementally more distant spatial locations. The requirements for the same spatial codes are loosened sequentially. Later, in the Purchase.hhs submodule, the profits from sales ${\left((Sl\right)}_{t}^{est.(sup)})$, demand ($Q_{t}^{d_{est.(sup)}}$), stock (${Inv}_{t}^{est.(sup)}$), liquid assets (${\left(LA\right)}_{t}^{est.(sup)}$) of suppliers are updated. In addition, deposits of consumers are updated.
(65)$$\begin{array}{c} {\left(Sl\right)}_{t}^{est.(sup)}+=P_{t}^{est(.\sup)}\times Q_{t}^{HH.(cons)} \end{array}$$
(66)$$\begin{array}{c} Q_{t}^{d_{est.(sup)}}+\;=\;Q_{t}^{HH.(cons)} \end{array}$$
(67)$$\begin{array}{c} {Inv}_{t}^{est.(sup)}-\;=Q_{t}^{HH.(cons)} \end{array}$$
(68)$$\begin{array}{c} {\left(LA\right)}_{t}^{est.(sup)}+=P_{t}^{est(.sup)}\times Q_{t}^{HH.(cons)} \end{array}$$
(69)$$\begin{array}{c} d_{it}^{ind}-=\frac{1}{N_{t}^{adults.HH}}P_{t}^{est(.sup)}\times Q_{t}^{HH.(cons)}. \end{array}$$

### Module 4: Public Contracts (M.21)

In this module, we complement the demand of the private sector with demand from the public sector. Public contracts are usually signed by large companies. A public contract is awarded when the product price is lower than the average price of the product in the sector ($P_{t}^{est}\leq P_{t}^{s}$). In addition, the probability of signing contracts increases with the size of the business and the quality of the product. If the terms of the contract are met, the value of the stored goods (${Inv}_{t}^{est}$), the demand for good ($Q_{t}^{d_{est.(sup)}}$), liquid assets of the supplier ${\left(LA\right)}_{t}^{est}$ and the value of sales ${\left(Sl\right)}_{t}^{est}$ are updated. The stock cannot be lower than the minimum fraction of production $(Y_{t}^{est}$) given by the parameter $\alpha_{57}$.
(70)$$\begin{array}{c} {Inv}_{t}^{est.(sup)}-=max\left\{0;min\left({Inv}_{t}^{est.(sup)},\;\alpha_{57}\times Y_{t}^{est.(sup)}\right)\right\} \end{array}$$
(71)$$\begin{array}{c} Q_{t}^{d_{est.(sup)}}+=min\left({Inv}_{t}^{est.(sup)},\;\alpha_{57}\times Y_{t}^{est.(sup)}\right)\; \end{array}$$
(72)$$\begin{array}{c} {\left(LA\right)}_{t}^{est.(sup)}+=\alpha_{58}\times min\left({Inv}_{t}^{est.(sup)},\;{\;\alpha \;}_{57}\times Y_{t}^{est.(sup)}\right)\; \end{array}$$
(73)$$\begin{array}{c} {\left(Sl\right)}_{t}^{est.(sup)}+=\alpha_{58}\times min\left({Inv}_{t}^{est.(sup)},\;{\;\alpha \;}_{57}\times Y_{t}^{est.(sup)}\right)\;. \end{array}$$

### Module 5: Households Mobility Accommodation Cost and Housing Stress (M.22)

The term property refers to the value of apartment or houses, with or without land. Households live in properties they own (${\left(Own\right)}_{t}^{HH}=1)$ or rent property from other households (${\left(Own\right)}_{t}^{HH}=0)$. The household may own more than one property. Real estate may be subject to residential $\left(l_{t}^{H^{ind}}\right)\;$ and non-residential loans ($l_{t}^{{NH}^{ind}}$). If the household lives in their own property, then the cost of living is equal to the sum of the financial obligations of the owners of the building, that is, the adult individuals forming the household who bought the property:(74)$$\begin{array}{c} h_{t}^{HH}=\beta_{0}+\;\beta_{1}\times \left\{\overset{N^{ind.HH}}{\underset{i=1}{\sum }}(l_{t}^{H_{q}^{ind.HH}}+\;R_{t}^{H_{q}^{ind}}\right)\}, \end{array}$$
where parameters $\beta_{0}$ and $\beta_{1}$ adjust the fixed and the variable parts of accommodation cost.

The cost of renting is calculated as a part $\beta_{1}$ of the sum of liabilities to the bank (loans) and the part $\beta_{4}$ of the cost of rent, calculated as the ratio of the price of the property to the number of households that rent this property.
(75)$$\begin{array}{c} h_{t}^{HH}=\beta_{1}\times \left\{\overset{N^{ind.HH}}{\underset{i=1}{\sum }}(l_{t}^{H_{q}^{ind.HH}}+\;R_{t}^{H_{q}^{ind}}\right)\}+\beta_{4}\times \frac{P_{t}^{prop}}{{\left(\#Rent\right)}_{t}^{prop}} \end{array}$$
In the first case of ownership, if the rental cost is greater than the specified parameter $\beta_{2}$ (part of the income $y_{t}^{HH}$), the household decides to sell the property. If the property was already marked for sale (${\left(ForSale\right)}_{t}^{prop}=1$), then the price ($P_{t}^{prop}$) has to be decreased by a percentage $\beta_{3}$. In practice, the parameter $\beta_{3}$ reflects how much the price has to be lowered in order to sell the property in next iteration. In the second case, if a household rents a property and the cost of rent is too high, it will start looking for a new home. The household looks for another building considering the status on the labour market ($\Xi_{t}^{int}=\left\{3||5\right\}$) and the age of the household members (${\left(Age\right)}_{t}^{ind}\geq 18$). If two adults in the household are working, the program randomly selects one of them and searches for a building near the person’s workplace (the algorithm checks and compares the spatial codes: ${\vartheta \;}_{t}^{{prop}^{1}},\;{\vartheta \;}_{t}^{{prop}^{2}},\;{\vartheta \;}_{t}^{{prop}^{3}},{\vartheta \;}_{t}^{{prop}^{4}},\;{\vartheta \;}_{t}^{{est}^{1}},\;{\vartheta \;}_{t}^{{est}^{2}},\;{\vartheta \;}_{t}^{{est}^{3}},\;{\vartheta \;}_{t}^{{est}^{4}}$). In addition to property locations, households take into account the price of the property from the appropriate price range and whether the building is for sale. If more than one property meets the criteria, one is selected at random and the spatial codes attributing the individual to the respective spatial units are updated.

If two adults are not working ($\Xi_{t}^{int}=\left\{1||2\right\}$) and the difference between the sum of individual deposits and the sum of the individual liabilities of family members is less than the subsistence level expressed by the parameter $\beta_{7}$, then the individuals are removed from the database, the household is removed from the database, and the corresponding records from the object *Consumers* are deleted as well.
(76)$$\begin{array}{c} \overset{N^{ind.HH}}{\underset{i=1}{\sum }}d_{it}^{ind.HH}-\overset{N^{ind.HH}}{\underset{i=1}{\sum }}(l_{t}^{H_{q}^{ind}}+\;R_{t}^{H_{q}^{ind}})\}+\beta_{4}\times \frac{P_{t}^{prop}}{{\left(\#Rent\right)}_{t}^{prop}}\;<\;\beta_{7} \end{array}$$
The exception is a situation in which an adult is under 25. Then the assumption applies that they are still living with their parents. If individuals and households are removed from the database, all records related to insolvency are updated. Consequently, the non-performing loans for a given bank and sectors are increased. The probability of bank’s default increases as well. Especially,
(77)$$\begin{array}{c} {\left({NPL}_{l^{I}}\right)}_{t}^{b}+={\left(pl\right)}_{t}^{I^{ind}} \end{array}$$
(78)$$\begin{array}{c} {\left({NPL}_{l^{C}}\right)}_{t}^{b}+={\left(pl\right)}_{t}^{C^{ind}} \end{array}$$
(79)$$\begin{array}{c} {\left({NPL}_{l^{H}}\right)}_{t}^{b}+={\left(pl\right)}_{t}^{H^{ind}} \end{array}$$
(80)$$\begin{array}{c} {\left({NPL}_{l^{NH}}\right)}_{t}^{b}+={\left(pl\right)}_{t}^{{NH}^{ind}} \end{array}$$
where ${\left(pl\right)}_{t}$ is the sum of liabilities to the bank for outstanding (respectively investment, consumer, housing and non-housing) loans that have to be repaid in the given iteration.

### Module 5: Households Mobility Profits from Rent (Accommodation & Housing Stress) (M.23)

In this module, property attributes are updated if the household obtains profits from renting the property. The algorithm checks all household properties. The primary property (${\left(PH\right)}_{t}^{prop}=1$) cannot be sublet to another household. If the household has a second property (${\left(PH\right)}_{t}^{prop}=0$), the number of households that live in the house is checked. If none live there, it is marked for sale (${\left(ForSale\right)}_{t}^{prop}=1$). If it was previously marked for sale (${\left(ForSale\right)}_{t-1}^{prop}=1$), its price is reduced by a certain percentage of the value that is specified by the parameter $\beta_{3}$.
(81)$$\begin{array}{c} P_{t}^{prop}=\beta_{3}\times P_{t-1}^{prop}. \end{array}$$
Revenues from renting second property (${\left(Rent\right)}_{{RE}_{t}}^{HH}$) are updated according to:(82)$$\begin{array}{c} {\left(Rent\right)}_{{RE}_{t}}^{HH}+=\beta_{4}\times P_{t}^{prop}. \end{array}$$

### Module 5: Households Mobility - Decisions About Funding Housing and Non-Housing Purchase (M.24)

In this sub-module the household decides how to finance the purchase of houses and other non-housing purchases. If the current funds and savings in the bank account are greater than the price of the cheapest property on the market that has already been marked for sale (${\left(ForSale\right)}_{t-1}^{prop}=1$), it can potentially buy a property in cash.
(83)$$\begin{array}{c} \overset{N^{ind.HH}}{\underset{i=1}{\sum }}d_{it}^{ind.HH}-\overset{N^{ind.HH}}{\underset{i=1}{\sum }}(l_{t}^{H_{q}^{ind}}+\;R_{t}^{H_{q}^{ind}})\}\;>\;P_{{min}_{t}}^{prop}. \end{array}$$
In practice, we assume that a household buys a property in cash, only with a given probability (${pr}_{3}$) in the model. If a member of the household is an individual who owns the firm (${\left(Entr\right)}_{t}^{ind}=1$) or is unemployed ($\Xi_{t}^{ind}=2$) with a high entrepreneurial spirit (${\left(EntrS\right)}_{t}^{ind}>\;0.5$), then the funds will be first invested in the firm rather than housing or non-housing purchases. We update the status of individual as an entrepreneur ${\left((Entr\right)}_{t}^{ind}=1)$. If a household does not have enough resources to buy a property in cash, it will apply for a loan and the sub-module *Macroprudential ratios* is moved to.

### Module 5: Households Mobility Macroprudential Ratios (M.25)

In this submodule, macroprudential ratios are computed. In addition, the total indebtedness, debt servicing and total assets held by the household are computed as a temporal variables. The following macroprudential ratios were included in the system for residential and non-residential loans: total debt to assets ratio, debt to income ratio, debt service to income ratio, loan-to-value ratio.
(84)$$\begin{array}{c} {\left(IndebtQ\right)}_{temp}^{HH}=\overset{N^{ind.HH}}{\underset{i=1}{\sum }}\left(l_{it}^{C_{q}^{ind}}+l_{it}^{H_{q}^{ind}}+l_{it}^{NH_{q}^{ind}}+l_{it}^{I_{q}^{ind}}\right) \end{array}$$
(85)$$\begin{array}{c} {\left(DServiceQ\right)}_{temp}^{HH}=\overset{N^{ind.HH}}{\underset{i=1}{\sum }}\left({R_{it}^{C_{q}^{ind}}+R}_{it}^{H_{q}^{ind}}+R_{it}^{{NH}_{q}^{ind}}+R_{it}^{I_{q}^{ind}}\right) \end{array}$$
(86)$$\begin{array}{c} {\left(AssetsQ\right)}_{temp}^{HH}=\overset{N^{ind.HH}}{\underset{i=1}{\sum }}d_{it}^{ind.HH} \end{array}$$
(87)$$\begin{array}{c} {\left(DTA\right)}_{t}^{HH}=\frac{{\left(IndebtQ\right)}_{temp}^{HH}}{{\left(AssetsQ\right)}_{temp}^{HH}} \end{array}$$
(88)$$\begin{array}{c} {\left(DTI\right)}_{t}^{HH}=\;\frac{{\left(IndebtQ\right)}_{temp}^{HH}}{y_{t}^{HH}} \end{array}$$
(89)$$\begin{array}{c} {\left(DSTI\right)}_{t}^{HH}=\frac{{\left(DServiceQ\right)}_{temp}^{HH}}{y_{t}^{HH}} \end{array}$$
(90)$$\begin{array}{c} {\left(LTV\right)}_{t}^{HH}=\frac{\sum_{i=1}^{N^{ind.HH}}l_{it}^{H^{ind}}}{P_{t}^{prop}}\;\;. \end{array}$$

### Module 5: Households Mobility Housing and Non-Housing Loans (M.26)

In this sub-module, the household’s creditworthiness and supply side conditions are checked. Firstly, basic requirements are checked. If the household debt ratios exceed regulatory requirements, the household lacks creditworthiness (${\left(DSTI\right)}_{t}^{HH}>\beta_{8.b}||\;{\left(DTI\right)}_{t}^{HH}>\beta_{9.b}||\;{\left(DTA\right)}_{t}^{HH}>\beta_{10.b})$. Likewise, if all household members study, they are inactive on the labour market or are living on social benefits ($\Xi_{t}^{ind}=\left\{1\right|\left|2\right|\left|4\right\}$). The model also takes into account the situation of people under the age of 30 who work less than 1 year in a company that also do not have creditworthiness $((\Xi_{t}^{ind}\;=\;3\;\&\&\;\;{\left(Age\right)}_{t}^{ind}\;<\;30$ && $\Psi_{t}^{ind}\leq 4)\;||\;({\Lambda \;}_{t}^{ind}=2\;\;\&\&\;\;\Xi_{t}^{ind}=\left(1\left|\left|2\right|\right|4\right)\;\&\&\;\Xi_{t}^{ind}=3\;\;\;\&\&\;\;\;{\left(Age\right)}_{t}^{ind}<30$ && $\Psi_{t}^{ind}\leq 4\;))$. Then the value of collateral is estimated. For the all properties that the household owns, the property prices are summed up (${\left(Collat\right)}_{temp}^{max}+=P_{t}^{prop}$). Households may apply for a residential or non-residential loan under a pledge of real estate. If the household owns at least one property, it will apply for a residential loan with a given probability, and for a non-residential loan with one minus that probability. On the other hand, if a household has rented a property (i.e., is not an owner), it will first apply for a residential loan. In most cases, the first property is bought in cash. The value of the mortgage loan is equal to the difference between the price of the cheapest real estate on the market, and the savings (less the charges for other loans); this is according to the following equation:(91)$$\begin{array}{c} l_{temp}^{H^{HH}}=P_{{min}_{t}}^{prop}-\beta_{15}\times \left(\overset{N^{ind.HH}}{\underset{i=1}{\sum }}d_{it}^{ind.HH}-{\left(Debt\right)}_{temp}^{HH}-{\left(DebtServ\right)}_{temp}^{HH}\right) \end{array}$$
where ${\;(Debt)}_{temp}^{HH},\;{\left(DebtServ\right)}_{temp}^{HH}$ are given by the expressions:(92)$$\begin{array}{c} {\left(Debt\right)}_{temp}^{HH}=\overset{N^{ind.HH}}{\underset{i=1}{\sum }}\left({\left(pl\right)}_{it}^{C_{q}^{ind}}+(p{l)}_{it}^{H_{q}^{ind}}+{\left(pl\right)}_{it}^{NH_{q}^{ind}}+{\left(pl\right)}_{it}^{I_{q}^{ind}}\right) \end{array}$$(93)$$\begin{array}{c} {\left(DebtServ\right)}_{temp}^{HH}=\overset{N^{ind.HH}}{\underset{i=1}{\sum }}\left(R_{it}^{C_{q}^{ind}}+R_{it}^{H_{q}^{ind}}+R_{it}^{{NH}_{q}^{ind}}+R_{it}^{I_{q}^{ind}}\right) \end{array}$$
Households choose a bank with which to apply for a loan, taking into account the interest rate. At the same time, the bank must have sufficient funds and be willing to accept the LTV ratio of applicant (${LTV}_{temp}^{HH}=\left(\left(l_{t}^{H^{ind}}+l_{t}^{{NH}^{ind}}\right)/P_{t}^{prop}\right)$). If the household does not obtain credit in the bank that offers the lowest interest rate, it resubmits the request to another bank on the list that requires a higher interest rate. After receiving the loan, the household buys the property. The value of housing loans granted by the bank is also updated. The value of the household loans obtained is then updated. If the household consists of a marriage, the value of granted loans is halved. If the property is the main residence, the cost of accommodation is also updated. In the case of non-residential loans, the scheme works in an analogous way, with the value of the residential loan being equal to:(94)$$\begin{array}{c} l_{temp}^{{NH}^{HH}}=\beta_{13.tc}\times {\{P}_{{min}_{t}}^{prop}-\beta_{15}\times \left(\overset{N^{ind.HH}}{\underset{i=1}{\sum }}d_{it}^{ind.HH}-{\left(Debt\right)}_{temp}^{HH}-{\left(DebtServ\right)}_{temp}^{HH}\right)\}. \end{array}$$
After checking the creditworthiness and credit supply of the selected bank, the value of non-residential loans obtained are updated, taking into account the civil status of household members.

### Module 6: Individuals’ Records Updating

Individuals’ records updating module ensures the maintenance of demographic trends observed empirically in the simulation. The heterogeneity of individuals has been taken into account, as has population dynamics.

### Module 6: Individuals’ Records Updating Inheritor or Life (M.27)

In the first sub-module the individuals may die with probabilities $(\rho_{age.gender}$) depending on age (${\left(Age\right)}_{t}^{ind}$) and gender ($G^{ind}=\left\{0||1\right\}$). If a person dies, after the submodules *Inheritor* (M.28) and *Inheritance* (M.29 & M.30) have been applied, the individual is deleted from the database. This insolvency has a direct impact on the banks in the form of an increase in non-performing loans. In the case of survival of an individual, the program increases the age of the person (${\left(Age\right)}_{t}^{ind}++$) and the period since the last change of status on the labour market ($\Psi_{t}^{ind}++$) and continues in the sub-module *Updating consumer type* (M.31).

### Module 6: Individuals’ Records Updating Inheritor (M.28)

In this submodule, the heir in the event of death of an individual is determined. First, the age of the deceased person is checked. If the deceased person was an adult (${\left(Age\right)}_{t}^{ind}\geq 18$), inheritance may be considered. Then, all members of the households and adults in the households are counted. If an adult over the age of 18 has died, who does not have family, the labor status of this individual is checked ($\Xi_{t}^{ind}$). If this individual was an entrepreneur ($\Xi_{t}^{ind}=5$), the number of employees ($L_{t}^{firm}$), in the company is checked. If it was a sole proprietorship, then the firm is for sale (${\left(ForSale\right)}_{t}^{firm}=1$). Otherwise, one of workers in their company who earned the highest wage in the previous period is selected. If the deceased person was not an entrepreneur, then the algorithm selects an inheritor at random from the group of working adults ($\Xi_{t}^{ind}\neq 1\;\&\&\;{\left(Age\right)}_{t}^{ind}\geq 18$). If a child dies (${\left(Age\right)}_{t}^{ind}\leq 18$), they do not leave material property, rather the consumer type of the child’s parents changes. If the parents have no more children and if all household members are over 67 years old then we update the consumer type (${\left(ConsT\right)}_{t}^{cons}=3$). If both parents were under 67, they are also updated to another consumer type (${\left(ConsT\right)}_{t}^{cons}=4$). If the family consists of more than two adults then the consumer type is also updated (${\left(ConsT\right)}_{t}^{cons}=6$). If the deceased adult person has a family, then two cases must be differentiated. If they had children, the eldest person in the family is the inheritor and the consumer type does not change (${\left(ConsT\right)}_{t}^{cons}=\left\{5||6\right\}$). If there were no children, then the spouse is the inheritor and the consumer type changes depending on the age of the spouse $({\left(ConsT\right)}_{t}^{cons}=\left\{1||2\}\right)$. If the deceased adult was a single parent, then the adoption of a child is considered in the model in the module *Adoption*. The algorithm looks for a new couple or a single individual to be a parent and sum up deposits, ownership of properties and firms. If they own more than two properties, they are designated for sale (${\left(ForSale\right)}_{t}^{firm}=1$). We remove the deceased person from the database. We continue to the *Inheritance* modules (M.29 & M.30).

### Module 6: Individuals’ Records Updating Inheritance: Deposits & Firms (M.29)

In this submodule, we pass the deposits and firms to the inheritor. Firstly, savings after deductions of housing loans pending to be paid are given to the inheritor. If these savings are greater than zero ($d_{t}^{dec.ind}-\left({\left(pl\right)}_{t}^{H^{ind}}+{\left(pl\right)}_{t}^{{NH}^{ind}}\right)>0$), then the deposits of the heir are updated.
(95)$$\begin{array}{c} d_{t}^{heir.ind}+=\left(1-t_{inh}\right)\times \left(d_{t}^{dec.ind}-{\left(pl\right)}_{t}^{H^{ind}}-{\left(pl\right)}_{t}^{{NH}^{ind}}\right) \end{array}$$
The parameter $t_{inh}\;$ stands for taxes that must be paid. In the model we assumed that consumer loans are not inherited and hence the inflow of non-performing loans of banks (${\left({NPL}_{l^{C}}\right)}_{t}^{b}$) and the probability of default of bank (${\left(PD\right)}_{t}^{b}$) are updated.
(96)$$\begin{array}{c} {\left({NPL}_{l^{C}}\right)}_{t}^{b}+={\left(pl\right)}_{t}^{C^{ind}} \end{array}$$
If individual debts are greater than savings, then the variables are updated separately. In this way it is possible for the heir to inherit the debt from the loan taken in the past.
(97)$$\begin{array}{c} d_{t}^{heir.ind}+=\left(1-t_{inh}\right)\times d_{t}^{dec.ind} \end{array}$$
(98)$$\begin{array}{c} l_{t}^{H^{heir.ind}}+={\left(pl\right)}_{t}^{H^{dec.ind}} \end{array}$$
(99)$$\begin{array}{c} {\left(pl\right)}_{t}^{H^{heir.ind}}+={\left(pl\right)}_{t}^{H^{dec.ind}} \end{array}$$
(100)$$\begin{array}{c} l_{t}^{H_{q}^{heir.ind}}+=l_{t}^{H_{q}^{dec.ind}} \end{array}$$
(101)$$\begin{array}{c} R_{t}^{H^{heir.ind}}+=R_{t}^{H^{dec.ind}} \end{array}$$
(102)$$\begin{array}{c} R_{t}^{H_{q}^{heir.ind}}+=R_{t}^{H_{q}^{dec.ind}} \end{array}$$
(103)$$\begin{array}{c} l_{t}^{{NH}^{heir.ind}}+={\left(pl\right)}_{t}^{NH^{dec.ind}} \end{array}$$
(104)$$\begin{array}{c} {\left(pl\right)}_{t}^{{NH}^{heir.ind}}+={\left(pl\right)}_{t}^{{NH}^{dec.ind}} \end{array}$$
(105)$$\begin{array}{c} l_{t}^{{NH}_{q}^{heir.ind}}+=l_{t}^{{NH}_{q}^{dec.ind}} \end{array}$$
(106)$$\begin{array}{c} R_{t}^{{NH}^{heir.ind}}+=R_{t}^{{NH}^{dec.ind}} \end{array}$$
(107)$$\begin{array}{c} R_{t}^{{NH}_{q}^{heir.ind}}+=R_{t}^{{NH}_{q}^{dec.ind}} \end{array}$$
(108)$$\begin{array}{c} {\left({NPL}_{l^{C}}\right)}_{t}^{b}+={\left(pl\right)}_{t}^{C^{ind}}. \end{array}$$
As the non-performing consumer loans increase, the probability of default of bank (${\left(PD\right)}_{t}^{b}$) increases as well. If the deceased person was not the owner of the firm, then we update savings (deposits) only. Otherwise, the inheritor may or may not run the business in the future. The decision on running a business depends on their previous earnings relative to the potential income from business activity.
(109)$$\begin{array}{c} w_{t}^{inh.ind}+{\left(sub\right)}_{t}^{inh.ind}<rndm\left(0,1\right)\times w_{t}^{dec.ind} \end{array}$$
If the inheritor decides to run the business, the status of entrepreneur (${\left(Entr\right)}_{t}^{ind}=1$), the labor status ($\Xi_{t}^{ind}=5$) and the number of periods since the last change in labor status ($\Psi_{t}^{ind}=0$) are updated. The public assistance is ceased (${\left(sub\right)}_{t}^{ind}=0$). The inheritor is responsible for paying back the investment loans.
(110)$$\begin{array}{c} l_{t}^{I^{heir.ind}}+={\left(pl\right)}_{t}^{I^{dec.ind}} \end{array}$$
(111)$$\begin{array}{c} {\left(pl\right)}_{t}^{I^{heir.ind}}+={\left(pl\right)}_{t}^{I^{dec.ind}} \end{array}$$
(112)$$\begin{array}{c} l_{t}^{I_{q}^{heir.ind}}+=l_{t}^{I_{q}^{dec.ind}} \end{array}$$
(113)$$\begin{array}{c} R_{t}^{I^{heir.ind}}+=R_{t}^{I^{dec.ind}} \end{array}$$
(114)$$\begin{array}{c} R_{t}^{I_{q}^{heir.ind}}+=R_{t}^{I_{q}^{dec.ind}} \end{array}$$
If the inheritor decides to sell, the company is marked for sale (${\left(ForSale\right)}_{t}^{firm}=1$). The inflow of non-performing loans is increased by the amount of investment loans that will not be paid. The probability of default of bank (${\left(PD\right)}_{t}^{b}$) is also increased in that case.
(115)$$\begin{array}{c} {\left({NPL}_{l^{I}}\right)}_{t}^{b}+={\left(pl\right)}_{t}^{I^{ind}}. \end{array}$$

### Module 6: Individuals’ Records Updating Inheritance: Properties (M.30)

In this module, the number of adults who were in the household of deceased person is checked, as is whether the deceased person rented (${\left(Own\right)}_{t}^{HH}=0$) or owned their residence (${\left(Own\right)}_{t}^{HH}=1$). If the the deceased individual rented his residence, then the number of households that rents this property needs to decrease (${\left(\#Rent\right)}_{t}^{prop}--$). If the deceased person owned their residence and had no mortgage, then the property is marked for sale (${\left(ForSale\right)}_{t}^{firm}=1$). However, if the deceased person had a housing loan to buy the property, the bank receives the property and attempts to sell it (${\left(ForSale\right)}_{t}^{firm}=1$). The bank looks for any household which sum of deposits ($\sum_{i=1}^{N^{ind.HH}}d_{it}^{ind.HH}$) is greater than the property price ($P_{t}^{prop}$). If it is found, then the bank updates the inflow of non-performing loans (${\left({NPL}_{l^{H}}\right)}_{t}^{b},\;{\left({NPL}_{l^{NH}}\right)}_{t}^{b})$ and the revenue from selling the property (${\left({Rev}_{l^{H}}\right)}_{t}^{b}$).
(116)$$\begin{array}{c} {\left({NPL}_{l^{H}}\right)}_{t}^{b}+={\left(pl\right)}_{t}^{H^{dec.ind}} \end{array}$$
(117)$$\begin{array}{c} {\left({NPL}_{l^{NH}}\right)}_{t}^{b}+={\left(pl\right)}_{t}^{{NH}^{dec.ind}} \end{array}$$
(118)$$\begin{array}{c} {\left({Rev}_{l^{H}}\right)}_{t}^{b}+=\left(1-\beta_{22}\right)\times P_{t}^{prop} \end{array}$$
where $\beta_{22}$ expresses the transaction costs.

The new owners of the property proportionally update the deposits ($d_{t}^{ind.HH}$) after the purchase.
(119)$$\begin{array}{c} d_{t}^{ind.HH}-=\frac{1}{N^{ind.HH}}\times \left(1-\beta_{22}\right)\times P_{t}^{prop} \end{array}$$
If the bank is not able to sell the property, the price ($P_{t}^{prop}$) is lowered gradually:(120)$$\begin{array}{c} P_{t}^{prop}=0.95\times P_{t-1}^{prop}. \end{array}$$

### Module 6: Individuals’ Records Updating Updating Consumer Type (M.31)

In this module the consumer type of the household is updated. If the individual reaches the age of 67, then the consumer type of the household the individual belongs to is updated (ie. ${\left(ConsT\right)}_{t}^{cons}$=1 if the individual was single, while ${\left(ConsT\right)}_{t}^{cons}=3\;$ if was married). If the individual become an adult (${\left(Age\right)}_{t}^{ind}=18$), then a new household is created. The individual continues to live in the same property. The accommodation cost and the income are computed according to the formulas expressed in the previous modules (M.11, M.12 & M.22).

### Module 6: Individuals’ Records Updating Education Level (M.32)

In this submodule the level of education is updated. If an individual continues to study (${\left(EducP\right)}_{t}^{ind}+\;+$) and exceeds the number of periods that is needed to complete a particular level of education (i.e., primary school, secondary school, college, university degree, doctoral school), then a variable describing their education level (${\left(Educ\right)}_{t}^{ind}++$) is updated.

### Module 6: Individuals’ Records Updating Continue Education (M.33)

The individuals continue education with a probability depending on age and gender (${pr}_{age.gender}$). According to this probability, the completed level of education $({\left(Educ\right)}_{t}^{ind}$), labour market status ($\Xi_{t}^{ind}$) of an individual and the number of periods that have passed since the last change of status in the labour market ($\Psi_{t}^{ind}++$) are updated. In this module the labor status of individuals that continue education is set to one ($\Xi_{t}^{ind}$=1).

### Module 6: Individuals’ Records Updating Divorces (M.34)

In this submodule, individuals may divorce. The algorithm checks the civil status of individuals (${\Lambda \;}_{t}^{ind}$). The probability of divorce depends on the age of the spouses (${pr}_{40.age}$). After the divorce, a new household is created for ex-husband and the consumer types (${\left(ConsT\right)}_{t}^{cons}$) are adjusted. In the case of divorce, children always stay with the mother. The ex-husband pays the alimony $\left({\left(Don\right)}_{t}^{\left(husb\right).HH}\right)$ which depends on his earnings ($w_{t}^{ind}$) and the number of children (${\left(\#child\right)}_{temp}$).
(121)$$\begin{array}{c} {\left(Don\right)}_{t}^{\left(wife\right).HH}+=\beta_{39}\times w_{t}^{ind}\times {\left(\#child\right)}_{temp} \end{array}$$
(122)$$\begin{array}{c} {\left(Don\right)}_{t}^{\left(husb\right).HH}-=\beta_{39}\times w_{t}^{ind}\times {\left(\#child\right)}_{temp} \end{array}$$
Both adults remain in the same property hence they both contribute to rent to cover the accommodation cost. The ex-wife is the owner, while the ex-husband lives there temporary. If the marriage had more than two properties, then each of them stays in one of them and the rest is for sale (${\left(ForSale\right)}_{t}^{prop}=1$). Their status of the owner (${\left(Own\right)}_{t}^{HH}=1$), the variable principal housing (${\left(PH\right)}_{t}^{prop}=1$) and the number of families that rent the property (${\left(\#Rent\right)}_{t}^{prop}--$) are updated. If the property was purchased by credit, then the amount of loans pending to be paid by ex-wife has to be updated as well.
(123)$$\begin{array}{c} l_{t}^{H^{\left(wife\right).ind}}+=l_{t}^{H^{\left(husb\right).ind}},\;l_{t}^{H^{\left(husb\right).ind}}=0 \end{array}$$
(124)$$\begin{array}{c} {\left(pl\right)}_{t}^{H^{(wife).ind}}+={\left(pl\right)}_{t}^{H^{\left(husb\right).ind}},\;{\left(pl\right)}_{t}^{H^{\left(husb\right).ind}}=0 \end{array}$$
(125)$$\begin{array}{c} l_{t}^{H_{q}^{\left(wife\right).ind}}+=l_{t}^{H_{q}^{\left(husb\right).ind}},\;l_{t}^{H_{q}^{\left(husb\right).ind}}=0 \end{array}$$
(126)$$\begin{array}{c} R_{t}^{H^{\left(wife\right).ind}}+=R_{t}^{H^{\left(husb\right).ind}},\;R_{t}^{H^{\left(husb\right).ind}}=0 \end{array}$$
(127)$$\begin{array}{c} R_{t}^{H_{q}^{\left(wife\right).ind}}+=R_{t}^{H_{q}^{\left(husb\right).ind}},\;R_{t}^{H_{q}^{\left(husb\right).ind}}=0 \end{array}$$
(128)$$\begin{array}{c} l_{t}^{{NH}^{\left(wife\right).ind}}+=l_{t}^{{NH}^{\left(husb\right).ind}},\;l_{t}^{{NH}^{\left(husb\right).ind}}=0 \end{array}$$
(129)$$\begin{array}{c} l_{t}^{{NH}^{\left(wife\right).ind}}+=l_{t}^{{NH}^{\left(husb\right).ind}},\;l_{t}^{{NH}^{\left(husb\right).ind}}=0 \end{array}$$
(130)$$\begin{array}{c} {\left(pl\right)}_{t}^{{NH}^{\left(wife\right).ind}}+={\left(pl\right)}_{t}^{{NH}^{\left(husb\right).ind}},\;{\left(pl\right)}_{t}^{{NH}^{\left(husb\right).ind}}=0 \end{array}$$
(131)$$\begin{array}{c} l_{t}^{NH_{q}^{\left(wife\right).ind}}+=l_{t}^{NH_{q}^{\left(husb\right).ind}},\;l_{t}^{NH_{q}^{\left(husb\right).ind}}=0 \end{array}$$
(132)$$\begin{array}{c} R_{t}^{{NH}^{\left(wife\right).ind}}+=R_{t}^{{NH}^{\left(husb\right).ind}},\;R_{t}^{{NH}^{\left(husb\right).ind}}=0 \end{array}$$
(133)$$\begin{array}{c} R_{t}^{{NH}_{q}^{\left(wife\right).ind}}+=R_{t}^{{NH}_{q}^{(husb).ind}},\;R_{t}^{{NH}_{q}^{(husb).ind}}=0. \end{array}$$

### Module 6: Individuals’ Records Updating Marriages (M.35) & Births (M.36)

Each single adult can get married in the model. The probability of getting married depends on civil status, age and gender (${pr}_{civ.st.age.gend}$). In the model, the age difference between partners should not exceed 10 years (${\left|(Age\right)}_{t}^{ind1}-{\left(Age\right)}_{t}^{ind2}|<10\;\&\&\;G^{ind1}\neq G^{ind2}\;\&\&\;{\left(Age\right)}_{t}^{ind1},{\left(Age\right)}_{t}^{ind2}\;>\;18,\;{\Lambda \;}_{t}^{ind}\;\neq \;2$). In the case of marriage, the marital status of individuals (${\Lambda \;}_{t}^{ind}=2$) as well as the codes identifying the households $\left({\left(Id\right)}^{HH}\right)$ are updated. In addition, the total number of properties and their ownership (${\left(Own\right)}_{t}^{HH}$) are updated. One of properties is marked as the principal residence (${\left(PH\right)}_{t}^{prop}\;=\;1$). If the marriage owns more than two properties, they are marked for sale (${\left(ForSale\right)}_{t}^{prop}=1$). The algorithm looks for a household-buyer who has sufficient deposits to pay for the property.
(134)$$\begin{array}{c} \overset{N^{ind.HH}}{\underset{i=1}{\sum }}d_{it}^{ind.HH}-\overset{N^{ind.HH}}{\underset{i=1}{\sum }}(l_{t}^{H_{q}^{ind.HH}}+\;R_{t}^{H_{q}^{ind}})>P_{t}^{prop}. \end{array}$$
When the buyer is found, then deposits are reduced by the amount equal to the sum of prices of *J* properties.
(135)$$\begin{array}{c} d_{it}^{buy.ind.HH}-=\frac{1}{N^{ind.HH}}\times \overset{J}{\underset{j=1}{\sum }}P_{jt}^{prop} \end{array}$$
If the buyer is not found, the price is decreased by 10%. After selling the properties, the marriage can pay back the housing and non-housing loans. If housing loans pending to be paid are greater than the revenue from selling the properties, then the algorithm decrease the amount of housing loans to be paid.
(136)$$\begin{array}{c} {\left(pl\right)}_{t}^{H^{seller.ind}}-=\frac{1}{N^{ind.HH}}\times \overset{J}{\underset{j=1}{\sum }}P_{jt}^{prop}. \end{array}$$
Otherwise, we check whether the individual could also pay back the non-housing loans. In that case, if the non-housing loans pending to be paid are greater than the difference between the revenue from selling the properties and housing loans pending to be paid, then the non-housing loans are decreased.
(137)$$\begin{array}{c} {\left(pl\right)}_{t}^{{NH}^{seller.ind}}-=\frac{1}{N^{ind.HH}}\times \overset{J}{\underset{j=1}{\sum }}P_{jt}^{prop}-{\left(pl\right)}_{t}^{H^{seller.ind}}. \end{array}$$
Then, housing loans pending to be paid are set to zero. If the price for which the properties were sold was greater than the sum of all liabilities then, deposits are updated.
(138)$$\begin{array}{c} d_{it}^{seller.ind.HH}+=\frac{1}{N^{ind.HH}}\times \overset{J}{\underset{j=1}{\sum }}P_{jt}^{prop}-{\left(pl\right)}_{t}^{H^{seller.ind}}-{\left(pl\right)}_{t}^{{NH}^{seller.ind}}. \end{array}$$
If the wife had children from a previous marriage, then the consumer type does not change. Otherwise, the model checks the age of the individuals. If at least one member of the household is over 67, the consumer type is updated (${\left(ConsT\right)}_{t}^{cons}=3$). In the *Births* sub-module, children are born with the probability depending on age and civil status of the mother (${pr}_{age.st.civil}$). All necessary records are created for a newborn in submodule (M.36).

### Module 6: Individuals’ Records Updating Updating Entrepreneurs (M.37)

In this module, decisions are made on whether an adult individual $\left({\left(Age\right)}_{t}^{ind}>18\right)$ becomes an entrepreneur. If the individual is already an entrepreneur (${\left(Entr\right)}_{t}^{ind}=1$), then the program skips this module and continues in the *Probability of opening a new firm* module (M.38). For the individuals unemployed ($\Xi_{t}^{ind}=2$), employed in private sector ($\Xi_{t}^{ind}=3$), inactive ($\Xi_{t}^{ind}=4$), or employed in the public sector ($\Xi_{t}^{ind}=5$), the probability of becoming an entrepreneur is computed. The probability depends on the experience in running a business ($({EntrP)}_{t}^{ind}=1$), gender ($G^{ind}$), age (${\left(Age\right)}_{t}^{ind}$), level of education (${\left(Educ\right)}_{t}^{ind}$) and the period that has passed since the last change of status on the labour market ($\Psi_{t}^{ind})$
(139)$$\begin{array}{ll} ({EntrS)}_{t}^{ind}= & [(\max\{\;\min(1,\;({EntrS)}_{t-1}^{ind}\times \\ & \times \left(\beta_{G}G^{ind}+\beta_{Educ}{\left(Educ\right)}_{t}^{ind}+\beta_{EntrP}\left({EntrP)}_{t}^{ind}+\beta_{Age}{\left(Age\right)}_{t}^{ind}+\beta_{\Psi }\Psi_{t}^{ind}\right)\right),\;\beta_{EntrS_{max}}\}]. \end{array}$$
If the entrepreneur spirit is high enough, the individual becomes an entrepreneur (${\left(Entr\right)}_{t}^{ind}=1$). Otherwise, the program moves to module *Updating individual labour status* (M.47).

### Module 7: Firm Demography Probability of Opening a New Firm (M.38)

In the *Firm demography* module, we analyse the possibility of entrepreneurs opening new companies in the system. As a part of the first *Probability of opening a new firm* submodule, the probability of opening a firm in a given industry is determined. This probability depends on previous experience in the industry ($({EntrP)}_{t}^{ind}$), level of education ${\left((Educ\right)}_{t}^{ind})$ and age of the entrepreneur (${\left(Age\right)}_{t}^{ind}$). Firstly, temporary variables are calculated that are required to calculate the probability of creating a new company in a given sector.
(140)$$\begin{array}{c} {temp}_{1}={\gamma \;}_{2}+{\gamma \;}_{3}\times ({EntrP)}_{t}^{ind}+{\gamma \;}_{4}\times {\left(Educ\right)}_{t}^{ind}+{\gamma \;}_{5}\times \left({\left(Age\right)}_{t}^{ind}+{\gamma \;}_{6}\right) \end{array}$$
(141)$$\begin{array}{c} {temp}_{2}={\gamma \;}_{7}+{\gamma \;}_{8}\times ({EntrP)}_{t}^{ind}+{\gamma \;}_{9}\times {\left(Educ\right)}_{t}^{ind}+{\gamma \;}_{10}\times \left({\left(Age\right)}_{t}^{ind}+{\gamma \;}_{11}\right) \end{array}$$
According to a given probability, the entrepreneur will open a new company in the most profitable sector (if $pr<{temp}_{1}$) and then in the second most profitable sector (if $pr<{temp}_{2}$). In this module we use information from the *Initialisation* module (M.1). If the entrepreneur cannot open up activity in these two sectors, they will start operating in a randomly selected industry.

### Module 7: Firm Demography New License (M.39)

A number of criteria must be met to create a new business. First and foremost, it is necessary to obtain a license. The likelihood of obtaining a license depends on the size of the company compared to the average firm size in the industry and industry-specific parameters. The company’s initial size ${\left(Size\right)}_{temp}$ is drawn from the normal distribution with the expected value $\left({\left(AL\right)}_{t}^{s}\right)$ and the standard deviation of the size of firms (empirically determined) $\left({\left(SD\right)}_{t}^{s}\right)$.
(142)$$\begin{array}{c} {\left(Size\right)}_{temp}\;was\;drawn\;from\;N\left({\left(AL\right)}_{t}^{s},{\left(SD\right)}_{t}^{s}\right) \end{array}$$
(143)$$\begin{array}{c} {\left(InitSize\right)}_{temp}^{firm}=\max\left\{\min\left({\gamma \;}_{12}\times {\left(AL\right)}_{t}^{s},\;{\left(Size\right)}_{temp}\right)\;,0\right\}. \end{array}$$

### Module 7: Firm Demography Funding New Firm Creation (M.40)

This module analyses the possibility of opening a company from funds held by the entrepreneur or co-financing the opening of the company from external sources. Large companies can be co-financed by various individuals. In this case, the initial capital of the company is equal to the sum of the contributions of the individuals. The amount depends on the type of household to which the individual belongs. These individuals hold shares in the company. According to the algorithm, the cost of setting up a company depends on the size of the company measured by the number of employees (${\left(InitSize\right)}_{temp}^{firm}$), average salaries in the industry ${\bar{W}}_{t}^{s}$, and industry-specific fixed costs specified by the parameter ${\gamma \;}_{17.s}$:(144)$$\begin{array}{c} {\left(CostNew\right)}_{temp}^{firm}={\left(InitSize\right)}_{temp}^{firm}\;\times {\bar{W}}_{t}^{s}\;\times {\gamma \;}_{16}+{\gamma \;}_{17.s}. \end{array}$$
The value of the individual’s savings after deducting the charges for other repayments of loans is then calculated. If these funds are greater than the cost of establishing a business, the entrepreneur will allocate funds to set up a business. Consequently, the amount of savings in the entrepreneur’s account is updated. Otherwise, depending on the size of the business, the entrepreneur can apply for bank credit (*Bank firm creation funding* (M.41.1.) & *New firm creation* (M.41.1.1)) or look for co-shareholders (*Shares* (M.41.2)).

### Module 7: Firm Demography Bank Firm Creation Funding (M.41.1)

The entrepreneur seeks a bank which is both willing to lend funds and that offers a lower interest rate than other banks ($i_{l^{I}}$). In addition, the sectoral exposure requirements are checked ${\left(CostNew\right)}_{temp}^{firm}\leq {\gamma \;}_{sect.exp}$). If the selected bank has the requested funds (${S_{t}^{l.I^{est}}\geq \left(CostNew\right)}_{temp}^{firm}$), creditworthiness is checked. In order to obtain the loan, the following criteria are to be fulfilled:(1)the leverage ratio needs to be acceptable:
(145)$$\begin{array}{c} \frac{{\left(CostNew\right)}_{temp}^{firm}}{K_{init}^{firm}}<{\gamma \;}_{LR.b}, \end{array}$$
where $K_{init}^{firm}\;$ is a sum of funds (deposits) that the owners or shareholders provided to fund the capital of the company.(2)the financial risk of the business has to be relatively low in comparison to the sectorial risk
(146)$$\begin{array}{c} ({\left(Risk\right)}_{t}^{firm}<{\gamma \;}_{Risk.b\;}\&\&{\;(Risk)}_{t}^{firm}<\;{\bar{\left(Risk\right)}}_{t}^{s}), \end{array}$$(3)the size of the company has to be at least average for the industry ($L_{t}^{firm}/{\bar{L}}_{t}^{s}$) and credit history has to be good (${\left(PD\right)}_{t}^{firm}<{\gamma \;}_{cred.h.b}$).

Banks may set their own thresholds for the proposed criteria. In this way banks compete not only on interest rates but also creditworthiness criteria. Thresholds are an expression of their degree of risk aversion and the adopted strategy of a bank’s activity. If the trader does not get the requested amount in a given bank, they try again to apply for a loan in another bank, starting with the bank that offers the lowest interest rate. In extreme cases, the amount of funding obtained from the bank is reduced, gradually up to the limit below which it would not be possible to open a business in the sector. In the *New firm creation*(41.1.1) sub-module, records are created for the new firm, and the firm loans of the individual are updated. Maturity of the loan ($M^{I^{ind}}$) is assigned depending on the amount of loan taken.
(147)$$\begin{array}{c} l_{t}^{I^{ind}}+={\left(CostNew\right)}_{temp}^{firm} \end{array}$$
(148)$$\begin{array}{c} {\left(pl\right)}_{t}^{I^{ind}}+={\left(CostNew\right)}_{temp}^{firm} \end{array}$$
(149)$$\begin{array}{c} l_{t}^{I_{q}^{ind}}+=\frac{{\left(CostNew\right)}_{temp}^{firm}}{M^{I^{ind}}} \end{array}$$
(150)$$\begin{array}{c} R_{t}^{I^{ind}}+=\frac{1}{{\kappa \;}_{4}}\times {({\left(CostNew\right)}_{temp}^{firm}\times \left(1+0.25\times i_{l^{I}}\right)}^{M^{I^{ind}}}-{\left(CostNew\right)}_{temp}^{firm}) \end{array}$$
(151)$$\begin{array}{c} R_{t}^{I_{q}^{ind}}+=\frac{R_{t}^{I^{ind}}}{M^{I^{ind}}} \end{array}$$
The bank’s revenues are updated as well.
(152)$$\begin{array}{c} {\left({Rev}_{l^{I}}\right)}_{t}^{b}+=R_{t}^{I_{q}^{ind}}. \end{array}$$

### Module 7: Firm Demography Shares (M.41.2)

For large companies, their initial capital is the sum of the contributions of the individuals. The share of each individual is determined by the consumer type of the household to which this individual belongs and the deposits. The algorithm looks for *M* individuals which shares will fund the cost of setting up the firm (${\left(CostNew\right)}_{temp}^{firm}$). For all individuals we update the values of shares ($\psi_{t}^{ind}$) and deposits ($d_{t}^{ind}$). Large companies can apply for short-term loans but are funded by their shares. In the case of short-term loans, mechanisms are analogous to the one presented for small and medium-sized businesses but different credit ratings are adopted.

### Module 8: Mergers & Acquisitions Updating Firm Market Value (M.42)

The *Mergers & Acquisitions* module analyses the possibility of one company buying another. First, the value of the company on the market (${\left(MV\right)}_{t}^{firm}$) is updated, depending on the company’s generated profits relative to average firm profits in the sector (${\left(RelProf\right)}_{temp}^{firm}=\Pi_{t}^{firm}/({\bar{\pi })}_{t}^{s})$ and the company’s financial risk relative to the average risk in the industry (${\left(RelRisk\right)}_{tem}^{firm}={\left(Risk\right)}_{t}^{firm}/{\bar{\left(Risk\right)}}_{t}^{s}$). The market value of a company is correlated with the likelihood of bankruptcy of a particular company. By default, the maximum value of this variable is 1. The closer this value is to 1, the higher the probability of bankruptcy. If the variable is close to 1, then the probability of the business declaring insolvency increases.

### Module 8: Mergers & Acquisitions Mergers (M.43)

If the business debt (${\left(Debt\right)}_{t}^{firm}$) is lower than that specified by the parameter ${\gamma \;}_{\pi }$ part of the firm’s profits $\left(\Pi_{t}^{firm}\right)$, the firm was not marked as for sale (${\left(ForSale\right)}_{t}^{firm}=0$), the firm debt is lower than that determined by the parameter ${\gamma \;}_{LA\_K\;}$ part of liquid assets (${\left(LA\right)}_{t}^{firm}$) and capital ($K_{t}^{firm}$), the firm value $\left({\left(MV\right)}_{t}^{firm}\right)$ is greater than that specified by the parameter ${\gamma \;}_{MV\;}$, a percentage ${\gamma \;}_{merg1}$ of firms in the sector is searched for. The company will merge with one of these firms. The company to be merged with must meet several requirements. First, the sum of liquid assets of the firm and the fixed capital of this firm need to be lower than that of the second company.
(153)$$\begin{array}{c} {\left((LA\right)}_{t}^{merged.firm}+K_{t}^{merged.firm})<{\left((LA\right)}_{t}^{firm}+K_{t}^{firm}). \end{array}$$
Secondly, it should be a smaller company, in a sense that the work force of this firm is smaller than the work force of the second company ($L_{t}^{merged.firm}<L_{t}^{firm}$). Next, the market value of the firm should be acceptable (${\left(MV\right)}_{t}^{merged.firm}>{{\gamma \;}_{MV}\times \left(MV\right)}_{t}^{firm}$) and this firm should not be marked for sale (${\left(ForSale\right)}_{t}^{firm}=0$). If the merger occurs, then the attributes of the firm and the entrepreneur are updated.
(154)$$\begin{array}{c} {\left(Debt\right)}_{t}^{firm}+={\left(Debt\right)}_{t}^{merged.firm}+{\left(MV\right)}_{t}^{merged.firm}\times {\left((LA\right)}_{t}^{merged.firm}+K_{t}^{merged.firm}) \end{array}$$
(155)$$\begin{array}{c} {\left(N_{est}\right)}_{t}^{firm}+={\left(N_{est}\right)}_{t}^{merged.firm} \end{array}$$
(156)$$\begin{array}{c} {\left((LA\right)}_{t}^{firm}+={\left((LA\right)}_{t}^{firm} \end{array}$$
(157)$$\begin{array}{c} K_{t}^{firm}+=K_{t}^{firm}. \end{array}$$
The owner of the firm changes and the corresponding variables such as labor status ($\;\Xi_{t}^{ind}=5)$, entrepreneurship (${\left(Entr\right)}_{t}^{ind}=1$), entrepreneurship in the past (${\left(EducP\right)}_{t}^{ind}=1$), periods since the last change on the labor market ($\Psi_{t}^{ind}=0$) and the deposits are updated as well.
(158)$$\begin{array}{c} d_{t}^{owner.ind}={\gamma \;}_{53}\times {\left(MV\right)}_{t}^{merged.firm}\times {\left((LA\right)}_{t}^{merged.firm}+K_{t}^{merged.firm}). \end{array}$$
If the firm wants to merge but there is no company to merge within the set ${\gamma \;}_{merg1}$ of firms, the procedure is repeated in the broader set specified by ${\gamma \;}_{merg2}.$

### Module 8: Mergers & Acquisitions Acquisitions (M.44)

If a company is successful and its financial risk is below average risk in the industry (${\left(Risk\right)}_{t}^{firm}\;\leq \;{\bar{\left(Risk\right)}}_{t}^{s}$) and the firm is not marked for sale (${\left(ForSale\right)}_{t}^{firm}=0$), then it seeks to acquire another firm. In practice, this company should be relatively small ($L_{t}^{est}<{\gamma \;}_{55}\times L_{t}^{est}$), debt should be lower than the one expressed by debt percentage of the firm before acquisition (${\left(Debt\right)}_{t}^{firm}<{{\gamma \;}_{56}\times \left(Debt\right)}_{t}^{firm}$), and this company should be marked for sale (${\left(ForSale\right)}_{t}^{firm}=1$). If a company to acquire is found, then the attributes of the firms and the entrepreneur are updated. The entrepreneur is now the co-owner of the firm.
(159)$$\begin{array}{ll} {\left(Debt\right)}_{t}^{firm} & +={\left(Debt\right)}_{t}^{acq.firm}+{\left(MV\right)}_{t}^{acq.firm}\times ({\left(LA\right)}_{t}^{acq.firm}+K_{t}^{acq.firm}+\\ & -{\left(Debt\right)}_{t}^{acq.firm})\times \sqrt{rndm(0,1)} \end{array}$$
(160)$$\begin{array}{c} K_{t}^{firm}+=K_{t}^{acq.firm} \end{array}$$
(161)$$\begin{array}{c} {\left(N_{est}\right)}_{t}^{firm}+={\left(N_{est}\right)}_{t}^{acq.firm} \end{array}$$
(162)$$\begin{array}{c} {\left(LA\right)}_{t}^{firm}+={\left(LA\right)}_{t}^{acq.firm} \end{array}$$
(163)$$\begin{array}{c} L_{t}^{firm}+=L_{t}^{acq.firm}. \end{array}$$

### Module 9: Firm Growth Individuals Records Updating When Firm Closes (M.45)

This module deals with the opening and closure of companies. If the company was registered more than 1.5 years ago (${\left(\#Oper\right)}_{t}^{firm}>6$) and the number of employees is 0, then the company is removed from the system. If the entrepreneur does not own other firms, the value of variables which describe the experience of being an entrepreneur (${\left(Entr\right)}_{t}^{ind}=0,\;({EntrP)}_{t}^{ind}=1$), status on the labour market ($\Xi_{t}^{ind}\neq 5$) and the period since the last change of status on the labour market ($\Psi_{t}^{ind}=0$) are updated. If the company qualifies for closure, all employees will be first fired, and then the establishments will be closed.

### Module 9: Firm Growth New Establishment Creation (M.46)

In this sub-module, firms with a strong position can also increase the number of establishments they own. If the company’s financial risk is lower than the average risk in the sector (${\left(Risk\right)}_{t}^{firm}<{\bar{\left(Risk\right)}}_{t}^{s}$) and the profit per employee $\frac{\Pi_{t}^{firm}}{{(L}_{t}^{firm}+1)}$ is higher than the percentage of the average profit per employee in the sector $\upsilon_{2}\times \frac{{\bar{\pi }}_{t}^{s}}{\left(N_{t}^{f{irm}_{s}}\times \left({\bar{L}}_{t}^{s}+1\right)\right)}$, then the company can create new establishments.

### Module 10: Labour Market Updating Individual Labour Status (M.47)

Within the framework of the model, a stylized labour market has been designed. In the model the parameters $\alpha_{57.age.gender-62.age.gender}\;$ depend on age and gender. In the module the status on the labour market of all individuals ($\Xi_{t}^{ind}$) and the time since the last change of status on the labor market ($\Psi_{t}^{ind}$) are updated. If the individual is a woman over the age of 55, employed in the private or public sector, she is likely to be unemployed with the probability $\alpha_{57.age}.$ If the individual is a man over the age of 55, employed in the private or public sector, he is likely to be unemployed with the probability $\alpha_{58.age}$. If the individual is a woman, who is unemployed for more than 2 years, then she becomes inactive with probability $\alpha_{59.age}.$ If the individual is a man who is unemployed for more than 2 years, then he becomes inactive with probability $\alpha_{62.age}.\;$ If a person is inactive on the labour market and is under the age of 55 and their total family income is less than the established threshold, then this person will be registered as an active job seeker, even if only for the purpose of receiving social benefits.

### Module 10: Labour Market Workers Skills (M.48)

This sub-module updates the variable describing employee skills. These skills depend on age, gender, time elapsed since the last change of status on the labour market and completed education.
(164)$$\begin{array}{c} \Theta_{t}^{ind}=max\left\{0;{\gamma \;}_{\Psi }\times \Psi_{t}^{ind}+{{\gamma \;}_{Educ}\times \left(Educ\right)}_{t}^{ind}+{\gamma \;}_{G}\times G^{ind}{+\gamma \;}_{\Xi }\times \Xi_{t}^{ind}{+\gamma \;}_{Age}\times {\left(Age\right)}_{t}^{ind}\right\}. \end{array}$$

### Module 10: Labour Market Hiring & Firing in Establishments (M.49)

This sub-module carries one from the *Firm growth* module. In this sub-module a decision is made to reduce or increase the number of employees. First, the number of employees to be fired ($L_{t}^{{FI}^{est}}$) and hired in the establishment ($L_{t}^{{HI}^{est}}$) is compared. Employees with the lowest working skills are fired. If the number of employees to be hired is greater than the number of employees to be fired, then employees are searched for according to their location (using NUTS1-4 codes: ${\vartheta \;}_{t}^{{ind}^{1}},\;{\vartheta \;}_{t}^{{ind}^{2}},\;{\vartheta \;}_{t}^{{ind}^{3}},\;{\vartheta \;}_{t}^{{ind}^{4}}$). The model assumes that employees living closer to the firm are beneficial to the company.

### Module 11: Cycle Ends

The final module updates the remaining agent records that have not been updated in previous modules. Loans and deposits of individuals, households, companies and, above all, banks are settled. Non-performing loans in the period are also counted.

### Module 11: Cycle Ends Labor Status & Wage & Subsidy Updating (M.50)

Individuals from the *Labor market* modules (M.47–M.49) are carried on into this sub-module. If individuals have just become unemployed ($\Xi_{t}^{ind}=2\;\&\&\;\Psi_{t}^{ind}=0$), they receive unemployment benefits (${{\left(sub\right)}_{t}^{ind}=\delta \;}_{1}$) and their earnings from previous employment ($w_{t}^{ind}=0$) are equal to zero. If individuals have become unemployed more than one quarter ago ($\Xi_{t}^{ind}=2\;\&\&\;\Psi_{t}^{ind}>1$), then they receive lower benefits (${{\left(sub\right)}_{t}^{ind}=\delta \;}_{2},\;{\delta \;}_{2}<{\delta \;}_{1}$). If a person is employed in the private sector ($\Xi_{t}^{ind}=3$), then no social benefits are to be paid (${\left(sub\right)}_{t}^{ind}=0$) and the earnings are calculated according to the formula that depends on age (${\left(Age\right)}_{t}^{ind}$), gender ($G^{ind}$), average wage per employee in the sector $\frac{{\bar{W}}_{t}^{s}}{{\bar{L}}_{t}^{s}}$ and a random factor.
(165)$$\begin{array}{c} w_{t}^{ind}=\frac{{\bar{W}}_{t}^{s}}{{\bar{L}}_{t}^{s}}\times \left({\delta \;}_{6}+{\delta \;}_{7}\times G^{ind}\right)\times \left({\delta \;}_{8}+{\delta \;}_{9}\times {\left(Age\right)}_{t}^{ind}\right)\times \left(rndm\left(0.04,0.06\right)\right). \end{array}$$
If the person is 67 years old and the subsidy was equal to 0, then the subsidy depending on average wage in the sector, gender and values of parameters is computed. If the person is inactive, then the gender and age of the individual are checked. If individuals are older than 55 ($G^{ind}=1\;\&\&\;{\left(Age\right)}_{t}^{ind}>\;55$), and had positive wage ($w_{t}^{ind}>0$), they may retire early. The subsidy depending on gender and previous wage is computed, and the wage is set to zero ($w_{t}^{ind}=0,\;{{\left(sub\right)}_{t}^{ind}=\delta \;}_{2}\times w_{t}^{ind}\;$ for women and ${{\left(sub\right)}_{t}^{ind}=\delta \;}_{2}\times w_{t}^{ind}$ for man). If the person in question runs a business and the profits of the firm are positive ($\Pi_{t}^{firm}>0$), then for each of the companies they run, the portion of the profit paid to dividends is calculated; this is equal to the remuneration of the owner(s).
(166)$$\begin{array}{c} {\left(Divid\right)}_{temp}+=\Pi_{t}^{firm}\times {\delta \;}_{14}\times rndm\left(0,0.02\right) \end{array}$$
(167)$$\begin{array}{c} w_{t}^{ind}+={\left(Divid\right)}_{temp} \end{array}$$
(168)$$\begin{array}{c} {\left(sub\right)}_{t}^{ind}=0 \end{array}$$
(169)$$\begin{array}{c} {\left(Debt\right)}_{t}^{firm}+={\left(Divid\right)}_{temp}. \end{array}$$
If the total family income is below the subsistence level (below ${\delta \;}_{3}$), the public assistance is provided (${\left(sub\right)}_{t}^{ind}={\delta \;}_{4}$)

### Module 11: Cycle Ends Establishments & Firms Records Updating (M.51) & (M.52)

The submodules *Updating establishment & firm records* first change the status of new establishments to experienced ones (${\left(New\right)}_{t}^{est}--$) and wages of these establishments are rested from the liquidity assets (${\left(LA\right)}_{t}^{est}-=W_{t}^{est}$) and then set to zero ($W_{t}^{est}=0$). Then we compute the cost of wages ($W_{t}^{est}$+=$w_{t}^{ind}$) and the total costs of the establishments.
(170)$$\begin{array}{c} {\left(TC\right)}_{t}^{est}+={{\delta \;}_{16}\times \left(Inp\right)}_{t}^{est}+{\delta \;}_{16}\times \left({\left(Debt\right)}_{t}^{firm}/{\left(\#Oper\right)}_{t}^{firm}\right)+t_{pr}+{\delta \;}_{18.s}\times Y_{t}^{est}+{\delta \;}_{19.s}\times L_{t}^{est}+{\delta \;}_{20}\times W_{t}^{est} \end{array}$$
Then the costs are paid from the liquid assets. If the liquid assets of the firm are negative (${\left(LA\right)}_{t}^{est}<0$), the establishment has liquidity shortage. We first check whether the firm is able to move sources from one establishment to the other one. If the firm owns more than one establishment and the liquidity of the firm and establishment are positive (${\left(N_{est}\right)}_{t}^{firm}>1\;\&\&\;{(\left(LA\right)}_{t}^{est}+{\left(LA\right)}_{t}^{firm})>0$), one of the establishments is closed (${\left(N_{est}\right)}_{t}^{firm}--$) and liquidity debts of this establishment are paid (${\left(LA\right)}_{t}^{firm}-={\left(LA\right)}_{t}^{est}$), and the value of default is updated ($({PD)}_{t}^{est}++\;$). The firm is at risk of default, but still has not defaulted (${\left(PD\right)}_{t}^{firm}$ increases). Otherwise, if the firm consists of only one establishment that has serious liquidity shortage, the firm defaults ${\left((PD\right)}_{t}^{firm}=1$). The non-performing loans are updated (${\left({NPL}_{l^{I}}\right)}_{t}^{b}+={\left(pl\right)}_{t}^{I^{ind}}$). If the owner of this firm does not own any other firm, then the status on the labor market ($\Xi_{t}^{ind}=2$), the periods since the last change of the status on the labor market ($\Psi_{t}^{ind}=0$) and the variable expressing experience in being an entrepreneur (${\left(Entr\right)}_{t}^{ind}=0$, $({EntrP)}_{t}^{ind}=1$) are updated. In this sub-module, the sales per employee are computed (${\left(SE\right)}_{t}^{est}=\frac{{\left(SE\right)}_{t}^{est}}{L_{t}^{est}}$) and the liquidity assets and fixed capital are updated according to generated profits (or loses). If the profits per employee are sufficiently high (i.e., if $\frac{\Pi_{t}^{firm}}{\left(L_{t}^{firm}+1\right)}>{\delta \;}_{21}$), then the liquid assets and the fixed capital are updated.
(171)$$\begin{array}{c} {\left(LA\right)}_{t}^{firm}+=\left({\delta \;}_{22}+{\delta \;}_{23}\right)\times K_{t}^{firm} \end{array}$$
(172)$$\begin{array}{c} K_{t}^{firm}=\left({1+\delta \;}_{22}{-\delta \;}_{23}\right)\times K_{t}^{firm} \end{array}$$
Otherwise, only fixed capital is updated.
(173)$$\begin{array}{c} K_{t}^{firm}=\left(1{-\delta \;}_{23}\right)\times K_{t}^{firm} \end{array}$$
If the liquid assets per employee are sufficiently high (i.e., if $\frac{{\left(LA\right)}_{t}^{firm}}{\left(L_{t}^{firm}+1\right)>{\delta \;}_{25}}$), then part of liquid assets can be used to pay back the debts. If the firm’s debt is higher than the liquid assets, then the firm’s debt and liquid assets are updated.
(174)$$\begin{array}{c} {\left(Debt\right)}_{t}^{firm}-={\left(LA\right)}_{t}^{firm}-\left(L_{t}^{firm}+1\right)\times {\delta \;}_{25} \end{array}$$
(175)$$\begin{array}{c} {\left(LA\right)}_{t}^{firm}=\left(L_{t}^{firm}+1\right)\times {\delta \;}_{25} \end{array}$$
Otherwise, the debt is paid in full.
(176)$$\begin{array}{c} {\left(LA\right)}_{t}^{firm}-={\left(Debt\right)}_{t}^{firm} \end{array}$$
(177)$$\begin{array}{c} {\left(Debt\right)}_{t}^{firm}=0. \end{array}$$
In the worst case scenario, the firm is accumulating debt and the corresponding variables are updated.
(178)$$\begin{array}{c} {\left(Debt\right)}_{t}^{firm}+=\left(L_{t}^{firm}+1\right)\times {\delta \;}_{26}-{\left(LA\right)}_{t}^{firm} \end{array}$$
(179)$$\begin{array}{c} {\left(LA\right)}_{t}^{firm}=\left(L_{t}^{firm}+1\right)\times {\delta \;}_{26}. \end{array}$$
At the end of this sub-module, lump sum property tax is updated ($t_{pr}$). In the *Updating firm records* sub-module, the age of the firm (${\left(\#Oper\right)}_{t}^{firm}++$) and the financial risk are updated.
(180)$$\begin{array}{c} {\left(Risk\right)}_{t}^{firm}={\delta \;}_{29.s}+{\delta \;}_{30}\times \left(\frac{1}{L_{t}^{firm}+1}\right)+{\delta \;}_{30}\times \left(\frac{L_{t}^{firm}}{{\left(Debt\right)}_{t}^{firm}}\right). \end{array}$$
The values of a firm’s fixed capital, liquidity assets, work force, firm profit and tax to be paid are the sum of respective values of variables of firms’ establishments.

### Module 11: Cycle Ends Updating Industries (Sectors) (M.53)

This module recalculates average values for sectors that become benchmarks for agents’ decisions. The values of the quality and average price of each sector are updated (${{\left(Ql\right)}_{t-1}^{s}=\left(Ql\right)}_{t}^{s},\;P_{t-1}^{s}=P_{t}^{s}$). Then, the profit in the sector and number of firms in the sector are computed as a sum of firms’ profits and the sum of firms. The average financial risk is computed as well. Then, the values of average sales in the sector, price in the sector, quality, average work force of the establishments in the sector, percentage of establishments that import and export as well as average wage in the sector are updated.

### Module 11: Cycle Ends Paying Back of Loans (M.54)

This submodule computes how much has already been paid back from the loan, how much is due and how much interest was paid and is due. This is performed for all types of credit: consumer loans ($l_{t}^{C^{ind}}-=l_{t}^{C_{q}^{ind}}$, ${\left(pl\right)}_{t}^{C^{ind}}-=l_{t}^{C_{q}^{ind}}$, $R_{t}^{C^{ind}}-=R_{t}^{C_{q}^{ind}}$), real estate housing ($l_{t}^{H^{ind}}-=l_{t}^{H_{q}^{ind}}$, ${\left(pl\right)}_{t}^{H^{ind}}-=l_{t}^{H_{q}^{ind}}$, $R_{t}^{H^{ind}}-=R_{t}^{H_{q}^{ind}}$) and non-housing loans ($l_{t}^{{NH}^{ind}}-=l_{t}^{NH_{q}^{ind}},\;{\left(pl\right)}_{t}^{{NH}^{ind}}-=l_{t}^{NH_{q}^{ind}}$, $R_{t}^{{NH}^{ind}}-=R_{t}^{{NH}_{q}^{ind}}$), and firm investment loans ($l_{t}^{I^{ind}}-=l_{t}^{I_{q}^{ind}}$, ${\left(pl\right)}_{t}^{I^{ind}}-=l_{t}^{I_{q}^{ind}}$, $R_{t}^{I^{ind}}-=R_{t}^{I_{q}^{ind}}$). The same is computed for establishments paying back short-term loans ($l_{t}^{{SH}^{est}}-=l_{t}^{SH_{q}^{est}},\;{\left(pl\right)}_{t}^{{SH}^{est}}-=l_{t}^{SH_{q}^{est}},\;R_{t}^{{SH}^{est}}-=R_{t}^{{SH}_{q}^{est}}$).

### Module 11: Cycle Ends Bank’s Balance Sheet Positions With Non-Financial Sector Updating (M.55)

This submodule computes how much supply was used to give new credits in this iteration. For this purpose, the value of supply from the *Initialisation module* (M.1) and the value of supply after all updates (module M.55) is used. The amount of loans of each type which have been granted is computed.
(181)$$\begin{array}{c} l_{temp}^{C.granted.b}=S_{temp}^{{l.C}^{ind}}-S_{t}^{{l.C}^{ind}} \end{array}$$
(182)$$\begin{array}{c} l_{temp}^{H.granted.b}=S_{temp}^{{l.H}^{ind}}-S_{t}^{{l.H}^{ind}} \end{array}$$
(183)$$\begin{array}{c} l_{temp}^{NH.granted.b}=S_{temp}^{{l.NH}^{ind}}-S_{t}^{{NH}^{ind}} \end{array}$$
(184)$$\begin{array}{c} l_{temp}^{I.granted.b}=S_{temp}^{{l.I}^{ind}}-S_{t}^{l.I^{est}} \end{array}$$
(185)$$\begin{array}{c} l_{temp}^{SH.granted.b}=S_{temp}^{{l.SH}^{ind}}-S_{t}^{l.{SH}^{est}} \end{array}$$
(186)$$\begin{array}{c} l_{temp}^{sum.granted.b}=l_{temp}^{C.granted.b}+l_{temp}^{H.granted.b}+l_{temp}^{NH.granted.b}+l_{temp}^{I.granted.b}+l_{temp}^{SH.granted.b} \end{array}$$
(187)$$\begin{array}{c} (p{l)}_{temp}^{C.granted}+={\left(pl\right)}_{t}^{C^{ind}} \end{array}$$
(188)$$\begin{array}{c} (p{l)}_{temp}^{H.granted}+={\left(pl\right)}_{t}^{H^{ind}} \end{array}$$
(189)$$\begin{array}{c} (p{l)}_{temp}^{NH.granted}+={\left(pl\right)}_{t}^{{NH}^{ind}} \end{array}$$
(190)$$\begin{array}{c} (p{l)}_{temp}^{I.granted}+={\left(pl\right)}_{t}^{I^{ind}} \end{array}$$
(191)$$\begin{array}{c} (p{l)}_{temp}^{I.granted}+={\left(pl\right)}_{t}^{{SH}^{est}} \end{array}$$
(192)$$\begin{array}{c} {\left(pl\right)}_{temp}^{sum.granted}=(p{l)}_{temp}^{C.granted}+(p{l)}_{temp}^{H.granted}+(p{l)}_{temp}^{NH.granted}+(p{l)}_{temp}^{I.granted}+(p{l)}_{temp}^{SH.granted} \end{array}$$
(193)$$\begin{array}{c} {\left(NPL\right)}_{temp}^{sum}={\left({NPL}_{l^{C}}\right)}_{t}^{b}+{\left({NPL}_{l^{H}}\right)}_{t}^{b}+{\left({NPL}_{l^{NH}}\right)}_{t}^{b}+{\left({NPL}_{l^{I}}\right)}_{t}^{b}+{\left({NPL}_{l^{SH}}\right)}_{t}^{b} \end{array}$$
(194)$$\begin{array}{c} {\left(NPL\right)}_{temp}^{ratio}=\frac{{\left(NPL\right)}_{temp}^{sum}}{{\left(pl\right)}_{temp}^{sum.granted}} \end{array}$$
(195)$$\begin{array}{c} {\left(NPL\right)}_{temp}^{ratio.l^{C}}=\frac{{\left({NPL}_{l^{C}}\right)}_{t}^{b}}{{\left(pl\right)}_{temp}^{sum.granted}} \end{array}$$
(196)$$\begin{array}{c} {\left(NPL\right)}_{temp}^{ratio.l^{NH}}=\frac{({\left({NPL}_{l^{H}}\right)}_{t}^{b}}{{\left(pl\right)}_{temp}^{sum.granted}} \end{array}$$
(197)$$\begin{array}{c} {\left(NPL\right)}_{temp}^{ratio.l^{NH}}=\frac{{\left({NPL}_{l^{NH}}\right)}_{t}^{b}}{{\left(pl\right)}_{temp}^{sum.granted}} \end{array}$$
(198)$$\begin{array}{c} {\left(NPL\right)}_{temp}^{ratio.l^{I}}=\frac{{\left({NPL}_{l^{I}}\right)}_{t}^{b}}{{\left(pl\right)}_{temp}^{sum.granted}} \end{array}$$
(199)$$\begin{array}{c} {\left(NPL\right)}_{temp}^{ratio.l^{SH}}=\frac{{\left({NPL}_{l^{SH}}\right)}_{t}^{b}}{{\left(pl\right)}_{temp}^{sum.granted}} \end{array}$$
(200)$$\begin{array}{c} d_{temp}^{ind}+=d_{t}^{ind} \end{array}$$
(201)$$\begin{array}{c} d_{temp}^{firm}+={\left(LA\right)}_{t}^{est} \end{array}$$
(202)$$\begin{array}{c} d_{t}^{b}=d_{temp}^{ind}+d_{temp}^{firm}. \end{array}$$
Equity is updated in the next module, after computing the profits and costs of banks’ activity.

### Module 11: Cycle Ends Updating Banks’ Profits & Costs & Equity (M.56)

This submodule updates the equity by adding the profits generated in this period and subtracting costs associated with the banks’ activity.
(203)$$\begin{array}{c} {\left({Rev}^{b}\right)}_{temp}^{sum}=\;{\left({Rev}_{l^{C}}\right)}_{t}^{b}+{\left({Rev}_{l^{H}}\right)}_{t}^{b}+{\left({Rev}_{l^{NH}}\right)}_{t}^{b}+{\left({Rev}_{l^{I}}\right)}_{t}^{b}+{\left({Rev}_{l^{SH}}\right)}_{t}^{b} \end{array}$$
(204)$$\begin{array}{c} {\left(Cost\right)}_{temp}^{b}\;draw\;from\;N\left({\left(AvC\right)}_{t}^{b},{\left(SdC\right)}_{t}^{b}\right) \end{array}$$
(205)$$\begin{array}{c} E_{t}^{b}+={\left({Rev}^{b}\right)}_{temp}^{sum}-{\left(Cost\right)}_{temp}^{b}-i_{d}\times d_{t}^{b} \end{array}$$
Also considered is the possibility of the banks’ default. If the following value of temporal variable is below zero, the bank defaults. In practice, the bank can default when it has not enough equity or in the case of withdrawal of deposits or accumulation of non-performing loans.
(206)$$\begin{array}{c} {\left(Default\right)}_{temp}^{b}=E_{t}^{b}+d_{t}^{b}+{\left({Rev}^{b}\right)}_{temp}^{sum}-i_{d}\times d_{t}^{b}-{\left(NPL\right)}_{temp}^{sum}. \end{array}$$

### Module 11: Cycle Ends Supply Side Decisions for T+1 and Regulatory Requirements (M.57)

This submodule analyses the regulatory requirements and the supply side decisions. Definitions of macroprudential ratios have been adapted from Popoyan et al. (2017). Firstly, the value of most liquid assets held by the bank (cash) as a percentage of all loans granted to the entities in the model is approximated.
(207)$$\begin{array}{c} {\left(Appr.Cash\right)}_{temp}^{b}=\varphi_{1}\times l_{temp}^{sum.granted.b} \end{array}$$
We also compute the quantity of loans less the most liquid assets held.
(208)$$\begin{array}{c} {\Theta \;}_{temp}^{b}={l_{temp}^{sum.granted.b}-\left(Appr.Cash\right)}_{temp}^{b} \end{array}$$
Next, the quantity of liquid assets demanded from the central bank is approximated.
(209)$$\begin{array}{c} {\left(LiqDemCB\right)}_{temp}=\varphi_{LCR.min}\times d_{t}^{b}-B_{t}^{b}-{\left(Res\right)}_{t}^{b}-\varphi_{3}\times {\left(Appr.Cash\right)}_{temp}^{b} \end{array}$$
The liquid assets from the central bank are only demanded when LCR is lower than $\varphi_{LCR.min}$. The computation of risk weighted assets and the approximation of TIER 1 equity allows the value of capital adequacy ratio according to Basel III to be obtained.
(210)$$\begin{array}{c} {\left(RWA\right)}_{temp}={\left(pl\right)}_{temp}^{sum.granted}\times {\delta \;}_{RW} \end{array}$$
(211)$$\begin{array}{c} {\left(E_{T1}^{b}\right)}_{temp}=E_{t}^{b}\times {\delta \;}_{T1} \end{array}$$
(212)$$\begin{array}{c} {\left({CAR}^{b}\right)}_{temp}=\frac{{\left(E_{T1}^{b}\right)}_{temp}}{{\left(RWA\right)}_{temp}} \end{array}$$
The model also computes the value of bank leverage ratio, as a ratio of equity TIER 1 and total assets, as well as high quality liquid assets, expected cash inflow and outflow to be able to compute the liquidity coverage ratio.
(213)$$\begin{array}{c} {\left({LR}^{b}\right)}_{temp}=\frac{{\left(E_{T1}^{b}\right)}_{temp}}{\left(l_{temp}^{sum.granted.b}+B_{t}^{b}+{\left(Res\right)}_{t}^{b}+\varphi_{3}\times {\left(Appr.Cash\right)}_{temp}^{b}\right)} \end{array}$$
(214)$$\begin{array}{c} {\left({HQLA}^{b}\right)}_{temp}={\left(Res\right)}_{t}^{b}+{\left(Appr.Cash\right)}_{temp}^{b}+\min\left\{0.85;0.75\times \left({\left(Res\right)}_{t}^{b}+{\left(Appr.Cash\right)}_{temp}^{b}\right)\right\} \end{array}$$
(215)$$\begin{array}{c} {\left({E(CashOutflow)}^{b}\right)}_{temp}=\varphi_{4}\;+\varphi_{5}\times d_{t}^{b}\;+\varphi_{6}\times {\left(LiqDemCB\right)}_{temp} \end{array}$$
(216)$$\begin{array}{c} {\left({E(CashInflow)}^{b}\right)}_{temp}=\varphi_{7}\;-\varphi_{8}\;+\varphi_{9}\times {\left(Appr.Cash\right)}_{temp}^{b}\;+\varphi_{10}\times B_{t}^{b}\;+\varphi_{11}\times {\left(Res\right)}_{t}^{b} \end{array}$$
(217)$$\begin{array}{c} {\left({NCOF}^{b}\right)}_{temp}={\left({E(CashOutflow)}^{b}\right)}_{temp}-{\left({E(CashInflow)}^{b}\right)}_{temp} \end{array}$$
(218)$$\begin{array}{c} {\left({LCR}^{b}\right)}_{temp}=\frac{{\left({HQLA}^{b}\right)}_{temp}}{{\left({NCOF}^{b}\right)}_{temp}} \end{array}$$
Then, the decisions about the supply of credit are made:(219)$$\begin{array}{c} {\left(S^{b}\right)}_{temp}=\frac{E^{b_{t}}}{\varphi_{min.c.req}}-\;{\left(pl\right)}_{temp}^{sum.granted} \end{array}$$
If the capital requirements are not met, the supply is set to zero (${\left(S^{b}\right)}_{temp}=0$) and the value of variable $\;({\left(PD\right)}_{t}^{b}=0.5)$ is updated to emphasise that this bank is at threat of default and the riskiness of the bank’s activity increases. Next, it is checked if the liquidity requirement is met. If so, the bank does not have to recur to the central bank. Otherwise $\left({\left({LCR}^{b}\right)}_{temp}<\varphi_{min.LCR}\times d_{t}^{b}\right),$ bank deposits are increased by the resources obtained from the central bank.
(220)$$\begin{array}{c} d_{t}^{b}+\;={\left(LiqDemCB\right)}_{temp} \end{array}$$
Finally, the leverage ratio requirement is checked. If the leverage ratio is lower than the value specified by the parameter (${\left({LR}^{b}\right)}_{temp}\leq \varphi_{13}$), the supply is set to zero (${\left(S^{b}\right)}_{temp}=0$) and the value of the variable expressing the probability of default ${(\left(PD\right)}_{t}^{b}=0.5)$ is updated to emphasise that this bank is at threat of default and the riskiness of the bank’s activity increases.

Next, each bank needs to compute the supply of each type of loan to the market, taking into account the profitability of each and the non-performing loans ratios. There are two procedures that can be initiated in the system. The first one is initiated if the values of parameters $\mu_{1},\;\mu_{2},\;\mu_{3},\;\mu_{4},\;\mu_{5}$, are not equal to zero. It is implicitly assumed that the banks are committed to providing a supply of credit of each type and all banks are universal. In the consequence of application of the first strategy, groups of banks that pursue similar strategies on the market are obtained.

At the beginning, the following auxiliary variables are computed:(221)$$\begin{array}{c} ({SumForRatio)}_{temp}^{{l.C}^{ind}}=S_{temp}^{{l.C}^{ind}}+{\left({Rev}_{l^{C}}\right)}_{t}^{b}-{\left({NPL}_{l^{C}}\right)}_{t}^{b} \end{array}$$(222)$$\begin{array}{c} ({SumForRatio)}_{temp}^{{l.H}^{ind}}=S_{temp}^{{l.H}^{ind}}+{\left({Rev}_{l^{H}}\right)}_{t}^{b}-{\left({NPL}_{l^{H}}\right)}_{t}^{b} \end{array}$$(223)$$\begin{array}{c} ({SumForRatio)}_{temp}^{{l.NH}^{ind}}=S_{temp}^{{l.NH}^{ind}}+{\left({Rev}_{l^{NH}}\right)}_{t}^{b}-{\left({NPL}_{l^{NH}}\right)}_{t}^{b} \end{array}$$(224)$$\begin{array}{c} ({SumForRatio)}_{temp}^{{l.I}^{ind}}=S_{temp}^{{l.I}^{ind}}+{\left({Rev}_{l^{I}}\right)}_{t}^{b}-{\left({NPL}_{l^{I}}\right)}_{t}^{b} \end{array}$$(225)$$\begin{array}{c} ({SumForRatio)}_{temp}^{{l.SH}^{est}}=S_{temp}^{{l.SH}^{est}}+{\left({Rev}_{l^{SH}}\right)}_{t}^{b}-{\left({NPL}_{l^{SH}}\right)}_{t}^{b} \end{array}$$
These auxiliary variables help us to understand how much has been earned and lost from the granted loans of each type. Then, the following ratios are computed: (226)$$\begin{array}{c} ({Sum1ForRatio)}_{temp}^{{l.C}^{ind}}=\frac{({SumForRatio)}_{temp}^{{l.C}^{ind}}}{S_{t}^{{l.C}^{ind}}} \end{array}$$(227)$$\begin{array}{c} ({Sum1ForRatio)}_{temp}^{{l.H}^{ind}}=\frac{({SumForRatio)}_{temp}^{{l.H}^{ind}}}{S_{t}^{{l.H}^{ind}}} \end{array}$$(228)$$\begin{array}{c} ({Sum1ForRatio)}_{temp}^{{l.NH}^{ind}}=\frac{({SumForRatio)}_{temp}^{{l.NH}^{ind}}}{S_{t}^{{NH}^{ind}}} \end{array}$$(229)$$\begin{array}{c} ({Sum1ForRatio)}_{temp}^{{l.I}^{ind}}=\frac{({SumForRatio)}_{temp}^{{l.I}^{ind}}}{S_{t}^{l.I^{est}}} \end{array}$$(230)$$\begin{array}{c} ({Sum1ForRatio)}_{temp}^{{l.SH}^{est}}=\frac{({SumForRatio)}_{temp}^{{l.SH}^{est}}}{S_{t}^{l.{SH}^{est}}} \end{array}$$
The ratios are ordered from highest to the lowest. According to the ratios, it is possible to indicate the most profitable and less profitable types of loans. In the first procedure, it is assumed that the regulator introduced the maximum sectorial exposures, that is, $\mu_{1},\;\mu_{2},\;\mu_{3},\;\mu_{4},\;\mu_{5}$, are fixed by the regulator. In practice, banks invest the maximum allowed by the regulator in the most profitable type of loans according to them. In the second procedure, it is assumed that the regulator introduced only one recommendation; that the exposure to any loan cannot be higher than 35%, that is $\mu_{i}\;\leq$ 0.35 for i∈{1,2,3,4,5}. The auxiliary assumption needed states that all weights have to sum up to 1: $\mu_{1}+\;\mu_{2}+\;\mu_{3}+\;\mu_{4}+\;\mu_{5}=1.$

In this method, $\mu_{1}$ corresponds always to the first ratio, $\mu_{2}$ corresponds always to the second ratio, $\mu_{3}$ corresponds always to the third ratio, $\mu_{4}$ corresponds always to the fourth ratio and, $\mu_{5}$ corresponds always to the fifth ratio. The highest ratio is checked, and then the value of $\mu_{i}$, is drawn from the corresponding interval of the highest allowed values, $\mu_{i}\in$ (0:30; 0:35]. Then, the second best, $\mu_{j}$ is drawn from the interval $\mu_{j}\;\in \;$ ($\mu_{i}-0.06$; $\mu_{i}$). Analogically, the third and the fourth best, ie. $\mu_{k}$ is drawn from the interval $\mu_{k}\in \;\left(\mu_{j}-0.06;\;\mu_{j}\right),\;\mu_{l}\in \;\left(\mu_{k}-0.06;\;\mu_{k}\right).$ The fifth one is computed from the restriction that states that all weights have to sum up to one. In this procedure, each bank has a different ordering of ratios and in addition weights are obtained in a stochastic procedure.

### Module 11: Cycle Ends Interest Rates (M.58)

This sub-module computes the value of interest rates for next iteration. If the values of perception indicators are equal to zero then the same empirical interest rates from the database in the cycle can be used. Otherwise, the indicators of perception to change the values of interest rates in the first iteration are used and then random values of perception indicators from the distribution are used. The perception of risk is different on the ON (${\varsigma \;}^{{ind}^{1}}$) and longer-maturity markets (SW, 1M, 3M) (respectively ${\varsigma \;}^{{ind}^{2}}$, ${\varsigma \;}^{{ind}^{3}}$,${\varsigma \;}^{{ind}^{4}}$). The parameter ${\zeta \;}_{t}^{ind}$ expresses the value of perception of uncertainty indicator.
(231)$$\begin{array}{c} \;{\varsigma \;}_{t}^{{ind}^{1}}=0\;\&\&\;{\varsigma \;}_{t}^{{ind}^{2}}=0\;\&\&\;{\varsigma \;}_{t}^{{ind}^{3}}=0\;\&\&\;{\varsigma \;}_{t}^{{ind}^{4}}=0\;\&\&\;{{\zeta \;}_{t}}^{ind}=0. \end{array}$$

## 5. Inequality Measures and Distributional Effects

### 5.1. Income Distribution, Inequality and Concentration Measures

At t=0, the simulation uses empirical data. In Figure 2 the household income according to the percentile has been presented as well as the approximation of wealth has been shown.

Based on empirical data, it is possible to calculate the approximation of the degree of income and wealth inequality in the sample at t=0, which has been compared with results of simulations of wealth and income distributions in counterfactual scenarios (at t>0).

### 5.2. Gini Coefficient and Measures of Asymmetry

Typically, the degree of inequality is calculated based on the Gini coefficient [105,106], and the Lorenz curve [107]. This section presents the results of the calculation of the basic Gini coefficient and the further studies on income and wealth inequality. Based on the sample, both the Lorenz curve and the income Gini coefficient were determined from the data on a quarterly and annual basis. The Gini coefficient for income distribution calculated on the basis of the quarterly sample data is equal to 0.3943179 (39.43%). In the case of calculations on annual figures, the coefficient value is slightly different, 0.3945987 (39.46%). If we include the corrections for unbiasedness, the value is 0.3947129 (39.37%).

These values differ slightly from the Gini coefficient for income distributions reported in the 2008 Report on Household Wealth and Debt, 39.4% versus 38.4%. This is due to the adoption of a different procedure of over-sampling of most affluent households. In addition, Grejcz and Żołkiewski [108] present the coefficient as 39.2%. At the same time, the authors emphasise that the procedure of oversampling of the most affluent households makes the value of Gini coefficient higher in comparison to the ones published by OECD, 35.5% or World Bank in 2014, 32.08%. In literature, other values of Gini coefficients for income distributions, computed using different samples, have been presented: 32.6% [109], 30.7% [110], 28.5% [111]. Accordingly, the Gini coefficient for wealth is equal to 0.6004731 (60.04%). The obtained results are relatively similar to the ones presented in the NBP [112] report 57.9%, and Grejcz and Żołkiewski [108] 58.7%.

According to Eurostat, Gini coefficient for income distribution in Poland in 2014 and 2015, were respectively 30.8% and 30.6%. The average Gini coefficient in the European Union for 28 countries was 30.9% in 2014 and 31% in 2015. A similar Gini coefficient for income distribution, suggesting a similar level of inequality, was reported in countries such as Croatia, 30.2% in 2014 and 2015, Ireland, 31.1% in 2014 and 29.8% in 2015, Germany, 30.7% in 2014 and 30.1% in 2015, and Great Britain, 31.6% in 2014 and 32.4% in 2015. The reported value of Gini coefficient of wealth for euro area by Grejcz and Żołkiewski [108] was 68.6%. In the case of the Gini coefficient for wealth distribution, similar results to the one for Poland were obtained in countries such as Malta, Belgium and Italy [113,114]. Interestingly, Gini coefficients for income and wealth of countries geographically and socially closer to Poland differ significantly from values reported for Poland.

However, as Chen [105] notes, “not all inequality curves yielding the same Gini coefficient are unequal in the same way”. This observation is important not only for comparisons between countries but also for monitoring changes in inequalities in a given country when applying or analysing policies in a particular counterfactual scenario. Using agent-based simulation, we can analyse how the Gini coefficients change between iterations as a consequence of the introduction of specific policies. Furthermore, we can also describe changes in inequalities using more accurate measures of asymmetry and measures of spatial inequality.

Among the most useful measures in identifying patterns of inequality are the Lorenz asymmetry coefficient [115,116,117], and radial triangular measures derived from the Lorenz asymmetry coefficient expressed in polar coordinates. The Lorenz asymmetry coefficient (LAC; abbrev. *S*) is defined as:(232)$$\begin{array}{c} S=F(\mu )+L(\mu ) \end{array}$$
where the functions *F* and *L* are defined as for the Lorenz curve, and μ is the mean. Based on an ordered set of data $\left(x_{1},x_{2},\ldots x_{m},x_{m+1},\ldots ,x_{n}\right)$, we can compute statistics *S* in the following way
(233)$$\begin{array}{c} \delta \;=\frac{\mu -x_{m}}{x_{m+1}-x_{m}} \end{array}$$
(234)$$\begin{array}{c} F\left(\mu \right)=\frac{m+\delta }{n} \end{array}$$
(235)$$\begin{array}{c} L\left(\mu \right)=\frac{L_{m}+\delta \;x_{m+1}}{L_{n}}. \end{array}$$
Damgaard and Weiner [115] show that Lorenz’s asymmetry coefficient allows us to notice that although two Lorenz curves allow in practice the same Gini coefficient to be obtained, their shapes are different, which translates into other patterns of inequality. If the LAC is less than one, the inequality is related to the presence of relatively much poorer units. If the LAC is greater than one then the inequality is related to the existence of extremely rich individual entities. For the purposes of further discussion, the point at which the first derivative of the Lorenz curve is equal to 1 in the interval (0,1), it can be denoted, similarly to Chen [105], [μ,f(μ)].

When calculating Lorenz’s asymmetry coefficient for income distribution based on quarterly data, a result of 0.9790896 is obtained. For annual data, this is 0.9788734. Similarly, for wealth distribution it is 0.942073944. Intuitively, if the value of the Lorenz coefficient is less than 1, the point [μ,f(μ)] is below and to the left of the symmetry axis. In fact, this case describes the situation in Poland, although the value does not differ significantly from the value of 1 corresponding to the point on the axis of symmetry. For the income distribution, the point [μ,f(μ)]=(0.63;0.35). This deals with the first of the two discussed situations. The income inequality in Poland is related to the relatively large number of poor agents. For wealth distribution, the point is equal (0.68;0.26) which is for the first case. Indeed, the wealth inequality is Poland is related to the relatively much poorer units.

The lack of access to disaggregated income and wealth data for countries such as Croatia, Ireland, Germany or Great Britain makes it impossible to compare patterns of inequality in these countries. However, it is still possible to compute Gini coefficients for Poland in different counterfactual scenarios. In the case of subtle changes in inequality in one country, it may be helpful not only to calculate the LAC but also to describe Lorenz’s asymmetry with polar coordinates and to calculate the additional measure of adjusted azimuthal asymmetry (AAA) (*cf.* Chen [105]).

In this new approach the location of a point [μ,f(μ)] relative to the point $\left(\frac{1}{2},\frac{1}{2}\right)$, a point dividing the line of equality in half can be described. In fact, [μ,f(μ)] can be designated according to its radial distance from $\left(\frac{1}{2},\frac{1}{2}\right)$ and according to the azimuthal angle formed by that radius, relative to the line of symmetry. According to convention, they can be denoted accordingly ρ and θ. For the graphical explanation of ρ and θ based on Chen [105], see: Figure 3. Both enhance the interpretation of Lorenz asymmetry coefficient. The maximum radial distance of [μ,f(μ)] from $\left(\frac{1}{2},\frac{1}{2}\right)$ is $\frac{1}{\sqrt{2}}$ and the azimuthal angle formed by the axis of symmetry and the line segment connecting $\left(\frac{1}{2},\frac{1}{2}\right)$ to [μ,f(μ)] falls within the range of +/−$\frac{\pi }{2}$.

The radial distance ρ measures the concavity of the Lorenz curve and it is expressed by the formula:(236)$$\begin{array}{c} \rho =\sqrt{{\left(\mu -\frac{1}{2}\right)}^{2}+{\left(f\left(\mu \right)-\frac{1}{2}\right)}^{2}.} \end{array}$$
Azimuthal angle θ measures the distance of [μ,f(μ)] from the axis of symmetry in precise angular terms. It is expressed by the formula:(237)$$\begin{array}{c} \theta \;=\arcsin(\frac{S-1}{\rho \sqrt{2}}). \end{array}$$
It adds precision and intuitive geometric appreciation to the Lorenz asymmetry coefficient. “Whereas the primary use of *S* has been binary negative values lie to the left of the axis of symmetry, while positive uses lie to the right θ locates [μ,f(μ)] with precision within the unit triangle” [105] Note that *S*, ρ and θ are constrained by the Gini coefficient. In practice, for a given value *G*, we have specified boundaries on the values *S*, ρ and θ. For any Gini coefficient *G*, the minimum value of *S* is *G* itself. The maximum value is 2−G.

For t=0 during the simulation, the following results are obtained for income distribution:(238)$$\begin{array}{c} \rho =0.19908,\theta \;=-0.07434,t=\frac{\theta }{\pi /2}=-0.04733. \end{array}$$
The distance from ($\frac{1}{2},\frac{1}{2}$) to the orthogonal intercept along the line of symmetry is 0.19886. The maximum θ given the value of Gini coefficient for quarterly data is equal to 0.99369. The adjusted azimuthal asymmetry coefficient for t=0 is equal to:(239)$$\begin{array}{c} AAA=\frac{\theta }{\theta_{max}}=-0.07481. \end{array}$$
Accordingly, for the wealth distribution, the following is obtained:(240)$$\begin{array}{c} \rho =0.30767,\theta \;=-0.13353,t=\frac{\theta }{\pi /2}=-0.08500. \end{array}$$
The distance from ($\frac{1}{2},\frac{1}{2}$) to the orthogonal intercept along the line of symmetry is 0.3049. The maximum θ given the value of Gini coefficient for quarterly data is equal to 0.5880. The adjusted azimuthal asymmetry coefficient for t=0 is equal to:(241)$$\begin{array}{c} AAA=\frac{\theta }{\theta_{max}}=-0.22783. \end{array}$$
As Chen [105] states, “adjusted azimuthal asymmetry promises a two-fold advantage over the unadorned Lorenz asymmetry coefficient. First, the asymmetry is more intuitively and more accurately expressed in angular terms than as the sum of two Cartesian coordinates. Second, because adjusted azimuthal asymmetry accounts for the maximum angular distance from the axis of symmetry for a particular value of *G*, it expresses a sense of proportionality that raw θ, to say nothing of *S*, cannot”.

### 5.3. Spatial Dimensions of Inequality and Inequality Aversion

Although the spatial analysis is based on basic Theil index (TI), the generalised entropy index (that the Theil index is a special case of) can be started at. The generalised entropy index is defined as: (242)$$GE\left(\alpha \right)=\left\{\begin{array}{ll} \frac{1}{N\alpha (\alpha -1)}\sum_{i=1}^{N}\left[{\left(\frac{y_{i}}{\bar{y}}\right)}^{\alpha }-1\right] & \alpha \neq 0,1\\ \frac{1}{N}\sum_{i=1}^{N}\frac{y_{i}}{\bar{y}}\ln\frac{y_{i}}{\bar{y}} & \alpha =1\\ -\frac{1}{N}\sum_{i=1}^{N}\ln\frac{y_{i}}{\bar{y}} & \alpha \;=0 \end{array}\right.$$
where *N* is the number of individuals or households, $y_{i}$ is the income of entity *i*, α is the weight given to distances between incomes at different parts of the income distribution. The generalised entropy index has been proposed as a better measure of income inequality with regard to Gini coefficient due to the fact that it is an additively decomposable inequality measure [118].

If it is assumed that α=1, that is GE(1), the TI is obtained. One of the advantages of the TI is that it is a weighted average of inequality within subgroups, plus inequality between those subgroups. For example, inequality within Poland is the average inequality within each region, weighted by region’s income, plus inequality between regions. We can express the decomposition formally. If for the TI the population is divided into *k* subgroups and $s_{i}$ is the income share of group *i*, $T_{T_{i}}$ is the Theil index for that subgroup, and ${\bar{x}}_{i}$ is the average income in group *i*, then then the TI is expressed by the formula:(243)$$\begin{array}{c} T_{T}=\overset{k}{\underset{i=1}{\sum }}s_{i}T_{T_{i}}+\overset{k}{\underset{i=1}{\sum }}s_{i}\ln\frac{{\bar{x}}_{i}}{\bar{x}}. \end{array}$$
The TI allows us to analyse the relative importance of spatial dimension of inequality, that is, the regional inequality [119].

For time t=0 during the simulation, based on empirical data of income and wealth, the entropy measure (EM) and TI for Poland is computed. We obtain respectively EM=0.2721004 and TI=0.2836961 for income, and EM=0.7599967 and TI=0.6753153 for wealth. The decomposition of TI allows us to analyse the spatial inequality in Poland. The results of measures of spatial inequality are not reported for confidentiality reasons. The authors cannot present which spatial codes 1-4 represent which region and districts in Central Statistical Office but it is feasible to present the data and visualise changes in spatial inequality in each counterfactual simulation using maps.

The generalised entropy index can also be transformed into a subclass of the Atkinson index (AI) with ϵ=1−α for 0≤α<1, defined as: (244)$$A_{\varepsilon }\left(y_{1},\ldots ,y_{N}\right)=\left\{\begin{array}{ll} 1-\frac{1}{\mu }{\left(\frac{1}{N}\sum_{i=1}^{N}y_{i}^{1-\varepsilon }\right)}^{\frac{1}{1-\varepsilon }} & \mathrm{for}0\leq \varepsilon \neq 1\\ 1-\frac{1}{\mu }{\left(\prod_{i=1}^{N}y_{i}\right)}^{\frac{1}{N}} & \mathrm{for}\varepsilon =1 \end{array}\right.,$$
where ε is an inequality aversion parameter. When ε approaches 0, it becomes more sensitive to changes in the upper end of the income distribution. Then, there is no aversion to inequality; no social utility is gained by complete redistribution. The higher ε, the more AI becomes sensitive to changes at the lower end of the income distribution. At the same time, more social utility can be gained from the redistribution as aversion to inequality is higher. Relatively low values of AI indicate a more equal distribution than higher values, given a particular degree of risk aversion. The computation of AI for the empirical distribution of income for Poland allows us to obtain the result of 0.1314228. The AI for the empirical distribution of wealth is equal to 0.3438987. A number of other measures of concentration and inequality have also been calculated to investigate the sources of income inequality in Poland. The Herfindahl concentration index (HHI) is equal 0.0005451883, while the Rosenbluth concentration index (RI) is 0.0004778672. The values of additional measures of wealth inequality are equal to 0.001255462 in the case of HHI and 0.4313275 in the case of RI.

### 5.4. Indebtedness of Households & Macroprudential Ratios

The distributions of debt service to income (DSTI), debt to income (DTI), loan to value for housing loans (LTV ${}_{H}$) and debt to income (DTA) at t=0 are analysed, and then the changes in the empirical distribution of these ratios at t>0 are presented in Appendix D. The percentage of households that have DSTI ≥ 30% (6%) and 40% (3.5%), as well as DTI ≥ 3 (3.6%) and 4.5 (1.7%) is computed. In the sample, 101 504 individuals have a total of 137 216 consumer loans. None of the individuals have more than 5 consumer loans. On average, individual customers have been charged statistically with 1.4 of consumer loans. In the sample, 43 008 housing loans were still to be repaid.

## 6. Results of Simulations

Using this agent-based simulation, it is possible to analyse the behaviour of a heterogeneous economy within certain counterfactual scenarios and the effects of public policies, especially stabilising and distributional effects of macroprudential policies. Firstly, the scenario is analysed in which the main mechanisms described by H. Minsky in ‘Stabilizing unstable economy’ (1986) [44] and in ‘Financial Instability Hypothesis’ (1992) [45] are simulated. Under the second scenario, the behaviour of the system is investigated, assuming the use of macroprudential policies to mitigate financial risk and stabilise an unstable economy. Then, the macroprudential policy stance which would best allow for the stabilization of heterogeneous economy is analysed, as is the distributional effects of these policies.

### 6.1. Minsky Moment

According to Reference [44], ‘capitalist economies exhibit inflations and debt deflations which seem to have the potential to spin out of control’. The economies are unstable by nature (this instability, in the most general terms, was mostly related to industrial structure and finance). In this statement Minsky refereed to the Kindleberger’s definition of self-sustaining disequilibrium from the 1970s. The capitalist economy is conditionally coherent. Instead of accepting the state of equilibrium, the focus should rather be on periods of tranquillity, apparent stability, which, in essence, are destabilising.

The neoclassical synthesis tries to explain how a decentralized economy achieves *coherence* and *coordination* in production and distribution (in other words, how market mechanisms lead to a sustained, stable-price, full-employment equilibrium). In opposition, the Minsky theory focuses on capital development of an economy and *the impact of financial institutions’ activity on production* and *distribution*. The optimum that is derived from the neoclassical decentralized market processes ‘rules out interpersonal comparisons of well-being and ignores the inequity of the initial distribution of resources and thus of income’. As Minsky stated, ‘inasmuch as our aim is to indicate how we can do better than we have, and as the best is often the enemy of the good, we can forget about the optimum [in neoclassical terms]. Even though a tendency toward coherence exists because of the processes that determine production and consumption in market economics, the processes of a market economy can set off interactions that disrupt coherence’ [44].

Minsky’s theory can be interpreted in two ways. In Keynesian terms, it should be understood in terms of accumulation of capital in the economy. In Knightian terms, it should rather be interpreted as a problem of allocating resources under risk and uncertainty. The paper refers to the first interpretation. In *Extensions*, the second interpretation is referred to. Hence, the role of risk perception and uncertainty in generation and amplification of risk within the system is analysed [42].

We begin with interpretations of the instability hypothesis in Keynesian terms. Minsky’s theory refers to the general theory of Keynes from the 1930s. Nonetheless, in ‘Stabilizing an unstable economy’, the process of capital accumulation described by Keynes is accompanied by an exchange of current money for the future. At the heart of this theory are not only capital stock and investment but primarily cash flows. Any attempt to model the instability hypothesis must therefore be stock-flow consistent. In reference to the Minsky theory, in the simulation, three types of stylized cash flows are developed: income, balance sheet and portfolio. The income cash flow refers to all payments in the production and sale of inputs and final goods, thus also the one in the supplier searching and purchase modules. The balance sheet cash flow refers to repayment of debts. The third type of cash flow occurs due to transactions in which capital and financial assets are acquired by a new agent. Money in cash form is not analysed explicitly in the model, but the general idea of money and settlements has been taken into account in the model. As Minsky noted, ‘money is created in the process of financing investment and positions in capital assets. An increase in the quantity of money first finances either an increase in the demand for investment output or an increase in the demand for items in the stock of capital or financial assets. As money is created, borrowers from banks enter upon commitments to repay money to the lending banks. In its origins in the banking process, money is part of a network of cash-flow commitments, a network that for the business side of the economy ultimately rests upon the gross profits, appropriately defined, that firms earn’ [44].

The current money is used to purchase inputs that are used in the production of investment and consumption goods. The purchase of inputs can be funded by the firm from the liquid assets and capital. Otherwise, external funding can be obtained. Financial institutions on the market generate profits from lending funds. ‘The financing terms affect the prices of capital assets, the effective demand for investment, and the supply price of investment outputs’ [44]. Firms, establishments and individuals are obliged to repay debt, that is, principal and interest rates, within designated deadlines. In Minsky’s basic theory, the focus was on investment. The government budget, the behaviour of consumption and the path of wages were secondary. To make the model more realistic and ensure it can be applied in modern policy making, all the above-mentioned elements were included. Similarly, the flow of money described by Minsky included only loans to firms, while in the simulation, loans to individuals and households were also added. The flow of money in Minsky’s theory, similarly to the simulation, has the following two directions. Individuals make deposits to banks and banks lend money to firms and individuals. In a later period, firms and individuals return funds to banks and banks to depositories.

In this theory the cash flows are influenced by the *expected* future profits; the flow of money from companies to banks takes place after the realisation of real profits. These expectations need not be always consistent with the final realisation of profits.

Funds can be obtained through negotiation, in which risk perception and uncertainty as well as expectations play an important role. Firms willing to obtain money for carrying out their activity interpret financial results and the economic situation in an enthusiastic way, and in fact often create overly optimistic expectations. While bankers are inherently conservative, though profit-seeking, they are more restrictive in assessing potential gains from a deal. Despite being more restrictive, bankers are also aware that investment in innovation and new products, services and industries is the most profitable. Consequently, their propensity to finance projects from the most dynamic and profitable sectors is usually greater than the projects from other industries, even if the riskiness of the project is higher than average.

Minsky’s theory incorporates elements of Kalecki-Levy theory, according to which the structure of aggregate demand determines profits. The profit expectations depend on the expected level of investment in the future and actual returns on investment. Investments continue on the assumption of both entrepreneurs and bankers that investment will also take place in the future. In order to increase investment, the agents borrow additional funds, frequently assuming unrealistic returns in the future.

According to Minsky, there are three types of agents that are characterised by their relation between income and debt. ‘Hedge financing units are those which can fulfil all of their contractual payment obligations by their cash flows. (...) Speculative finance units need to roll over their liabilities, [that is,] issue new debt to meet commitments on maturing debt. (...) For Ponzi units, the cash flows from operations are not sufficient to fulfil either the repayment of principle or the interest due on outstanding debts by their cash flows from operations. Such units can sell assets or borrow. Borrowing to pay interest or selling assets to pay interest on common stock lowers the equity of a unit, even as it increases liabilities and the prior commitment of future incomes. A unit that Ponzi finances lowers the margin of safety that it offers the holders of its debts’ [44]. The higher the leverage ratio of firms, the higher the probability of defaulting on debt. Therefore, the reduction of collateral required and credit rating requirements in this scenario leads to an increase in the percentage of speculative and Ponzi agents. Changes in creditworthiness requirements also affect the percentage of hedging, speculative and Ponzi-type agents in the economy. The higher the level of speculators and Ponzi-type agents, the greater the probability of a crisis.

Investment boom increases aggregate demand and spending through a multiplier effect and sales increase. Profits increase with increasing investment, encouraging further investment. In this way, the instability of the system is strengthened until the percentage of speculative investors increases significantly. In the case of prolonged prosperity, assuming that no prudential policies have been applied, the economy becomes unstable, due to the increasing number of speculative and Ponzi-type agents.

Minsky’s instability hypothesis is a study about the extent to which debt affects the behaviour of the system and how the level of debt is considered to be adequate in terms of dynamics of the system. According to this theory, there is no need for external shock for the crisis to occur. The crisis is generated endogenously by agents taking too much risk and by the desire by entrepreneurs and bankers to obtain ever-increasing profits. At the same time, this situation also has consequences for individuals and households. In the event of an increase in insolvencies, banks may default and thus depositors would lose the funds (the case in which by increasing uncertainty individuals decide to withdraw deposits is analysed in *Extensions*, see: [8]). In addition, the situation on the labour market determines changes in the income and wealth of households.

According to Minsky’s theory, increase in investment ‘would never trickle down to the poor and would tend to increase inequality by favouring the workers with the highest skills working in industries with the greatest pricing power’ [44]. In the simulation, this mechanism was modelled. Firms hire individuals according to their work skills and in industries with the greatest pricing power. Consequently, it leads to inequality. The changes in inequality can be measured using the Gini coefficient as well as other measures of asymmetry and spatial inequalities.

The dynamics of the real estate market is also changing. The increased percentage of job seekers is influencing their shift to cheaper properties and partly ineligible mortgage loan repayment, which again puts banks in a difficult position. Banks are in possession of properties for which there is no demand, and their prices are gradually lowered in order to find a buyer.

Two main components of the theory are the two-price system and the lender’s and borrower’s risk, both derived from theories of Kalecki and Keynes.

The first price system concerns current output prices, that is, costs and mark-ups, that need to be set at a level that will generate profits for the firm from the sale of consumer goods, investment goods, and goods and services purchased by the government in public contracts. If, in the analysis, the external fund increase is also taken into account, the supply price of capital plus the interest rate and lender’s risk must also be considered.

The second pricing system refers to assets. Assets are expected to generate cash flow in the future. These flows are not known and their estimation depends on subjective expectations. How much is able to be paid for such a financial asset depends on the amount of external finance required. The more the borrower becomes indebted, the greater the risk of insolvency; in this sense the price of the asset includes the borrower’s risk. Investments occur when demand price exceeds the supply price of assets. Prices include collateral. After the crisis, usually larger collateral are required, in the expansion period, they are lowered significantly.

As Minsky [44] states, ‘the costs of financing the production of investment is a cost that enters the supply price of output like the costs of labour and purchased inputs. The fact that a firm has to borrow to pay wages raises the effective costs by the interest payments on the borrowings’. ‘The decision to invest therefore involves a supply function of investment, which depends upon labour costs and short-term interest rates, a demand function for investment, which is derived from the price of capital assets, and the anticipated structure and conditions of financing. Whereas the structure of balance sheets reflects the mix of internal funds (gross retained earnings) and the external funds (bonds and equity issues) actually used, the investment decision is based upon expected flows of internal and external funds’ [44].

### 6.2. Simulating an Unstable Economy

In the first scenario, the role and impact of financial institutions on production and distribution of income and wealth in the economy is simulated. As Minsky pointed out, according to neoclassical assumptions, the initial distribution of income and wealth did not matter. Neoclassical models ignore the impact of policies on income and wealth distributions *ex post*. The inclusion of *ex ante* heterogeneity was not relevant to the analysis of outcome *ex post*. In contrast, in the simulated scenario, we show that the processes of a market economy do ‘set off interactions that disrupt coherence’, and that incorporation of heterogeneity *ex ante* into the model makes it possible to observe changes in distributions *ex post*, including income and wealth distributions.

In accordance with Minsky’s hypothesis, in the database and model design, the focus was put not solely on heterogeneity in capital stock and on different investment decisions but also on balance sheet changes and cash flows.

In the scenario, firms operate in eight different sectors through establishments. In the first iteration, the most profitable sector is the first sector, thus it attracts the highest percentage of new investors. Other sectors attract a smaller percentage of new businesses. During first iterations we observe fluctuations in the relative profitability of sectors, related to business fluctuations, that is, the effect of the business cycle. In the initial iterations, only stock fluctuations related to intrinsic dynamics of the economy are observed, while in further iterations, it is observed that increased stock fluctuations and a cease in production is experienced by a higher percentage of establishments. The number of firms in each of sectors decreased over a year due to both higher rate of bankruptcies and higher concentration on the market, *see:*
Figure A6 (on the right) in Appendix D. At t=0 there are 18727 establishments in the database. At t=1, 14112 establishments from the database at t=0 continue operating on the market. 4615 establishments ceased to produce, while 9887 new establishments were attracted by the market due to higher relative profitability of selected sectors. At t=2, 15727 establishments from t=1 operate on the market, while 2773 new establishments are created. In total, 18500 establishments are in operation. At t=3, 15357 establishments from t=2 operate on the market but only 1546 new establishments are established due to deterioration of market conditions. In total, a lower number of establishments (16,903) with respect to t=0 operate on the market. Some firms go bankrupt, while others cease their activity in selected establishments that are operating in less profitable sectors. At t=4, 15,357 establishments stay on the market and 1491 new establishments are created. In total, 16848 establishments continue operation.

In next iterations, there are higher levels of debt and leverage and increased financial risk in the sectors that were the most profitable in the previous periods. The risk spreading between sectors was visualised in Figure A5 (on the left and on the right side) and Figure A6 (on the left). Over a year, the higher number of firms’ bankruptcy with respect to the previous periods was observed, indicating the gradual *endogenous* generation of the crisis. Occasionally, though gradually more frequently, firms had problems finding a supplier, which is related to the cessation of business activity by selected establishments, and thus the interruption of network of transactions established between establishments and suppliers. The data could be analysed for further iterations than four (one year) but the goal was to show that a crisis can emerge endogenously in a relatively short period of time, such as 1–4 quarters, as it was the case during the financial crisis (Q2 2008–Q2 2009).

The problem of searching and matching the suppliers in the supply chain module gains importance. The time required to searching for supplier is increased due to the fact that new transaction relations are established with suppliers in further territorial units than previously.

When the uncertainty on the market increases and indicators of profitability and riskiness deteriorate, a lower percentage of firms is eligible for a loan. Establishments lose creditworthiness and are forced to restrict production, which in turn worsens economic conditions. At t=1, 62.5% of establishments were checked for creditworthiness in the module *Consumer credit admissibility* 1, while 37.5% in *Consumer credit admissibility* 2. In the following iterations, these values were respectively: 48.61% and 51.39% at t=2, 55% and 45% at t=3, and 57.5% and 42.5% at t=4. The firms and establishments’ conditions are also closely related to the problems of specific industries and fluctuations in the economy.

Expectations of lower profits lead to a reduction in the flow of money from companies to banks and from banks to companies, worsening the state of the economy. Banks are on the one hand aware of the higher risk related to granting loans to companies that qualify for the liquidity problem procedure, on the other hand it allows them to earn a premium. Overall, however, they provide loans for shorter periods, which is related to the general uncertainty in the market.

In the simulation cycle under first scenario, the number of companies with higher debt and higher leverage gradually increases. Using the simulation, changes can be observed in the distribution of the firms’ debt and leverage, rather than just analysing growth of the overall average debt and leverage. However, as the crisis becomes more severe, more firms with excessive debt and leverage go bankrupt or cease their activities in selected establishments; see: Table A11 in Appendix D.

The initial boom also encourages households to increase consumption, which in turn increases their overall debt, also affecting the value of debt service to income (DSTI), debt to income (DTI), debt to assets (DTA) and loan to value (LTV) ratios, see: Figure A12, Figure A13, Figure A14, Figure A15 and Figure A16. After the first iteration, the percentages of households that have DSTI ≥ 30% equal to 4.7%, and DSTI ≥ 40% equal to 2.7%, while the percentage of households with DTI ≥ 3 equal to 3% and DTI ≥ 4.5 equal to 1.3%. After the second iteration, the percentage of households with DSTI ≥ 30% was equal to 3.6%, and DSTI ≥ 40% equal to 2.2%, while the percentage of households with DTI ≥ 3 equal to 2.1% and with DTI ≥ 4.5 equal to 0.9%. After the third iteration, the percentage of households with DSTI ≥ 30% is equal to 4.6%, and with DSTI 40% ≥ 3.2%, while those with DTI ≥ 3 is equal to 1.9% and with DTI ≥ 4.5 equal to 0.9%. After one year, the percentage of households with DSTI ≥ 30% was equal to 5.04%, and with DSTI ≥ 40% was equal to 3.5%, while with DTI ≥ 3 this percentage was equal to 2.2% and with DTI ≥ 4.5 the percentage was equal to 1.1%. The percentage of households with respective values of DSTI and DTI at t>0 are lower than at t=0. There are two reasons that explain this pattern. Firstly, households with very excessive debt default are removed by the system. Secondly, highly indebted households will not apply for any new loans and they pay back part of the debt. Nonetheless, as soon as their creditworthiness improves due to loan repayment, they are attracted by the market via mechanisms described by Minsky and the ratios start to deteriorate. Changes in investment and external financing directly affect the market imbalances. Individuals with lower qualifications and level of education experience a gradually deteriorating situation on average.

After a year the Gini coefficient for income distribution is equal to 39.9% and for wealth distribution is 60.8%. After two years, the values are 41.1% and 61.3%, respectively. The LAC is equal to 0.96901 and 0.93896 after four quarters. The point [μ,f(μ)]=(0.63;0.33) for income, while it is equal (0.66;0.26) the case of wealth. The adjusted azimuthal asymmetry (AAA) is equal to −0.07541 for income and −0.21341 for wealth.

Due to endogenous changes in the decisions of agents and their interactions, the dynamics of the markets and economy also changes. In particular, the dynamics of the labour and real estate markets change. Higher mortgages and a difficult situation on the labour market affect the dynamics of the real estate market. More properties are marked for sale and their price is reduced accordingly. On average, the prices of properties decreased by 2% in one year. It is possible to analyse a decrease in prices in each spatial unit, for example, region, however the results are not reported for confidentiality reasons.

With the deterioration of a favourable economic situation, the number of insolvencies and the NPLs ratios of banks increased; see: Appendix D (Figure A7 (on the right) and Figure A9 and Figure A10 (on the left and right). Higher non-performing loans affect the supply of credit in subsequent iterations. In the model it is possible to use two methods of determining the supply. If the first one is used, banks adopt similar strategies in groups. If the second is used, there may be a greater degree in the heterogeneity of strategies. In both cases, these patterns were observed. The model allows the analysis of differences in non-performing loans according to the loan type which thus reflects another aspect of heterogeneity. Using simulation, we can also analyse the changes in profits (Figure A8 (on the left)), equity (Figure A8 (on the right)) or the fulfilment with the capital and liquidity requirements which were set on too low levels (Figure A11 (on the left) and Figure A11 (on the right)).

#### Stabilizing Unstable Economy via Macroprudential Tools

In the second scenario, the behaviour of the economy is simulated, assuming the implementation of macroprudential policy aimed at stabilising the unstable economy. All banks set the capital requirements set by the regulator as well as apply the recommendations with respect to debt to income (DTI), debt service to income (DSTI), debt to assets (DTA), loan to value (LTV) ratios and leverage ratio (LR). In the economy, the activity of companies is conducted in the eight sectors grouping NACE industries. Among the analysed sectors, the most attractive for potential entities is the third sector, while the relatively least profitable for investors is the eight sector. In the case of any industry, there is no significant decrease in relative profitability. In first scenario, it was observed that the role of some of the sectors, that is, real estate sector, increased sharply, and then was significantly reduced. In the second scenario, no sudden discontinuity of production or increasing number of bankruptcies were observed. There are were no significant changes in product prices.

The expectations are relatively much more consistent in time than in the case of the first scenario. In this scenario, these expectations are only slightly different than those formed in the previous period and adapt to the fluctuations of the economy. Stock fluctuations are consistent with the dynamics of the economy. There is no stock accumulation or stop to production and sales of stored goods. In this scenario, the leverage level is moderate. Only in the case of a small percentage of establishments was production stopped completely.

In this scenario there are no searching-matching problems between the establishments in the search for suppliers. Quality-to-price ratio of establishment in relation to quality-to-price ratio in a given sector is the appropriate determinant of the supplier’s search decision. In most cases, it is possible to find a supplier in a given spatial unit that has a sufficient quantity of produced and stored goods. The search for a supplier in the unit that exceeded the basic spatial unit is moderate and corresponds to normal time market dynamics. The additional costs generated by transportation of inputs for establishments’ production of goods were also moderate.

Most firms have adequate liquid assets and adequate creditworthiness. Additional funds will be used for further investment. A very low percentage of establishments were eligible for the creditworthiness check in connection with transitional liquidity problems. Transient problems of establishments on the market are related to temporary economic fluctuations in industries, as well as market dynamics. In this scenario, banks have adopted similar credit requirements for less risk-prone companies. However, banks differ in credit requirements for more risk-prone businesses, especially firms with temporary liquidity or financial problems. The tightening of the criteria has countered the financial crisis. Of particular importance was the observation of return on assets (ROA), return on equity (ROE), financial risk and credit history of companies.

In the case of a low number of establishments, a readjustment of quantity was required, and the final purchase of the inputs was lower than the intended one. Banks providing loans for the purchase of inputs were characterised by different risk aversion and strategies on the market. In some cases, the network effects have been maintained. The establishments applied for a loan in a matched bank, thus maintaining a transaction link (edges in the network). For other transactions, there was a change of the loan-granting bank. When searching for a bank that will grant short-term credit for the purchase of inputs, the companies were mainly driven by the interest rate and the supply of credit.

In the model it is possible to use two methods of determining the supply. If the first one is used, banks adopt similar strategies in groups. If the second one is used, a greater degree of heterogeneity of strategies is observed. In the first case, network effects associated with transaction relationships in the market have been maintained. Individuals choose banks to apply for a loan according to the interest rates on consumer loans offered by banks. Granting a loan also depended on credit supply and sectoral exposure requirements. Some banks have exhausted the supply of consumer loans, which means that according to the strategy defined for a given period, the funds were spent on other activities such as granting short-term and long-term (investment) loans to companies, and residential and non-residential loans to individuals.

Within the second method, during the year the same number of individuals applied for a loan at the matched banks; a very high percentage of these applications were successful. In this case, network effects associated with transaction relationships in the market were maintained. The supply of consumer loans and sectoral requirements limited the number of loans granted.

In both cases there were also no searching-matching problems between consumers and suppliers of goods. Also, the quality-to-price ratio in relation to quality-to-price ratio in a given sector is the appropriate determinant of the supplier’s search decision. In most cases, it has been possible to find a supplier in a given spatial unit that had a sufficient quantity of produced and stored goods. In some cases, the search for suppliers exceeded the basic spatial unit, generating a modest cost of goods transportation. It did not generate an excessive burden on households.

Demand of establishments and households was complemented by demand from the government, through public procurement. The establishments that signed the public contracts produce high-quality products. These establishments were also part of the largest companies on the market.

In the case of households within ‘Housing stress!’, the cost of accommodation, whether in the case of servicing a housing loan or renting, proved to be excessive. These households changed their residence to a cheaper property. The dynamics of the housing market was not excessive and was consistent with household income fluctuations. In the case of a very low percentage of households, their insolvency was observed, as was an increase of NPLs in the banks by the value of their loans. The prices of properties fluctuate on the market as a whole but the prices remain relatively stable within a determined region.

Using the simulation, it is also possible to compute which percentage of households purchased a new property in cash and which have applied for a residential or non-residential loan. Demographic trends were fully preserved in the scenario. The probability of survival or death has been specified according to Central Statistical Office data. As a consequence of the death of individuals, inheritors gained additional deposits. Most companies were taken over by the heirs, while in cases of negligible value the inheritance was rejected. In some cases, the deceased was the owner of the property charged with the loan. These properties were taken over by the bank and were resold at a lower value. In the case of survivors, the tendencies of completed education by individuals were also maintained.

Some companies were not created despite starting the company opening process. Some did not obtain a business license, and in some cases, funds were not sufficient to run a business. Credit applications are a special case in which the entrepreneurs compared the interest charged by the banks. The banks also checked whether the supply side restriction was fulfilled and whether the sectoral exposure did not exceed the requirements. Sectoral exposures have made it possible to limit potentially excessive credit growth in the most profitable sectors at any given moment, including in the real estate sector, which normally expands dynamically during prosperity. The larger companies were funded with contributions from entrepreneurs. There was an increase in average goodwill points during an expansive phase of a business cycle. At the same time, the structure on the market changed, with some activities ceasing and other new activities being created. New establishments were mostly opened in sectors with high or moderate risk. Changes in the number and structure of firms on the market corresponded to the usual dynamics of the economy. There was no increased concentration of capital, business clustering, excessive bankruptcies or escalation of reductions of labour force in already operating establishments.

An analysis of the distributions describing the attributes of firms and establishments does not allow the identification of situations typical of financial or economic crises and symptoms of overheating of the economy. In this scenario, a significant increase in the risk of activity of firms and establishments on the market was not observed, nor was the strong growth of the economy and the increased financial risk of a particular industry. The average risk to the business activity in the market was moderate.

The lack of concentration of financial risk in a given industry is largely due to the introduction of regulations for maximum exposure to a given industry. The situation of banks is stable. For the first and the second procedure of supply determination, the NPL ratios were moderate. Liquidity requirements are met in the case of most banks. Capital requirements were fulfilled at the level of 8%, introduced by bank regulator. None of the banks declared insolvency. A very low percentage of individuals declared insolvency as a result of net savings at a negative level. The NPL ratios and the growth (inflow) of NPLs of banks was modest.

The stabilising effects of macroprudential policies for the economy and the financial system is significant, however the effect on reducing inequality is ambiguous. The richest agents on the market seem to remain unaffected by the introduction of the policies. In extreme cases, the rich get richer. After one year, the Gini coefficient for income distributions was equal to 39.7% and 60.6% for wealth. After two years, the values were 40.1% and 60.9%, respectively. The Lorenz Asymmetry Coefficient (LAC) was equal to 0.97672 and 94782 respectively. The point [μ,f(μ)]=(0.62;0.36) for income, while it is equal (0.67;0.26) for wealth. The adjusted azimuthal asymmetry (AAA) is equal to −0.07541 in the case of income and −0.21341 in the case of wealth. The results do not differ significantly in the next four iterations.

### 6.3. Optimality of Macroprudential Policy Stance

Thus far, the possibility of stabilising an unstable economy using macroprudential policy has been analysed. Simulation has allowed analysis of the degree to which macroprudential policies influence the distribution of variables, including the distribution of income or the distribution of wealth, allowing a fuller welfare analysis. However, the main question is which policy combination to choose from, utilising possible combinations of tools to stabilise the economy and financial system, as well as to ensure a more homogeneous distribution of wealth and income. Optimal choice is understood here to mean the method of choosing the most advantageous combination of macroprudential tools (from the point of view of social well-being). In contrast to the social planner approach, the abstract concept of Bentham’s social utility is not focused on; rather there is focus on the stability of the economic and financial system and the measures of welfare that quantify changes in income and wealth distributions.

Based on the scenarios, it can be concluded that the use of a combination of macroprudential tools (appropriately calibrated) may lead to stabilisation of the system. Nonetheless, this combination of policies was not optimal in a sense that the statistical equilibrium was achieved at a level which does not guarantee a sufficient reduction of inequality in the system. The optimal policies would be the ones which enable a decrease in inequality in the system (to an acceptable level). The results were shown to be robust by carrying out the Monte Carlo procedure. The exact results for the Polish system (including optimal combination of macroprudential tools) are not reported here due to confidentiality of individual banks’ and firms’ data. The simulation can, however, be performed using data available from the National Bank of Poland, which would allow plausible results to be obtained. The results in the scenarios are mostly illustrative to the new methodology developed.

## 7. Final Remarks

The paper has analysed the stabilising effects of macroprudential policies on a heterogeneous economy using an agent-based approach. The presented simulation is a novel application of agent-based approach in systemic risk modelling. The model constitutes an alternative to the ‘3D’ model with three layers of default that is widely-used by experts of the European Systemic Risk Board and the European Central Bank. The main advantage of the ABM model with respect to the ‘3D’ model is the possibility to carry out counterfactual simulations within the framework of fully heterogeneous agents.

The simulation results show the stabilising effects of binding macroprudential policies on the unstable heterogeneous economy. However, in opposition to mainstream literature, the use of macroprudential policies as an alternative to redistribution policies does not always lead to the same results if heterogeneity is assumed in the model. Even if the shape of the income and wealth distributions could be smoothed for most percentiles, the macroprudential policies do not affect the richest individuals in the desired way; the richest get even richer with respect to the rest of society. In general, the macroprudential policies and regulation should counteract the negative effects of the crisis, including the impact of the crisis on society (distributional effects). Nonetheless, this paper shows that suboptimal or non-binding macroprudential policies in the economy with heterogeneous agents *ex ante* and *ex post* do not remove the all negative distributional effects. Poor calibration of macroprudential policies would not only counteract the endogenous generation of the crisis in Minsky’s sense but would also lead to higher inequality. This finding is consistent with the ECB studies of P. Hartmann [120] on the distibutional effects of macroprudential policies and the possible interactions between social policies and macroprudential policies. This view was also supported by van der Heuvel [13] and Claessens [121]. Moreover, the policy recommendations and results of welfare analysis obtained from the DSGE models with homogeneous agents may be in practice very misleading for central bankers.

## Figures and Tables

**Figure 1 entropy-22-00129-f001:**
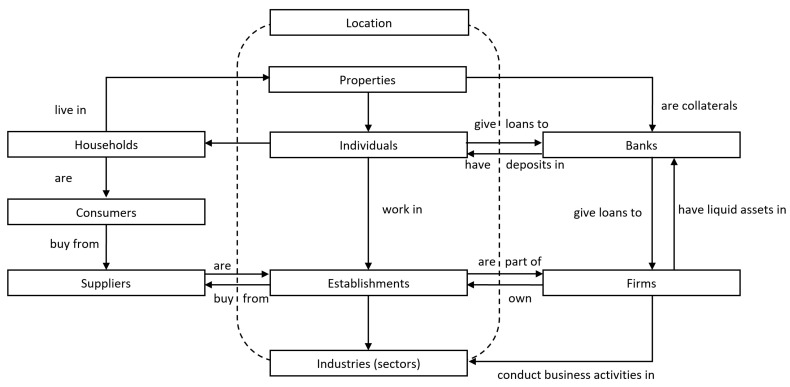
Relations between agents in the model.

**Figure 2 entropy-22-00129-f002:**
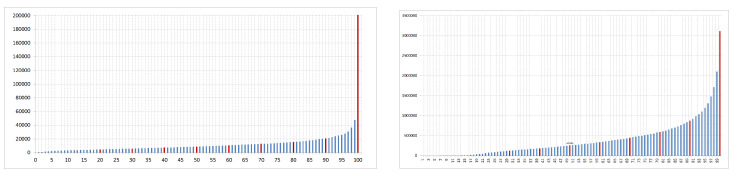
Household income by percentile (on the left) & Household wealth by percentile (on the right).

**Figure 3 entropy-22-00129-f003:**
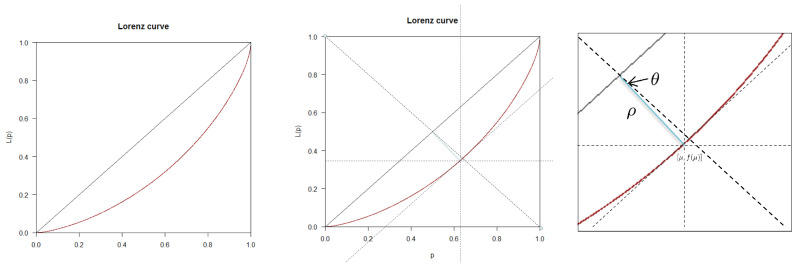
Lorenz curve (on the left) and the radial interpretation of ρ and θ (in the middle and on the right).

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
