# Peer review of "Macroprudential Policy in a Heterogeneous Environment—An Application of Agent-Based Approach in Systemic Risk Modelling"

_entropy, 2020, doi:10.3390/e22020129_

Round 1

Reviewer 1 Report

Dear Authors, 

I like your work very much, thank you for such (in my opinion) good paper.

Abstract - is very clear formulated,

Introduction - has all the necessary attributes and clearly sets out the main idea,

Models comparison - is good, the authors are not limited to only a few scientific sources, but provide the analysis of current scientific works.

The Modules are very clear described. 

The references list - relevant.

I don't have any negative comments.

Author Response

Regarding the first review, the reviewer emphasized that the abstract is clearly formulated; the introduction has all of the necessary attributes and clearly sets out the main idea; the model comparison is good (the authors are not limited to only a few scientific sources, but provide the analysis of current scientific works); the modules are clearly described and the references list is relevant.

The first reviewer did not submit any negative comments.

The reviewer did not feel qualified to judge about the English language and style. However, we would like to emphasize that the article was proofread by a native English speaker – Mr. Russell Pepper (University of Dundee).

Please find enclosed the whole response for comments addressed by both reviewers

Reviewer 2 Report

The paper is long;  I did not check the whole set of mathematical formulations.

(line 50)   The aim of this paper is to analyse the impact of selected macro-prudential policy tools on the economic and financial system using agent-based modelling (ABM).

(line 54) The paper contributes to the existing literature of agent-based modelling through detailed and relatively broad insight into heterogeneity of agents, and make a comparison between the ABM and DSGE-3D model(s) ; that means (line 36) Dynamic Stochastic General Equilibrium (DSGE) models. The DSGE models share the assumption of a perfectly rational representative agent that dynamically optimizes the use of resources.

One appreciate (line 273) that is presented … an ABM model suitable for performing simulations that provide detailed insight into the nature of the relationship between the financial system and the real part of the economy.

In conclusion, (line 1646), the paper has analysed the stabilising effects of macroprudential policies on a heterogeneous economy using an agent-based approach. The presented simulation is a novel application of agent-based approach in systemic risk modelling.

I suppose that the authors will take care of improving the reference mentioned on line 69 (Clerc et al. 2015) ; isn’t it [28] ?

A main point has to be raised : the manuscript seems to be missing some well grounded approaches to agent- based models , not found in the review of the literature, but of interest, whence should not be neglected. Strangely, many references pertain to Italian authors. However, work by a few are missing.

For example, co-evolutive models have been successfully applied in F. Petroni, M. Ausloos, G. Rotundo, “Generating synthetic time series from Bak-Sneppen coevolution models mixtures”, Physica A (ISSN: 0378-4371) 384 (2007) 359–367. doi:10.1016/j.physa.2007.04.127. IF 1.522, WOS:000249942400020.

and in

Rotundo, M. Ausloos, “Microeconomic co-evolution model for financial technical analysis signals”, Physica A ISSN: 0378-4371 373 (2007) 569–585. DOI:10.1016/j.physa.2006.04.062. IF 1.522, ISI(WOS):000242316000049, SCOPUS 2-s2.0-33751064105

for explaining the stylized fact of persistency in time series.

Moreover, general reviews like

Varela, L. M., Rotundo, G., Ausloos, M., & Carrete, J. (2015). Complex network analysis in socioeconomic models. In Complexity and Geographical Economics (pp. 209-245). Springer

Varela, L. M., & Rotundo, G. (2016). Complex network analysis and nonlinear dynamics. In Complex Networks and Dynamics (pp. 3-25). Springer

are missing ; so is a fundamental review paper

Ausloos, M., Herbert D., and Merlone, U., "Spatial interactions in agent-based modeling." In Complexity and Geographical Economics, pp. 353-377. (Springer,   2015).

Nevertheless, I recommend that the paper should be rejected on the following ground: a very annoying point concerns the framework of the paper, insisting on Polish data, and papers in polish, whence references. Many cannot be retrieved or read by most scholars. Should I mention them ? all ?

I have many friends (and enemies) in Poland and in Italy, and wish to put those into evidence, whence I wish this work to be known ; I do respect much the present aims of the paper, but, in my opinion, it cannot satisfy an international audience mainly used to reading papers written in English with pertinent references in English.

Very regretfully, I recommend to reject the paper, or revised it, but this is a huge work. I have no idea how to replace the Polish references and Polish data by others.

Author Response

Regarding the second review, we have made changes to the article as suggested by the reviewer.

We have improved the reference mentioned on line 69 (Clerc et al. 2015) (we changed it for [28], which is [33] in the new version of the references). With regard to the remark that the manuscript seems to be missing some well-grounded approaches to agent-based models, not found in the review of the literature, but of interest, we would like to thank the reviewer for his/her suggestions for including articles on novel approaches.

After careful consideration, we decided to include two articles that seemed most interesting to us in the literature review because, as the reviewer correctly pointed out, they explain the stylized fact of persistency in time series.

Petroni, F.; Ausloos, M.; Rotundo, G. Generating synthetic time series from Bak-Sneppen coevolution models mixtures. Physica A 2007, 384, 2, 359–367. doi: 10.1016/j.physa.2007.04.127.

Rotundo, G.; Ausloos, M. Microeconomic co-evolution model for financial technical analysis signals. Physica A 2007, 373, C, 569–585. doi: 10.1016/j.physa.2006.04.062.

In lines 245-246 we added: “In the broader sense, the study also refers to the coevolution models that were successfully applied in [110,121] to explain the stylized fact of persistency in a time series.”

In addition, in the footnote (7) on page 6, we also included the references to more general reviews (“For more general reviews on complex network theory, refer to [132,133], while spatial interactions in agent-based modelling were discussed in [7]”).

Varela, L. M.; Rotundo, G.; Ausloos, M.; Carrete, J. Complex network analysis in socioeconomic models. In Complexity and Geographical Economics; Commendatore, P., Kayam, S., Kubin, I., Eds.; Springer, Switzerland, 2015, pp. 209–245. doi: 10.1007/978-3-319-12805-4_9.

Varela, L. M.; Rotundo, G. Complex network analysis and nonlinear dynamics. In Complex Networks and Dynamics; Commendatore, P., Matilla-García, M., Varela, L., Cánovas, J. Eds.; Springer, Switzerland, 2016, pp. 3–25. doi: 10.1007/978-3-319-40803-3_1.

Ausloos, M.; Herbert D.; Merlone, U. Spatial interactions in agent-based modeling. In Complexity and Geographical Economics; Commendatore, P., Kayam, S., Kubin, I., Eds.; Springer, Switzerland, 2015; pp. 353–377, ISBN 978-3-319-12804-7.

We also included the following papers in the references:

Barkauskaite, A.; Lakstutiene, A.; Witkowska, J. Measurement of Systemic Risk in a Common European Union Risk-Based Deposit Insurance System: Formal Necessity or Value-Adding Process? Risks 2018, 6, 137. doi: 10.3390/risks6040137.

Boeing, G. The Effects of Inequality, Density, and Heterogeneous Residential Preferences on Urban Displacement and Metropolitan Structure: An Agent-Based Model. Urban Sci. 2018, 2, 76. doi: 10.3390/urbansci2030076.

Choi, Y. Masked Instability: Within-Sector Financial Risk in the Presence of Wealth Inequality. Risks 2018, 6, 65. doi: 10.3390/risks6030065.

Hitaj, A.; Mateus, C.; Peri, I. Lambda Value at Risk and Regulatory Capital: A Dynamic Approach to Tail Risk. Risks 2018, 6, 17. doi: 10.3390/risks6010017.

Zhang, X.; Li, F.; Li, Z.; Xu, Y. Macroprudential Policy, Credit Cycle, and Bank Risk-Taking. Sustainability 2018, 10, 3620. doi: 10.3390/su10103620.

The second reviewer felt that the framework (scope) of the paper was not sufficiently relevant for an international audience because the authors of the article use Polish data and papers that are in Polish as references. The reviewer expressed concerns that many [of these articles or data] cannot be retrieved or read by most scholars.

However, we were able to provide access to all of the papers and databases (on the webpages) in English, except for three relevant contributions, which, although written in Polish, may be of interest to experts in Central and Eastern European studies. In the case of these three articles, we have provided the English translations and have made them available (the access is provided through the website) as was suggested by the Editor.

(“In view of one clearly positive report and the other one whose main concern addresses the list of references we recommend that the Authors take into account the related suggestions for improvement and (i) incorporate some 2-3 references they find most appropriate from the list provided, and, in addition, they take a proper care of the Polish titles. A suggested solution seems to be providing in addition (in parentheses?) their English translation and making them available (if they are not yet) in some open access repository.”)

The websites cannot be archived using WebCite because this platform is currently not accepting archiving requests. However, in the current form, all of the papers and data can be read or retrieved by most scholars. In addition, the details of the algorithms and the program were deposited in the “CoMSes, Open ABM, Big Data Hub” (open access repository).

The second reviewer “did respect much the present aim of the paper”, which is to analyse the impact of selected macro-prudential policy tools on the economic and financial system using agent-based modelling (ABM). The reviewer indicated that the research design was appropriate, the methods adequately described and that the conclusions supported by the results.

However, what requires clarification from our viewpoint is the fact that the simulation results are specific to small open economies. We are convinced that our contribution will receive the attention of an international audience. At the end of introductory section, we clarified why analyses of systemic risk in the case of a small open economy – and Poland specifically – are interesting and valuable:

“We conducted extended simulation experiments that were based on an ABM model that had been calibrated to reflect the features of a small open economy. Our choice was Poland as an exemplar case. The reason for calibrating the model relying on Polish data is that among the EU countries, the Polish economy is relatively small, open and strongly connected to the rest of the European Union countries. Moreover, the Polish banking system still remains strongly influenced by investors from the European Union, who treat Poland as a host country. Generally, the smaller countries of the CEE region that host foreign financial institutions are exposed to various dimensions of systemic risks more strongly. At the same time, the degree of the development of financial intermediation is relatively low, which results in a rather weak credit channel, especially in the case of investments. Although the financial system in Poland generates limited systemic risk, it is more vulnerable to regulatory arbitrage and the propagation of the shocks that are caused by the activity of international financial groups.

Consequently, the CEE economies and other emerging economies may need to conduct a more active macroprudential policy because of the higher risks that stem from volatile capital flows or credit booms etc. These issues also relate to the Polish economy and its financial system. Hence, both the ABM model and the simulations presented in our paper are valuable for gaining a detailed insight into the effects of macroprudential policy, especially in the case of small emerging open economies.

In order to study the macroeconomic effects of macroprudential instruments and their interaction with monetary policy in the case of a hypothetical small open economy, Aoki et al. (2016) applied a DSGE framework. The analysed model captured some critical features of the emerging market economies with macroprudential instruments that were defined as the capital requirements that were imposed on banks and a tax on FX lending. However, there are some relevant aspects that were not taken into consideration in the Aoki et al. (2016) model. For example, the possibility of the government or central bank intervening in the foreign exchange markets through the use of official foreign reserves is not discussed. Moreover, what is missing in the model is a more flexible specification of international capital flows (no equity flows or foreign direct investment) and the role of cross border gross flows, which could play a destabilizing role for financial stability. The ABM construct that is presented in our paper and the simulation study seem to be a step forward in addressing some of these issues, but in particular in relaxing the assumption of the homogeneity of the economic units that interact in a system”.

Aoki K., Benigno G., Kiyotaki N. (2016). Monetary and financial policies in emerging markets. Mimeo, Princeton University.

We calibrated the model using Polish data as an example of a small open economy in the CEE region. Calibrating or estimating agent-based models is an extremely complex process that requires access to disaggregated data and the use of novel techniques of statistical inference. Extensive agent-based macro-models, which cover many interactions between the financial system and the economy have so far been calibrated. In the case of less complex structures, an attempt was made to estimate the models (for a literature review see Platt (2019) [1]). The initially simulated minimum distance (SMD) method was used [2] with subsequent improvements [3]. In this method, an objective function is constructed by considering the weighted sums of the squared errors between the simulated and empirically measured moments. However, the choice of the moments is entirely arbitrary, and the quality of the associated parameter estimate depends critically on selecting a sufficiently comprehensive set of moments. The procedure of reducing the large computational burden that is imposed by the SMD method by replacing the costly model simulation process with computationally efficient surrogates was studied by Salle and Yildizoglu (2014) [4] and Lamperti et al. (2018) [5]. Alternative estimation techniques were also presented by Recchioni et al. (2015) [6] (a direct method for matching time series), Barde (2017) [7] and Lamperti (2017) [8] (information criterion methods), Kukucka and Barunik (2017) [9] (simulated maximum likelihood estimation) and Guerini and Moneta (2017) [10] (comparing the causal mechanisms that underlie real and simulated data using sVAR regressions).

In most cases, the presented estimation methods are frequentist in nature. Moving forward, our goal is to develop Bayesian estimation methods by referring to the studies that were initiated by Grazzini et al. [3].

Developing new techniques for calibrating and estimating in the case of large-scale macro-models is only possible on a specific dataset. In the model that we present in the article, we used empirical data to approximate the initial conditions (the values of the attributes) for 14 groups of banks, 18,727 firms, 8 sectors and 971,520 individuals and 442,240 households in the model. The states, that is, the values of the attributes were updated in the course of the simulation. The empirical data were also used to calculate specific functions (in decision rules). Just as in the first iteration, 90,000,000 data records were processed. At this stage, it would be difficult to replace the dataset for Poland with another dataset because preparing the complete database required months of work.

Nonetheless, to date, the results of our study have been presented at five international scientific conferences and the development of a new methods and their testing on the dataset for small open economy have been enthusiastically received by the international audience as was also evidenced by inter alia the submission of another EU grant about agent-based modelling for joint implementation with Prof. J. Doyne Farmer (the University of Oxford) in 2019.

[1] D. Platt. “A Comparison of Economic Agent-Based Model Calibration Methods”. INET Oxford/Mathematical Institute, University of Oxford.

[2] J. Grazzini and M. Richiardi. “Estimation of ergodic agent-based models by simulated minimum distance”. In: JEDC 51 (2015).

[3] Grazzini et al. “Bayesian estimation of agent-based models”. In: JEDC 51 (2017).

[4] I. Salle and Yildizoglu. “Efficient sampling and meta-modeling for computational economic models”. In: Computational Economics 44 (2014).

[5] Lamperti et al. “Agent-based model calibration using machine learning surrogates”. In: JEDC 90 (2018).

[6] Recchioni et al. “A calibration procedure for analyzing stock price dynamics in an agent-based framework”. In: JEDC 60 (2015).

[7] S. Barde. “A practical, accurate, information criterion for nth order Markov processes.” In: Computational Economics 50 (2017).

[8] F. Lamperti. “An information theoretic criterion for empirical validation of simulation models”. In: Econometrics and Statistics 5 (2017).

[9] J. Kukucka and J. Barunik. “Estimation of financial agent-based models with simulated maximum likelihood”. In: JEDC 85 (2017).

[10] M. Guerini and A. Moneta. “A method for agent-based models validation”. In: JEDC 82 (2017).

Finally, we would like to note that we adjusted the format and references according to the Instructions provided on the MDPI website.

https://www.mdpi.com/authors/references

https://www.mdpi.com/journal/entropy/instructions

Records no. [1], [4], [7], [10], [17], [24], [26], [42], [43], [49], [51], [52], [73], [77], [77], [79], [81], [97], [103], [110], [114], [121], [127], [128], [129], [132], [133], [134] and [135] in the references in the revised version of the paper were changed. The three articles in Polish that are mentioned are: [35], [83] and [90].

We greatly appreciate your contribution to this study and all of your constructive comments. We are submitting the revised version based on yours and the reviewers’ comments according to the timing designated by editorial office.

Kind regards,

Jagoda Kaszowska-Mojsa and Mateusz Pipień

Round 2

Reviewer 2 Report

no more comments, but the choice for a reviewer commenting on language usage in the manuscript is limited to, for example,

English language and style are fine/minor spell check required

or to

 I don't feel qualified to judge about the English language and style

there is no way to say "english is ok"